# Towards Robust Out-of-Distribution Generalization Bounds via Sharpness

**Yingtian Zou, Kenji Kawaguchi, Yingnan Liu, Jiashuo Liu*, Mong-Li Lee, Wynne Hsu**
School of Computing, National University of Singapore
Institute of Data Science, National University of Singapore
*Department of Computer Science & Technology, Tsinghua University, China

## Abstract

Generalizing to out-of-distribution (OOD) data or unseen domain, termed OOD generalization, still lacks appropriate theoretical guarantees. Canonical OOD bounds focus on different distance measurements between *source* and *target* domains but fail to consider the optimization property of the learned model. As empirically shown in recent work, the *sharpness* of learned minima influences OOD generalization. To bridge this gap between optimization and OOD generalization, we study the effect of sharpness on how a model tolerates data change in domain shift which is usually captured by "robustness" in generalization. In this paper, we give a rigorous connection between sharpness and robustness, which gives better OOD guarantees for robust algorithms. It also provides a theoretical backing for "flat minima leads to better OOD generalization". Overall, we propose a sharpness-based OOD generalization bound by taking robustness into consideration, resulting in a tighter bound than non-robust guarantees. Our findings are supported by the experiments on a ridge regression model, as well as the experiments on deep learning classification tasks.

## 1 Introduction

Machine learning systems are typically trained on a given distribution of data and achieve good performance on new, unseen data that follows the same distribution as the training data. Out-of-Distribution (OOD) generalization requires machine learning systems trained in the source domain to generalize to unseen data or target domains with different distributions from the source domain. A myriad of algorithms (Sun & Saenko, 2016; Arjovsky et al., 2019; Sagawa et al., 2019; Koyama & Yamaguchi, 2020; Pezeshki et al., 2021; Ahuja et al., 2021) aim to learn the invariant components along the distribution shifting. Optimization-based methods such as (El Ghaoui & Lebret, 1997; Duchi & Namkoong, 2018; Liu et al., 2021; Rame et al., 2022) focus on maximizing robustness by optimizing for worst-case error over an uncertainty distribution set. While these methods are sophisticated, they do not always perform better than Empirical Risk Minimization (ERM) when evaluated across different datasets (Gulrajani & Lopez-Paz, 2021; Wiles et al., 2022). This raises the question of how to understand the OOD generalization of algorithms and which criteria should be used to select models that are provably better (Gulrajani & Lopez-Paz, 2021). These questions highlight the need for more theoretical research in the field of OOD generalization (Ye et al., 2021).

To characterize the generalization gap between the source domain and the target domain, a canonical method (Blitzer et al., 2007) from domain adaptation theory decouples this gap into an In-Distribution (ID) generalization and a hypothesis-specific Out-of-Distribution (OOD) distance. However, this distance is based on the notion of VC-dimension (Kifer et al., 2004), resulting in a loose bound due to the large size of the hypothesis class in the modern overparameterized neural networks. Subsequent works improve the bound based on Rademacher Complexity (Du et al., 2017), whereas Germain et al. (2016) improves the bound based on PAC-Bayes. Unlike the present paper, these works did not consider algorithmic robustness, which has natural interpretation and advantages for distribution shifts In this work, we consider algorithmic robustness to derive the OOD generalization bound. The key idea is to partition the input space into $K$ non-overlapping subspaces such that the error difference in the model's performance between any pair of points in each subspace is bounded by some constant $\epsilon$. Within each subspace, any distributional shift is considered subtle for the robust

model thus leading to less impact on OOD generalization. Figure 1 illustrates this with the two distributions where the target domain has a distributional shift from the source domain. Compared to existing non-robust OOD generalization bounds Zhao et al. (2018), our new generalization error does not depend on hypothesis size, which is more reliable in the overparameterized regime. Our goal is to measure the generalizability of a model by considering how it is robust to this shift and achieves a tighter bound than existing works.

Although robustness captures the tolerance to distributional shift, it is intractable to compute robustness constant $\epsilon$ due to the inaccessibility of target distribution. The robustness definition in Xu & Mannor (2012) indicates that the loss landscape induced by the model's parameters is closely tied to its robustness. To gain a deeper understanding of robustness, we further study the learned model from an optimization perspective. As shown in (Lyu et al., 2022; Petzka et al., 2021), when the loss landscape is "flat", there is a good generalization, which is also observed in OOD settings (Izmailov et al., 2018; Cha et al., 2021). However, the relationship between robustness and this geometric property of the loss landscape, termed *Sharpness*, remains an open question. In this paper, we establish a provable dependence between robustness and sharpness for ReLU random neural network classes. It allows us to replace robustness constant $\epsilon$ with the sharpness of a learned model which is only computed from the training dataset that addresses the problem mentioned above. Our result of the interplay between robustness and sharpness can be applied to both ID and OOD generalization bounds. We also show an example to generalize our result beyond our assumption and validate it empirically.

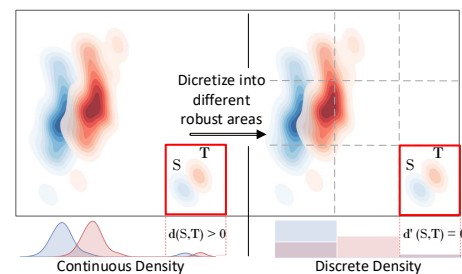

Figure 1: An example of a target distribution (red) directly translated from a source distribution (blue). The 1D density reflects the marginal distribution. Unlike existing works (left), we divide the distributions into disjoint partitions as a small change in distribution for a robust model is negligible (right). The sharpness of the model will decide the tolerance of change thus affecting the partitions. If two sub-distributions $S, T$ have small shifts such that they fall into the same partition (red grid), their distance measure $d'(S, T)$ by considering robustness will be zero.

Our main contributions can be summarized as follows:

- We proposed a new framework for Out-of-distribution/ Out-of-domain generalization bounds. In this framework, we use robustness to capture the tolerance of distribution shift which leads to tighter upper bounds generally.

- We reveal the underlying connection between the robustness and sharpness of the loss landscape and use this connection to enrich our robust OOD bounds under one-hidden layer ReLU NNs. This is the first optimization-based bound in Out-of-Distribution/Domain generalization.

- We studied two cases in ridge regression and classification which support and generalize our main theorem well. All the experimental results corroborate our findings well.

## 2 PRELIMINARY

**Notations**  We use $[n]$ to denote the integers set $\{i\}_{i=1}^n$. We use $\|\cdot\|$ to denote the $\ell_2$-norm (Euclidean norm). In vector form, $\boldsymbol{w}_i$ denotes the $i$-th instance while the $w_j$ is the $j$-th element of the vector $\boldsymbol{w}$ and we use $|\boldsymbol{w}|$ for the element-wise absolute value of vector $\boldsymbol{w}$. We use $n, d$ for the training set size and input dimension. $\mathcal{O}$ is the Big-O notation.

### 2.1 PROBLEM FORMULATION

Consider a source domain and a target domain of the OOD generalization problem where we use $\mathcal{D}_S, \mathcal{D}_T$ to represent the source and target distribution respectively. Let each $\mathcal{D}$ be the probability

measure of sample $z$ from sample space $\mathcal{Z} = \mathcal{X} \times \mathcal{Y}$ with $\mathcal{X} \in \mathbb{R}^d$. In the source domain, we have a training set $S = \{z_i\}_{i=1}^n, \forall i \in [n], z_i \sim \mathcal{D}_S$ while the target is to learn a model $f \in \mathcal{F}$ with $S$ and parameters $\boldsymbol{\theta} \in \Theta$ where $f : \Theta \times \mathcal{X} \mapsto \mathbb{R}$ generalizes well. Given loss function $\ell : \mathbb{R} \times \mathbb{R} \to \mathbb{R}_+$, which is for short, the expected risk over the source distribution $\mathcal{D}_S$ will be

$$\mathcal{L}_S(f_{\boldsymbol{\theta}}) \triangleq \mathbb{E}_{z \sim \mathcal{D}_S}[\ell_{\boldsymbol{\theta}}(z)] = \mathbb{E}_{z \sim \mathcal{D}_S}[\ell(f(\boldsymbol{\theta}, \boldsymbol{x}), y)], \quad \widehat{\mathcal{L}}_S(f_{\boldsymbol{\theta}}) \triangleq \frac{1}{n} \sum_{z_i \in S}[\ell_{\boldsymbol{\theta}}(z_i)].$$

We use $\ell_{\boldsymbol{\theta}}(z)$ as the shorthand. The OOD generalization is to measure between target domain expected risk $\mathcal{L}_T(f_{\boldsymbol{\theta}})$ and the source domain empirical risk $\widehat{\mathcal{L}}_S(f_{\boldsymbol{\theta}})$ which involves two parts: (1) In-domain generalization error gap between empirical risk and expected risk $\mathcal{L}_S(f_{\boldsymbol{\theta}})$ in the source domain. (2) Out-of-Domain distance between source and target domains. A model-agnostic example in Zhao et al. (2018) gave the following uniform bound:

**Proposition 2.1** (Zhao et al. Zhao et al. (2018) Theorem 2 & 3.2). *With hypothesis class $\mathcal{F}$ and pseudo dimension $\mathrm{Pdim}(\mathcal{F}) = d'$, unlabeled empirical datasets from source and target distribution $\widehat{\mathcal{D}}_S$ and $\widehat{\mathcal{D}}_T$ with size $n$ each, then with probability at least $1 - \delta$, for all $f \in \mathcal{F}$,*

$$\mathcal{L}_T(f) \leq \widehat{\mathcal{L}}_S(f) + \frac{1}{2} d_{\mathcal{F}\Delta\mathcal{F}}\left(\widehat{\mathcal{D}}_T; \widehat{\mathcal{D}}_S\right) + \mathcal{O}\left(\sqrt{d'/n}\right)$$

*where $d_{\mathcal{F}\Delta\mathcal{F}}(\widehat{\mathcal{D}}_T; \widehat{\mathcal{D}}_S) := 2 \sup_{A_f \in \mathcal{A}_{\{f(x) \oplus f'(x):f,f' \in \mathcal{F}\}}} \left| \mathbb{P}_{\widehat{\mathcal{D}}_S}[A_f] - \mathbb{P}_{\widehat{\mathcal{D}}_T}[A_f] \right|$ and $\oplus$ is the XOR operator. Specific form of $\mathcal{O}(\sqrt{|\mathcal{F}|/n})$ is defined in Appendix C.6.*

## 2.2 Algorithmic Robustness

**Definition 2.2** (Robustness, Xu & Mannor (2012)). A learning model $f_{\boldsymbol{\theta}}$ on training set $S$ is $(K, \epsilon(\cdot))$-robust, for $K \in \mathbb{N}$, if $\mathcal{Z}$ can be partitioned into $K$ disjoint sets, denoted by $\{C_i\}_{i=1}^K$, such that $\forall \boldsymbol{s} \in S$ we have

$$\boldsymbol{s}, \boldsymbol{z} \in C_i, \forall i \in [K] \Rightarrow |\ell_{\boldsymbol{\theta}}(\boldsymbol{s}) - \ell_{\boldsymbol{\theta}}(\boldsymbol{z})| \leq \epsilon(S).$$

This definition captures the robustness of the model in terms of the input. Within each partitioned set $C_i$, the loss difference between any sample $z$ belonging to $C_i$ and training sample $s \in C_i$ will be upper bounded by the robustness constant $\epsilon(S)$. The generalization result given by Xu & Mannor (2012) provides a framework to bound the empirical risk with algorithmic robustness which has been stated in Appendix C. Based on this framework, we are able to reframe the existing OOD generalization theory.

# 3 Main Results

In this section, we propose a new Out-of-Distribution (OOD) generalization bound for robust algorithms that have not been extensively studied yet. We then compare our result to the existing domain shift bound in Proposition 2.1 and discuss its implications for OOD and domain generalization problems by considering algorithmic robustness. To further explain the introduced robustness, we connect it to the *sharpness* of the minimum (a widely concerned geometric property in optimization) by showing a rigorous dependence between robustness and sharpness. This interplay will give us a better understanding of the OOD generalization problem, and meanwhile, provide more information on the final generalization bound. Detailed assumptions are clarified in Appendix B.1.

## 3.1 Robust OOD Generalization Bound

The main concern in OOD generalization is to measure the domain shift of a learned model. However, existing methods fail to consider the intrinsic property of the model, such as robustness. Definition 2.2 gives us a new robustness measurement to the model trained on dataset $S$ where the training set $S$ is a collection of *i.i.d.* data pair $(\boldsymbol{x}, y)$ sampled from source distribution $\mathcal{D}_S$ with size $n$. The measurement provides an intuition that if a test sample from the target domain is similar to a specific group of training samples, their losses will be similar as well. In other words, the model's robustness is a reflection of its ability to generalize to unseen data. Our first theorem shows that by utilizing the "robustness" measurement, we can more effectively handle domain shifts by setting a tolerance

range for distribution changes. This robustness measurement, therefore, provides a useful tool for addressing OOD generalization.

**Theorem 3.1.** *Let $\hat{\mathcal{D}}_T$ be the empirical distribution of size $n$ drawn from $\mathcal{D}_T$. Assume that the loss $\ell$ is upper bounded by $M$. With probability at least $1 - \delta$ (over the choice of the samples $S$), for every $f_{\boldsymbol{\theta}}$ trained on $S$ satisfying $(K, \epsilon(S))$-robust, we have*

$$\mathcal{L}_T(f_{\boldsymbol{\theta}}) \leq \widehat{\mathcal{L}}_S(f_{\boldsymbol{\theta}}) + M d_{(\epsilon,K)}(S, \hat{\mathcal{D}}_T) + 2\epsilon(S) + 3M\sqrt{\frac{2K \ln 2 + 2\ln(2/\delta)}{n}} \tag{1}$$

*where the total variation distance $d_{(\epsilon,K)}$ for discrete empirical distributions is defined by:*

$$\forall i \in [n], n_i(S) := \#(\boldsymbol{z} \in S \cap C_i), \quad d_{(\epsilon,K)}(S, \hat{\mathcal{D}}_T) := \sum_{i=1}^{K} \left| \frac{n_i(S)}{n} - \frac{n_i(\hat{\mathcal{D}}_T)}{n} \right| \tag{2}$$

*and $n_i(S), n_i(\hat{\mathcal{D}}_T)$ are the number of samples from $S$ and $\hat{\mathcal{D}}_T$ that fall into the set $C_i$, respectively.*

*Remark.* The result can be decomposed into in-domain generalization and out-domain distance $|\mathcal{L}_T(f_{\boldsymbol{\theta}}) - \mathcal{L}_S(f_{\boldsymbol{\theta}})|$ (please refer to Lemma C.1). Both of them depend on robustness $\epsilon(S)$.

See proof in Appendix C. The last three terms on the RHS of (1) are *distribution distance*, *robustness constant*, and *error term*, respectively. Unlike traditional distribution measures, we partition the sample space and the distributional shift separately in the $K$ sub-groups instead of measuring it point-wisely. We argue that the $d_{(\epsilon,K)}(S, \hat{\mathcal{D}}_T)$ can be zero measure if all small changes happen within the same partition where a 2D illustrative case is shown in Figure 1. Under the circumstances, our distribution distance term will be significantly smaller than Proposition 2.1 as the target distribution is essentially a perturbation of the source distribution. As a robust OOD generalization measure, our bound characterizes how robust the learned model is to negligible distributional perturbations. To prevent a bound that expands excessively, we also propose an alternate solution tailored for non-robust algorithms ($K \to \infty$) as follows.

**Corollary 3.2.** *If $K \to \infty$, the domain shift bound $|\mathcal{L}_T(f_{\boldsymbol{\theta}}) - \mathcal{L}_S(f_{\boldsymbol{\theta}})|$ can be replaced to the distribution distance in Proposition 2.1 where*

$$|\mathcal{L}_T(f_{\boldsymbol{\theta}}) - \mathcal{L}_S(f_{\boldsymbol{\theta}})| \leq \frac{1}{2} d_{\mathcal{F}\Delta\mathcal{F}}(S; \mathcal{D}_T) \leq \frac{1}{2} d_{\mathcal{F}\Delta\mathcal{F}}(S; \widehat{\mathcal{D}}_T) + \mathcal{O}(\sqrt{d'/n}) \tag{3}$$

*where the pseudo dimension $\mathrm{Pdim}(\mathcal{F}) = d'$.*

The proof is in Appendix C.1. As dictated in Theorem 3.1, when $K \to \infty$, the use infinite number of partitions on the data distribution leads to meaningless robustness. However, Corollary C.7 suggests that our bound can be replaced by $d_{\mathcal{F}\Delta\mathcal{F}}(\mathcal{D}_S; \mathcal{D}_T)$ in the limit of infinite $K$. This avoids computing a vacuous bound for non-robust algorithms. In summary, Theorem 3.1 presents a novel approach for quantifying distributional shifts by incorporating the concept of robustness. Our framework is particularly beneficial when a robust algorithm is able to adapt to local shifts in the distribution. Additionally, our data-dependent result remains valid and useful in the overparameterized regime, since $K$ does not depend on the model size.

## 3.2 SHARPNESS AND ROBUSTNESS

Clearly, robustness is inherently tied to the optimization properties of a model, particularly the curvature of the loss landscape. One direct approach to characterize this geometric curvature, referred to as "sharpness," involves analyzing the Hessian matrix (Foret et al., 2020; Cohen et al., 2021). Recent research (Petzka et al., 2021) has shown that the concept of "relative flatness", the *sharpness* in this paper, has a strong correlation with model generalization. However, the impact of relative flatness on OOD generalization remains uncertain, even within the convex setting. To address this problem, we aim to investigate the interplay between robustness and sharpness. With the following definition of sharpness, we endeavor to establish an OOD generalization bound rooted in optimization principles.

**Definition 3.3** (Sharpness, Petzka et al. (2021)). For a twice differentiable loss function $\mathcal{L}(\boldsymbol{w}) = \sum_{\boldsymbol{s} \in S} \ell_{\boldsymbol{w}}(\boldsymbol{s})$, $\boldsymbol{w} \in \mathbb{R}^m$ with a sample set $S$, the sharpness is defined by

$$\kappa(\boldsymbol{w}, S, A) := \langle \boldsymbol{w}, \boldsymbol{w} \rangle \cdot \mathrm{tr}\left(H_{S,A}(\boldsymbol{w})\right) \tag{4}$$

where $H_{S,A}$ is the Hessian matrix of loss $\mathcal{L}(\boldsymbol{w})$ w.r.t. $\boldsymbol{w}$ with hypothesis set $A$ and input set $S$.

As per Definition 3.3, sharpness is characterized by the sum of all the eigenvalues of the Hessian matrix, scaled by the parameter norm. Each eigenvalue of the Hessian reflects the rate of change of the loss derivative in the corresponding eigenspace. Therefore the smaller value of $\kappa$ indicates a flatter minimum. In Cha et al. (2021), they suggest that flatter minima will improve the OOD generalization, but fail to deliver an elaborate analysis of the Hessian matrix. In this section, we begin with the random ReLU Neural Networks parameterized by $\boldsymbol{\theta} = (\{\boldsymbol{a}_i\}_{i \in [m]}, \boldsymbol{w})$ where $\boldsymbol{w} = [w_1, ..., w_m]^\top$ is the trainable parameter. Let $A = [\boldsymbol{a}_1, ..., \boldsymbol{a}_m]$, the whole function class is defined as

$$f(\boldsymbol{w}, A, \boldsymbol{x}) \triangleq \frac{1}{\sqrt{d}} \sum_{i=1}^{m} w_i \sigma\left(\boldsymbol{x}, \boldsymbol{a}_i\right) : w_i \in \mathbb{R}, \boldsymbol{a}_i \sim \text{Unif}(\mathbb{S}^{d-1}(\sqrt{d})), i \in [m] \quad (5)$$

where $\sigma(\cdot)$ is the ReLU activation function and $\boldsymbol{a}$ are random vectors uniformly distributed on $n$-dim hypersphere whose surface is a $n-1$ manifold. We then define any convex loss function $\ell(f(\boldsymbol{w}, A, \boldsymbol{x}), y) : \mathbb{R} \times \mathbb{R} \to \mathbb{R}_+$. The corresponding empirical minimizer in the source domain will be: $\hat{\boldsymbol{w}} = \arg\min_{\boldsymbol{w}} \frac{1}{n} \sum_{i=1}^{n} \ell(f(\boldsymbol{w}, A, \boldsymbol{x}_i), y_i)$. With $\hat{\boldsymbol{w}}$, we are interested in loss geometry over the sample domain ($\ell_{\hat{\boldsymbol{w}}, A}(\boldsymbol{z})$ for short). Intuitively, a flatter minimum on the loss landscape is expected to be more robust to varying input. Suppose the sample space $\mathcal{Z}$ can be partitioned into $K$ disjoint sets. For each set $C_i, i \in [K]$, the loss difference is upper bounded by $\epsilon(S, A)$. Given $\boldsymbol{z} \in S$, we have

$$\epsilon(S, A) \triangleq \max_{i \in [K]} \sup_{\boldsymbol{z}, \boldsymbol{z}' \in C_i} |\ell_{\hat{\boldsymbol{w}}, A}(\boldsymbol{z}) - \ell_{\hat{\boldsymbol{w}}, A}(\boldsymbol{z}')| . \quad (6)$$

As an alternative form of robustness, the $\epsilon(S, A)$ in (6) captures the "maximum" loss difference between any two samples in each partition and depends on the convexity and smoothness of the loss function in the input domain. Given a training set $S$ and any initialization $\boldsymbol{w}_0$, the robustness $\epsilon(S, A)$ of a learned model $f_{\hat{\boldsymbol{w}}}$ will be determined. It explicitly reflects the smoothness of the loss function in each (pre-)partitioned set. Nevertheless, its connection to the sharpness of the loss function in parameter space still remains unclear. In order to address this gap, we establish a connection between sharpness and robustness in Theorem 3.4. Notably, this interplay holds implications not only for OOD but also for in-distribution generalization.

**Theorem 3.4.** *Assume for any $A$, the loss function $\ell_{\hat{\boldsymbol{w}}, A}(\boldsymbol{z})$ w.r.t. sample $\boldsymbol{z}$ satisfies the $L$-Hessian Lipschitz continuity (refer to Definition B.2) within every set $C_i, \forall i \in [K]$. Let $\boldsymbol{z}_i(A) = \arg\max_{\boldsymbol{z} \in C_i \cap S} \ell_{\hat{\boldsymbol{w}}, A}(\boldsymbol{z})$. Define $\mathcal{M}_i$ to be the set of global minima in $C_i$, suppose $\exists \boldsymbol{z}_i^*(A) \in \mathcal{M}_i$ such that for some $\rho_i(L) > 0$, $\|\boldsymbol{z}_i(A) - \boldsymbol{z}_i^*(A)\| \leq \frac{\rho_i(L)}{L}$ almost surely, then let $\rho_{\max}(L) = \max\{\rho_i(L), i \in [K]\}$, $\|\boldsymbol{x}\|^2 \equiv R(d)$ and $n' \leq n, \in \mathbb{N}^+$, w.p $p = \min\left\{\frac{2}{\pi} \arccos\left(R(d)^{-\frac{1}{2}}\right), \left|1 - \frac{\sqrt{2d-4}}{\sqrt{\pi R(d)}} e^{\frac{1}{4d-9}}\right|\right\}$ over $\{\boldsymbol{a}_i\}_{i=1}^{m} \sim \text{Unif}(\mathbb{S}^{d-1}(\sqrt{d}))$ we have*

$$\epsilon(S, A) \leq \frac{\rho_{\max}(L)^2}{2L^2} \left(\left[n' + \mathcal{O}\left(\frac{d}{m}\right)\right] \kappa(\hat{\boldsymbol{w}}, S, A) + \frac{4\rho_{\max}(L)}{3}\right). \quad (7)$$

*Remark.* Given the training set $S$, we can estimate factor $\hat{n}$ that $n' \leq \hat{n}$ by comparing the maximum Hessian norm w.r.t. $\boldsymbol{z}_j$ to the sum of all the Hessian norms over $\{\boldsymbol{z}_i\}_{i \in [n]}$. Note that the smoothness condition only applies to every partitioned set (locally) where it is much weaker than the global requirement for the loss function to be satisfied We also discuss the difference between our results and Petzka et al. (2021) in Appendix F. The chosen family of loss functions that applied to our theorem can be found in Appendix B.1.

**Corollary 3.5.** *Let $\hat{w}_{\min}$ be the minimum value of $|\hat{\boldsymbol{w}}|$. Suppose $\forall \boldsymbol{x} \sim \text{Unif}(\mathbb{S}^{d-1}(\sqrt{d}))$ and $|\partial^2 \ell(f(\hat{\boldsymbol{w}}, A, \boldsymbol{x}), y)/\partial f^2|$ is bounded by $[\tilde{M}_1, \tilde{M}_2]$. If $m = Poly(d), d > 2, \rho_{\max}(L) < (\hat{w}_{\min}^2 \tilde{M}_1 \tilde{\sigma}(d, m))/(2d)$ taking expectation over all $\boldsymbol{x}_j \in S, j \in [n]$ and all $\boldsymbol{a}_i \in A \sim \text{Unif}(\mathbb{S}^{d-1}(\sqrt{d})) \forall i \in [m]$, we have*

$$\mathbb{E}_{S, A}[\epsilon(S, A)] \leq \mathbb{E}_{S, A} \frac{7\rho_{\max}(L)^2}{6L^2} \left(n' \kappa(\hat{\boldsymbol{w}}, S, A) + \tilde{M}_2\right). \quad (8)$$

*where $\tilde{\sigma}(d, m) = \mathbb{E}_{\boldsymbol{a} \sim \text{Unif}(\mathbb{S}^{d-1}(\sqrt{d}))} \lambda_{\min}(\sum_{i=1}^{m} \boldsymbol{a}_i \boldsymbol{a}_i^\top G_{ii}) > 0$ is the minimum eigenvalue and $G_{ii}$ is product constant of Gegenbauer polynomials (definition can be founded in Appendix B).*

See proof in Appendix D. From Theorem 3.4 we can see, the robustness constant $\epsilon(S, A)$ is (pointwise) upper bounded by the sharpness of the learned model, as measured by the quantity $\kappa(\hat{\boldsymbol{w}}, S, A)$, and the parameter $\rho_{\max}(L)$. It should be noted that the parameter $\rho_{\max}(L)$ depends on the partition, and as the number of partitions increases, the region $\rho_{\max}(L)$ of the input domain becomes smaller, thereby making sharpness the dominant term in reflecting the model's robustness within the small local area. In Corollary 3.5, we show a stronger connection when the partition satisfies some conditions. Overall, this bound states that the larger the sharpness of the model $\kappa(\hat{\boldsymbol{w}}, S, A)$, the larger the upper bound on the robustness parameter $\epsilon(S, A)$. This result aligns with the intuition that a sharper model is more prone to overfitting the training domain and is less robust in the unseen domain. While the dependency is not exact, it still can be regarded as an alternative approach that avoids the explicit computation of the intractable robustness term. By substituting this upper bound for $\epsilon(S, A)$ into Theorem 3.1, we derive a sharpness-based OOD generalization bound. This implies that the OOD generalization error will have a high probability of being small if the learned model is flat enough. Unlike existing works, our generalization bound provides more information about how optimization property influences performance when generalizing to OOD data. It bridges the gap between robustness and sharpness which can also be generalized to non-OOD learning problems. Moreover, we provide a better theoretical grounding for an empirical observation that a flat minimum improves domain generalization (Cha et al., 2021) by pinpointing a clear dependence on sharpness.

## 3.3 CASE STUDY

To better demonstrate the relationship between sharpness and robustness, we provide two specific examples: (1) linear ridge regression; (2) two-layer diagonal neural networks for classification.

**Example 3.6.** *In ridge regression models, $\epsilon(S, A)$ has a reverse relationship to the regularization parameter $\beta$. $\beta \uparrow$, the more probably flatter minimum $\kappa \downarrow$ and less sensitivity $\epsilon \downarrow$ of the learned model could be. Following the previous notation, we have $\exists \tilde{c}_1 > 0$ such that $\epsilon(S, A) \leq \tilde{c}_1 \kappa(\hat{\boldsymbol{\theta}}, S) + \tilde{o}_d$ where $\tilde{o}_d$ has a smaller order than $\kappa(\hat{\boldsymbol{\theta}}, S)$ for large $d$ (proof refer to Appendix E.1).*

As suggested in Ali et al. (2019), let's consider a generic response model $\boldsymbol{y}|\boldsymbol{\theta}_* \sim (X\boldsymbol{\theta}_*, \sigma^2 I)$ where $X \in \mathbb{R}^{n \times d}, d > n$. The least-square empirical minimizer of the ridge regression problem will be:

$$\hat{\boldsymbol{\theta}} = \arg\min_{\boldsymbol{\theta}} \frac{1}{2n}\|X\boldsymbol{\theta} - \boldsymbol{y}\|^2 + \frac{\beta}{2}\|\boldsymbol{\theta}\|^2 = (X^\top X + n\beta I_n)^{-1} X^\top \boldsymbol{y} = M^{-1} X^\top \boldsymbol{y} \qquad (9)$$

Let $S$ be the training set. It's trivial to get the sharpness of a quadratic loss function where

$$\kappa(\hat{\boldsymbol{\theta}}, S) = \|\hat{\boldsymbol{\theta}}\|^2 \operatorname{tr}(X^\top X/n + \beta I_n) = \|\hat{\boldsymbol{\theta}}\|^2 \operatorname{tr}(M) \qquad (10)$$

It's obvious that both of the above two equations depend on the same matrix $M = X^\top X/n + \beta I_n$. For fixed training samples $X$, we have $\kappa(\hat{\boldsymbol{\theta}}, S) = \mathcal{O}(\beta^{-1})$ in the limit of $\beta$. Then it's clear that a higher penalty $\beta$ leads to a flatter minimum. This intuition is rigorously proven in Appendix E.1. According to Theorem 3.1 and Theorem 3.4, a flatter minimum probably associates with lower robustness constant $\epsilon(S, A)$. Thus it enjoys a lower OOD generalization error gap. In ridge regression, this phenomenon can be reflected by the regularization coefficient $\beta$. Therefore, in general, the larger $\beta$ is, the lower the sharpness $\kappa(\hat{\boldsymbol{\theta}}, S)$ and variance are. As a consequence, larger $\beta$ learns a more robust model resulting in a lower OOD generalization error gap. This idea is later verified in the distributional shift experiments, shown as Figure 2.

**Example 3.7.** *We consider a classification problem using a 2-layer diagonal linear network with exp-loss. The robustness $\epsilon(S, A)$ has a similar relationship in Theorem 3.4. Given training set $S$, after iterations $t > T_\epsilon$, $\exists \tilde{c}_2 > 0, \epsilon(S, A) \leq \tilde{c}_2 \sup_{t \geq T_\epsilon} \kappa(\boldsymbol{\theta}(t), S)$.*

In addition to the regression and linear models, we have obtained a similar relationship for 2-layer diagonal linear networks, which are commonly used in the kernel and rich regimes as well as in intermediate settings (Moroshko et al., 2020). Example 3.7 demonstrates that the relationship also holds true when the model is well-trained, even exp-loss does not satisfy the PŁ condition. By extending our theorems to these more complex frameworks, we go beyond our initial assumptions and offer insights into broader applications. Later experiments on non-linear NN also support our statements. However, we still need a unified theorem for general function classes with fewer assumptions.

## 4 RELATED WORK

Despite various methods (Sun & Saenko, 2016; Sagawa et al., 2019; Shi et al., 2021; Shafieezadeh Abadeh et al., 2015; Li et al., 2018; Cha et al., 2021; Du et al., 2020; Zhang et al., 2022) that have been proposed to overcome the poor generalization brought by unknown distribution shifts, the underlying principles and theories still remain underexplored. As pointed out in Redko et al. (2020); Miller et al. (2021), different tasks that address distributional shifts, such as domain adaptation, OOD, and domain generalization, are collectively referred to as "transfer transductive learning" and share similar generalization theories. In general, the desired generalization bound will be split into In-Distribution/Domain (ID) generalization error and Out-of-Distribution/Domain (OOD) distance. Since Blitzer et al. (2007) establish a VC-dimension-based framework to estimate the domain shift gap by a divergence term, many following works make the effort to improve this term in the following decades, such as Discrepancy (Mansour et al., 2009), Wasserstein measurement (Courty et al., 2017; Shen et al., 2018), Integral Probability Metrics (IPM) (Zhang et al., 2019b; Ye et al., 2021) and $\beta$-divergence (Germain et al., 2016). Among them, new generalization tools like PAC-Bayes, Rademacher Complexity, and Stability are also applied. However, few of them discuss how the sharpness reacts to data distributional shifts.

Beyond this canonical framework, Ye et al. (2021) reformulate the OOD generalization problem and provide a generalization bound using the concepts of "variation" and "informativeness." The causal framework proposed in Peters et al. (2017); Rojas-Carulla et al. (2018) focuses on the impact of interventions on robust optimization over test distributions. However, none of these frameworks consider the optimization process of a model and how it affects OOD generalization. Inspired by previous investigation on the effect of sharpness on ID generalization (Lyu et al., 2022; Petzka et al., 2021), recent work in Cha et al. (2021) found that flatter minima can also improve OOD generalization. Nevertheless, they lack a sufficient theoretical foundation for the relationship between the "sharpness" of a model and OOD generalization, but end with a union bound of Blitzer et al. (2007)'s result. In this paper, we aim to provide a rigorous examination of this relationship.

## 5 EXPERIMENTS

In light of space constraints, we present only a portion of our experimental results to support the validity of our theorems and findings. For comprehensive results, please refer to the Appendix G

### 5.1 RIDGE REGRESSION IN DISTRIBUTIONAL SHIFTING

Following Duchi & Namkoong (2021), we investigated the ridge regression on distributional shift. We randomly generate $\theta_0^* \in \mathbb{R}^d$ in spherical space, and data from the following generating process: $X \overset{\text{iid}}{\sim} \mathcal{N}(0,1), \quad \boldsymbol{y} = X\theta_0^*$. To simulate distributional shift, we randomly generate a perpendicular unit vector $\theta_0^\perp$ to $\theta_0^*$. Let $\theta_0^\perp, \theta_0^*$ be the basis vectors, then shifted ground-truth will be computed from the basis by $\theta_\alpha^* = \theta_0^* \cdot \cos(\alpha) + \theta_0^\perp \cdot \sin(\alpha)$. For the source domain, we use $\theta_0^*$ as our training distribution. We randomly sample 50 data points and train a linear classifier with a gradient descent of 3000 iterations. By minimizing the objective function in (9), we can get the empirical optimum $\hat{\theta}$. Then we gradually shift the distribution by increasing $\alpha$ to get different target domains. Along distribution shifting, the test loss $\ell(\hat{\theta}, \boldsymbol{y}_\alpha)$ will increase. As shown in Figure 2, the test loss will culminate in around 3 rads due to the maximum distribution shifting. Comparing different levels of regularization, we found that the larger L2-penalty $\beta$ brings lower OOD generalization error which is shown as darker purple lines. This plot bears out our intuition in the previous section. As stated in the aforementioned case, the sharpness of ridge regression should inversely depend on $\beta$. Correspondingly, we compute sharpness using the definition equation (4) by averaging ten different results. For each trial, we use the same training and test data for every $\beta$. The sharpness of each ridge regressor is shown in the legend of Figure 2. As we can see, larger $\beta$ leads to less sharpness.

### 5.2 SHARPER MINIMUM HURTS OOD GENERALIZATION

In our results, we proved that the upper bound of OOD generalization error involves the sharpness of the trained model. Here we empirically verified our theoretical insight. We follow the experiment setting in DomainBed (Gulrajani & Lopez-Paz, 2021). To easily compute the sharpness, we choose

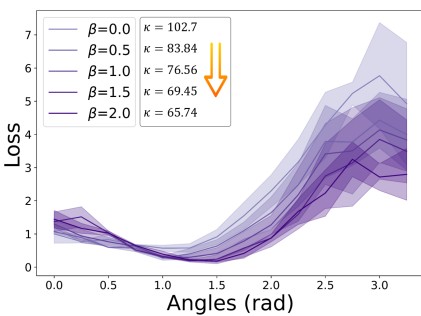

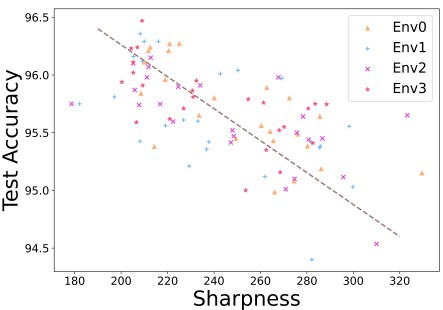

Figure 2: OOD test losses increase along distributional shifting. The X-axis is the shifting angle $\alpha$ and the Y-axis is the test loss of the model which is trained on distribution $\alpha = 0$. Lines are average test losses and shadows are variances of 10 trials. Larger regularization $\beta$ (darker color) causes a lower increase in test loss but smaller sharpness.

Figure 3: The relationship between out-of-domain test accuracy and model sharpness on RotatedMNIST dataset. Here we show 4 different OOD environments: $15°, 30°, 45°, 60°$ rotation as the OOD test set respectively. Each marker denotes a minimum of an algorithm with a specific seed. The marker style means the models trained in the same environment.

the 4-layer MLP on RotatedMNIST dataset where RotatedMNIST is a rotation of MNIST handwritten digit dataset (LeCun, 1998) with different angles ranging from $[0°, 15°, 30°, 45°, 60°, 75°]$. In this codebase, each environment refers to selecting a domain (a specific rotation angle) as the test domain/OOD test dataset while training on all other domains. After getting the trained model of each environment, we compute the sharpness using all domain training sets based on the implementation of Petzka et al. (2021). To this end, we plot the performances of Empirical Minimization Risk (ERM), SWAD (Cha et al., 2021), Mixup (Yan et al., 2020) and GroupDRO (Sagawa et al., 2019) with 6 seeds of each. Then we measure the sharpness of all these minima. Figure 3 shows the relationship between model sharpness and out-of-domain accuracy. The tendency is clear that flat minima give better OOD performances. In general, different environments can not be plotted together due to different training sets. However, we found the middle 4 environments are similar tasks and thus plot them together for a clearer trend. In addition, different algorithms lead to different feature scales which may affect the scale of the sharpness. To address this, we align their scales when putting them together. For more individual results, please refer to Figure 8 in the appendix.

## 5.3 COMPARISON OF GENERALIZATION BOUNDS

To analyze our generalization bounds, we follow the toy example experiments in Sagawa et al. (2020). In this experiment, the distribution shift terms and generalization error terms can be explicitly computed. Furthermore, their synthetic experiment considers the spurious correlation across distribution shifts which is now a general formulation of OOD generalization (Wald et al., 2021; Aubin et al., 2021; Yao et al., 2022). Consider data $x = [x_{\text{core}}, x_{\text{spu}}] \in$ that consist of two features: core feature and spurious feature. The features are generated from the following rule:

$$x_{\text{core}} \mid y \sim \mathcal{N}\left(y\mathbf{1}, \sigma_{\text{core}}^2 I_d\right) \quad x_{\text{spu}} \mid a \sim \mathcal{N}\left(a\mathbf{1}, \sigma_{\text{spu}}^2 I_d\right)$$

where $y \in \{-1, 1\}$ is the label, and $a \in \{-1, 1\}$ is the spurious attribute. Data with $y = a$ forms the majority group of size $n_{\text{maj}}$, and data with $y = -a$ forms minority group of size $n_{\text{min}}$. Total number of training points $n = n_{\text{maj}} + n_{\text{min}}$. The spurious correlation probability, $p_{\text{maj}} = \frac{n_{\text{maj}}}{n}$ defines the probability of $y = a$ in training data. In testing, we always have $p_{\text{maj}} = 0.5$. The metric, worst-group error Sagawa et al. (2019) is defined as

$$\text{Err}_{\text{wg}}(w) := \max_{i \in [4]} \mathbb{E}_{x, y \mid g_i}\left[\ell_{0-1}(w; (x, y))\right]$$

where $\ell_{0-1}$ is the $0-1$ loss in binary classification. Here we compare the robustness of our proposed OOD generalization bound and the baseline in Proposition 2.1. We also give the comparison to other baselines, like PAC-Bayes DA bound in the Appendix G.

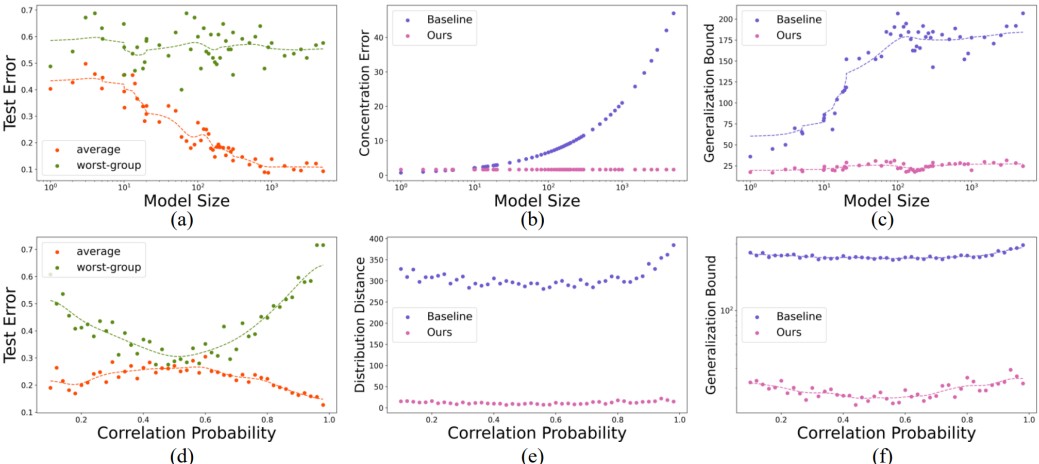

Figure 4: Spurious feature synthetic experiment. Each dot represents a trained model. The dash curves are the smoothed function fit by the test data points. The baseline is Proposition 2.1. **(a),(d)**: the generalization error of the logistic regression models with increasing the model size/correlation probability. **(b)**: concentration error term in domain shift bound. **(e)**: comparison of distribution distance bounds. **(c),(f)**: comparisons of generalization bounds. Note that model size $> 500$ is the overparameterized regime. The further the correlation probability is from $0.5$, the greater the distributional shift is.

**Along model size** We plot the generalization error of the random feature logistic regression along the model size increases in Figure 4(a). In this experiment, we follow the hyperparameter setup of Sagawa et al. (2020) by setting the number of points $n = 500$, data dimension $2d = 200$ with 100 on each feature. majority fraction $p_{\text{maj}} = 0.9$ and noises $\sigma^2_{\text{spu}} = 1, \sigma^2_{\text{core}} = 100$. The worst-group error turns out to be nearly the same as the model size increases. However, in Figure 4(b), the error term in domain shift bound Proposition 2.1(A) will keep increasing when the model size is increasing. In contrast, our domain shift bound at order $\sqrt{K}$ is independent of the model size which addresses the limitation of their bound. We follow Kawaguchi et al. (2022) to compute $K$ in an inverse image of the $\epsilon$-covering in a randomly projected space (see details in appendix). We set the same value $K = 1,000$ in our experiment. Different from the baseline, $K$ is data dependent and leads to a constant concentration error term along with model size increases. Analogously, our OOD generalization bound will not explode as model size increases (shown in Figure 4(c)).

**Along distribution shift** In addition, we are interested in characterizing OOD generalization when test distribution shifts from train distribution by varying the correlation probability $p_{\text{maj}}$ during data generation. As shown in Figure 4(d), when $p_{\text{maj}} = 0.5$, there is no distributional shift between training and test data due to no spurious features correlated to training data. Thus, the training and test distributions align closer and closer when $p_{\text{maj}} < 0.5$ and increase, resulting in an initial decrease in the test error for the worst-case group. However, as $p_{\text{maj}} > 0.5$ and deviates from 0.5, introducing the spurious features, a shift in the distribution occurs. This deviation is likely to impact the worst-case group differently, leading to an increase in the test error. As displayed in Figure 4(e) and Figure 4(f), our distribution distance and generalization bound can capture the distribution shifts but are tighter than the baseline.

## 6 CONCLUSION

In this paper, we provide a more interpretable and informative theory to understand Out-of-Distribution (OOD) generalization Based on the notion of robustness, we propose a robust OOD bound that effectively captures the algorithmic robustness in the presence of shifting data distributions. In addition, our in-depth analysis of the relationship between robustness and sharpness further illustrates that sharpness has a negative impact on generalization. Overall, our results advance the understanding of OOD generalization and the principles that govern it.

## 7 ACKNOWLEDGEMENT

This research/project is supported by the National Research Foundation, Singapore under its AI Singapore Programme (AISG Award No: AISG-GC-2019-001-2A), (AISG Award No: AISG2-TC-2023-010-SGIL) and the Singapore Ministry of Education Academic Research Fund Tier 1 (Award No: T1 251RES2207). We would appreciate the contributions of Fusheng Liu, Ph.D. student at NUS.

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

# A   ADDITIONAL EXPERIMENTS

## A.1   SHARPNESS V.S. OOD GENERALIZATION ON PACS AND WILDS-CAMELYON17

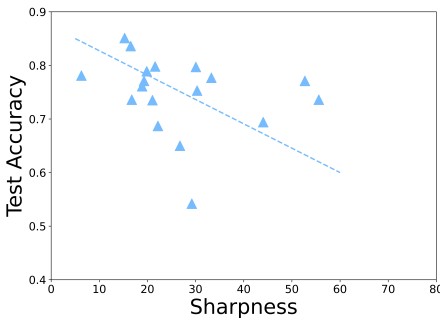
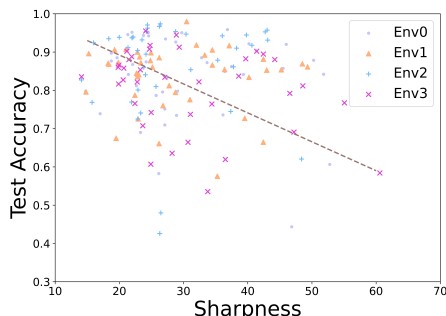

Figure 5: The relationship between out-of-distribution (OOD) test accuracy on the test environment and model sharpness (of last FC layer) on the **Wilds-Camelyon17** dataset. Each marker denotes a model trained using ERM with different seed and hyperparameters.

Figure 6: The relationship between OOD test accuracy and model sharpness on the **PACS** dataset. Each marker denotes a model trained using ERM with different seeds and hyperparameters. The marker style shows the out-of-distribution test environment of the model.

To evaluate our theorem more deeply, we examine the relationship between our defined sharpness and OOD generalization error on larger-scale real-world datasets, **Wilds-Camelyon17** Bandi et al. (2018); Koh et al. (2021) and **PACS** Li et al. (2017). Wilds-Camelyon17 dataset includes 455,954 tumor and normal tissue slide images from five hospitals (environments). One of the hospitals is assigned as the test environment by the dataset publisher. Distribution shift arises from variations in patient population, slide staining, and image acquisition. PACS dataset contains 9,991 images of 7 objects in 4 visual styles (environments): art painting, cartoon, photo, and sketch. Following the common setup in Gulrajani & Lopez-Paz (2021), each environment is used as a test environment in turn. We follow the practice in Petzka et al. (2021) to compute the sharpness using the Hessian matrix from the last Fully-Connected (FC) layer of each model. For the Wilds-Camelyon17 dataset, we test the sharpness of 18 ERM models trained with different random seeds and hyperparameters. Figure 5 shows the result. For the PACS dataset, we run 60 ERM models with different random seeds and hyperparameters for each test environment. To get a clearer correlation, we align the points from 4 environments by their mean performance. Figure 6 shows the result. From the two figures, we can observe a clear correlation between sharpness and out-of-distribution (OOD) accuracy. Sharpness tends to hurt the OOD performance of the model. The result is consistent with what we report in Figure 3. It shows that the correlation between sharpness and OOD accuracy can also be observed on large-scale datasets.

# B   NOTATIONS AND DEFINITIONS

**Notations**   We use $[n]$ denote the integers set $\{i\}_{i=1}^{n}$. $\|\cdot\|$ represents $\ell_2$-norm $\|\cdot\|_2$ for short. Without loss of generality, we use $\ell(f(\boldsymbol{\theta}, \boldsymbol{x}), \boldsymbol{y})$ for the loss function of model $f_{\boldsymbol{\theta}}$ on data pair $\boldsymbol{z} = (\boldsymbol{x}, \boldsymbol{y})$, which is denoted as $\ell_{\boldsymbol{\theta}}(\boldsymbol{z})$ and we use $n, d$ for training set size and input dimension. Note that we generally follow the notations in the original papers.

- $\mathcal{L}_S, \mathcal{L}_T$: expected risk of the source domain and target domain, respectively. The corresponding empirical version will be $\widehat{\mathcal{L}}_S, \widehat{\mathcal{L}}_T$

- $\{C_i\}_{i=1}^{K}$: $K$ partitions on sample space and $C_i$ denotes each partitioned set.

- $\mathcal{D}_S, \mathcal{D}_T$: distributions of source and target domain. Their sampled dataset will be denoted as $\hat{\mathcal{D}}_S, \hat{\mathcal{D}}_T$ accordingly.

- $\boldsymbol{\theta}$: In our setting, $\boldsymbol{\theta} = (\boldsymbol{w}, \{\boldsymbol{a}\}_i^m)$ denotes the model parameters where $\boldsymbol{w}$ is the trainable parameters and $\{\boldsymbol{a}_i\}^m$ are the random features (we also use $A = [\boldsymbol{a}_1, ..., \boldsymbol{a}_m]$ for short notation in many places). $\hat{\boldsymbol{w}}$ is the minimizer of empirical loss.

- $M$: upper bound of the loss function.

- $S = \{(\boldsymbol{x}_i, y_i)\}_i^n / X = [\boldsymbol{x}_1, ...\boldsymbol{x}_n]$: training data of size $n$.

**Definition B.1** (Robustness, Xu & Mannor (2012)). A learning algorithm $\mathcal{A}$ on training set $S$ is $(K, \epsilon(\cdot))$-robust, for $K \in \mathbb{N}$, if $\mathcal{Z}$ can be partitioned into $K$ disjoint sets, denoted by $\{C_k\}_{k=1}^K$, such that for all $s \in S, z \in \mathcal{Z}$ we have

$$\forall s, z \in C_i, \forall k \in [K], |\ell(\mathcal{A}_S, s) - \ell(\mathcal{A}_S, z)| \le \epsilon(S, A).$$

**Definition B.2** (Hessian Lipschitz continuous). For a twice differentiable function $f : \mathbb{R}^n \to \mathbb{R}$, it has $L$-Lipschitz continuous Hessian for domain $\boldsymbol{x}, \boldsymbol{y}$ are vectors in $C_i$ if

$$\left\| \nabla^2 f(\boldsymbol{y}) - \nabla^2 f(\boldsymbol{x}) \right\| \le L_i \|\boldsymbol{y} - \boldsymbol{x}\|$$

where $L_i > 0$ depends on input domain $C_i$ and $\| \cdot \|$ is $L_2$ norm. Then for all $K$ domains $\cup_{i=1}^K C_i$, let $L := \max\{L_i | i \in [K]\}$ be the uniform Lipschitz constant, so we have

$$\left\| \nabla^2 f(\boldsymbol{y}) - \nabla^2 f(\boldsymbol{x}) \right\| \le L_i \|\boldsymbol{y} - \boldsymbol{x}\| \le L\|\boldsymbol{y} - \boldsymbol{x}\|, \forall i \in [K], (\boldsymbol{x}, \boldsymbol{y}) \in C_i$$

which is uniformly bounded with $L$.

**Lemma B.3** (Hessian Lipschitz Lemma). *If $f$ is twice differentiable and has $L$-Lipschitz continuous Hessian, then*

$$\left| f(\boldsymbol{y}) - f(\boldsymbol{x}) - \langle \nabla f(\boldsymbol{x}), \boldsymbol{y} - \boldsymbol{x} \rangle - \frac{1}{2} \left\langle \nabla^2 f(\boldsymbol{x})(\boldsymbol{y} - \boldsymbol{x}), (\boldsymbol{y} - \boldsymbol{x}) \right\rangle \right| \le \frac{L}{6} \|\boldsymbol{y} - \boldsymbol{x}\|^3.$$

**Gegenbauer Polynomials**

We briefly define Gegenbauer polynomials here whose details can be found in Appendix of Mei & Montanari (2022). First, we denote $\mathbb{S}^{d-1}(r) = \{\boldsymbol{x} \in \mathbb{R}^d : \|\boldsymbol{x}\| = r\}$ as the uniform spherical distribution with the radius $r$ on $d-1$ manifold. Let $\tau_d$ be the probability measure on $\mathbb{S}^{d-1}$ and and the inner product in functional space $L^2([-d, d], \mu_d)$ denoted as $\langle \cdot, \cdot \rangle_{L^2}$ and $\| \cdot \|_{L^2}$ :

$$\langle f, g \rangle_{L^2} \equiv \int_{\mathbb{S}^{d-1}(\sqrt{d})} f(\boldsymbol{x}) g(\boldsymbol{x}) \mu_d(\,\mathrm{d}\boldsymbol{x}).$$

For any function $\sigma \in L^2([-d, d], \tau_d)$, where $\tau_d$ is the distribution of $\langle \boldsymbol{x}, \boldsymbol{y} \rangle / \sqrt{d}$ ($\boldsymbol{x}, \boldsymbol{y} \sim \mathbb{S}^{d-1}(\sqrt{d})$), the orthogonal basis $\{Q_t^d\}$ forms the Gegenbauer polynomial of degree $t(t \ge 0)$, its spherical harmonics coefficients $\lambda_{d,t}(\sigma)$ can be expressed as:

$$\lambda_{d,t}(\sigma) = \int_{[-\sqrt{d}, \sqrt{d}]} \sigma(x) Q_t^{(d)}(\sqrt{d}x) \tau_d(x),$$

then the Gegenbauer generating function holds in $L^2\left([-\sqrt{d}, \sqrt{d}], \tau_d\right)$ sense

$$\sigma(x) = \sum_{k=0}^{\infty} \lambda_{d,t}(\sigma) N_{d,t} Q_t^{(d)}(\sqrt{d}x)$$

where $N_{d,t}$ is the normalized factor depending on the norm of input.

## B.1 ASSUMPTIONS

We discuss and list all assumptions we used in our theorems. The purposes are to offer clarity regarding the specific assumptions required for each theorem and ensure that the assumptions made in our theorems are well-founded and reasonable, reinforcing the validity and reliability of our results.

**OOD Generalization**

(Setting): Given a full sample space $\mathcal{Z}$, source and target distributions are two different measures over this whole sample domain $\mathcal{Z}$. The purpose is to study the robust algorithms in the OOD generalization setting.

(Assumptions): For any sample $\forall \boldsymbol{z} \in \mathcal{Z}$, the loss function is bounded $\ell_{\boldsymbol{\theta}}(\boldsymbol{z}) \in [0, M]$. This assumption generally follows the original paper Xu & Mannor (2012). While it is possible to relax this assumption and derive improved bounds, our primary objective is to formulate a framework for robust OOD generalization and establish a clear connection with the optimization properties of the model.

**Robustness and Sharpness**

(Setting): In order to give a fine-grained analysis, we follow the common choice where a two-layer ReLU Neural Network function class is widely analyzed in most literature, i.e. Neural Tangent Kernel, non-kernel (rich) regime Moroshko et al. (2020) and random feature models. Among them, we select the following random feature models as our function class:

$$f(\boldsymbol{w}, A, \boldsymbol{x}) \triangleq \frac{1}{\sqrt{d}} \sum_{i=1}^{m} w_i \sigma\left(\boldsymbol{x}, \boldsymbol{a}_i\right) : w_i \in \mathbb{R}, \boldsymbol{a}_i \sim \mathrm{Unif}(\mathbb{S}^{d-1}(\sqrt{d})), i \in [m]$$

where $m$ is the hidden size and $A = [\boldsymbol{a}_1, ..., \boldsymbol{a}_m]$ contains random vectors uniformly distributed on $n$-dim hypersphere whose surface is a $n-1$ manifold. $\sigma(\boldsymbol{a}^\top \boldsymbol{x}) = (\boldsymbol{a}^\top \boldsymbol{x})\mathbb{I}\{\boldsymbol{a}^\top \boldsymbol{x}\}$ denotes the ReLU activation function and $\mathbb{I}$ is the indicator function. $\boldsymbol{w} = [w_1, ..., w_m]^\top$ is the trainable parameter. We choose the common loss functions: (1) Homogeneity in regression; (2) (Binary) Cross-Entropy Loss; (3) Negative Log Likelihood (NLL) loss;

(Assumptions):

(i) Let $C_i, i \in [K]$ be any set from whole partitions $\cup_{i=1}^K C_i$, we assume $\forall \boldsymbol{z} \in C_i$, the loss function $\ell_{\hat{\boldsymbol{w}}, A}(\boldsymbol{z})$ satisfies $L$-Hessian Lipschitz for all $i \in [K]$ (See details in Definition B.2). Note that we only require this assumption to hold within each partition, instead of holding globally. In general, the smoothness and convexity condition is actually equivalent to locally convex which is a weak assumption for most function classes.

(ii) Consider an optimization problem in each partition $C_i, i \in [K]$. Let one of the training points $\boldsymbol{z}_i(A) \in S \cap C_i$ be the initial point and $\boldsymbol{z}_i^*(A) \in \mathcal{M}_i$ is the corresponding nearest local minima where $\mathcal{M}_i$ is the local minima set of partition $C_i$. For some $\rho_i(L) > 0$, we assume $\|\boldsymbol{z}_i(A) - \boldsymbol{z}_i^*(A)\| \leq \rho_i(L)/L$ holds a.s.. It ensures the hessian norm $\|H(\boldsymbol{z}_i(A))\|$ has the lower bound. Similar conditions and estimations can be found in Zhang et al. (2019a).

(iii) To simplify the computation of probability, we assume $\xi_i = \boldsymbol{a}_i^\top \boldsymbol{x}$ obeys a rotationally invariant distribution.

(iv) For Corollary 3.5, we make additional assumptions that loss function $\ell()$ satisfied a bounded condition where the second derivative $\forall \boldsymbol{x} \sim \mathrm{Unif}(\mathbb{S}^{d-1}(\sqrt{d})), |\partial^2 \ell(f(\hat{\boldsymbol{w}}, A, \boldsymbol{x}), y)/\partial f^2|$ with respect to its argument $f(\hat{\boldsymbol{w}}, A, \boldsymbol{x})$ should be bounded by $[\tilde{M}_1, \tilde{M}_2]$. Note, we consider the case where the data $\forall \boldsymbol{x} \sim \mathrm{Unif}(\mathbb{S}^{d-1}(\sqrt{d}))$ while $m = \mathrm{Poly}(d)$ is to ensure the positive definiteness of $\sum_{i=1}^{m} \boldsymbol{a}_i \boldsymbol{a}_i^\top \in \mathbb{R}^{d \times d}$ almost surely.

## C  PROOF TO DOMAIN SHIFT

**Lemma C.1.** *Let $\hat{\mathcal{D}}_T$ be the empirical distribution of size $n$ drawn from $\mathcal{D}_T$. The loss $\ell$ is upper bounded by M. With probability at least $1 - \delta$ (over the choice of the samples), for every $f_{\boldsymbol{\theta}}$ trained on S, we have*

$$\mathcal{L}_T(f_{\boldsymbol{\theta}}) \leq \mathcal{L}_S(f_{\boldsymbol{\theta}}) + M d_{(\epsilon, K)}(S, \hat{\mathcal{D}}_T) + \epsilon(S) + 2M \sqrt{\frac{2K \ln 2 + 2 \ln(1/\delta)}{n}} \tag{11}$$

*where*

$$\forall i \in [n], n_i(S) := \#(\boldsymbol{z} \in S \cap C_i), \quad d_{(\epsilon, K)}(S, \hat{\mathcal{D}}_T) := \sum_{i=1}^{K} \left| \frac{n_i(S)}{n} - \frac{n_i(\hat{\mathcal{D}}_T)}{n} \right| \tag{12}$$

*and $n_i(S), n_i(\hat{\mathcal{D}}_T)$ are the number of samples from $S$ and $\hat{\mathcal{D}}_T$ that fall into the set $C_i$, respectively.*

*Proof.* In the following generalization statement, we use $\ell(f_{\boldsymbol{\theta}}, \boldsymbol{z})$ to denote the error obtained with input $\boldsymbol{z}$ and hypothesis function $f_{\boldsymbol{\theta}}$ for better illustration. By definition we have,

$$\mathcal{L}_T(f_{\boldsymbol{\theta}}) - \mathcal{L}_S(f_{\boldsymbol{\theta}}) := \mathbb{E}_{\boldsymbol{z}' \sim \mathcal{D}_{\boldsymbol{\theta}}} \ell(f_{\boldsymbol{\theta}}, \boldsymbol{z}') - \mathbb{E}_{\boldsymbol{z} \sim \mathcal{D}_S} \ell(f_{\boldsymbol{\theta}}, \boldsymbol{z}). \tag{13}$$

Then we make the $K$ partitions for source distribution $\mathcal{D}_S$. Let $n_i$ be the size of collection set of points $\boldsymbol{x}$ fall into the partition $C_i$ where $n_i$ is the i.i.d.multinomial random variable with $(p_s(C_1), ..., p_s(C_K))$. We use parallel notation for target distribution $\mathcal{D}_T$ with $S_i' \sim (p_t(C_1), ..., p_t(C_K))$. Since

$$\mathrm{E}_{\boldsymbol{z} \sim \mathcal{D}_S} \ell(f_{\boldsymbol{\theta}}, \boldsymbol{z}) = \sum_{i=1}^{K} \mathbb{E}_{\boldsymbol{z} \sim \mathcal{D}_S} (\ell(f_{\boldsymbol{\theta}}, \boldsymbol{z}') | \boldsymbol{z} \in C_i) p_s(C_i)$$

$$\mathrm{E}_{\boldsymbol{z}' \sim \mathcal{D}_T} \ell(f_{\boldsymbol{\theta}}, \boldsymbol{z}') = \sum_{i=1}^{K} \mathbb{E}_{\boldsymbol{z}' \sim \mathcal{D}_T} (\ell(f_{\boldsymbol{\theta}}, \boldsymbol{z}') | \boldsymbol{z}' \in C_i) p_t(C_i) \tag{14}$$

and thus we have

$$\begin{aligned}
\mathcal{L}_T(f_{\boldsymbol{\theta}}) - \mathcal{L}_S(f_{\boldsymbol{\theta}}) &= \sum_{i=1}^{K} \mathbb{E}(\ell(f_{\boldsymbol{\theta}}, \boldsymbol{z}') | \boldsymbol{z}' \in C_i) p_t(C_i) - \mathbb{E}(\ell(f_{\boldsymbol{\theta}}, \boldsymbol{z}) | \boldsymbol{z} \in C_i) p_s(C_i) \\
&\quad \pm \mathbb{E}(\ell(f_{\boldsymbol{\theta}}, \boldsymbol{z}') | \boldsymbol{z}' \in C_i) p_s(C_i) \\
&= \sum_{i=1}^{K} \left( \mathbb{E}(\ell(f_{\boldsymbol{\theta}}, \boldsymbol{z}') | \boldsymbol{z}' \in C_i) \right) (p_t(C_i) - p_s(C_i)) \\
&\quad + \sum_{i=1}^{K} \left[ \mathbb{E}(\ell(f_{\boldsymbol{\theta}}, \boldsymbol{z}') | \boldsymbol{z}' \in C_i) - \mathbb{E}(\ell(f_{\boldsymbol{\theta}}, \boldsymbol{z}) | \boldsymbol{z} \in C_i) \right] p_s(C_i) \\
&\leq \sum_{i=1}^{K} \left( \mathbb{E}(\ell(f_{\boldsymbol{\theta}}, \boldsymbol{z}') | \boldsymbol{z}' \in C_i) \right) (p_t(C_i) - p_s(C_i)) + \epsilon(S, A).
\end{aligned} \tag{15}$$

If we sample empirical distribution $S, \hat{\mathcal{D}}_T$ of size $n$ each drawn from $\mathcal{D}_S$ and $\mathcal{D}_T$, respectively. $(n_1, ..., n_K)$ are the i.i.d. random variables belongs to $C_i$. We use the parallel notation $n_i'$ for target distribution.

$$d_{(\epsilon, K)}(S, \hat{\mathcal{D}}_T) := \sum_{i}^{K} \left| \frac{n_i(S)}{n} - \frac{n_i(\hat{\mathcal{D}}_T)}{n} \right|. \tag{16}$$

Further, we have

$$\begin{aligned}
\sum_{i=1}^{K} \left( \mathbb{E}(\ell(f_{\boldsymbol{\theta}}, \boldsymbol{z}') | \boldsymbol{z}' \in C_i) \right) &(p_t(C_i) - p_s(C_i)) - \sum_{i=1}^{K} \left( \mathbb{E}(\ell(f_{\boldsymbol{\theta}}, \boldsymbol{z}') | \boldsymbol{z}' \in C_i) \right) \left( \frac{n_i(\hat{\mathcal{D}}_T)}{n} - \frac{n_i(S)}{n} \right) \\
&= \sum_{i=1}^{K} \left( \mathbb{E}(\ell(f_{\boldsymbol{\theta}}, \boldsymbol{z}') | \boldsymbol{z}' \in C_i) \right) \left( p_t(C_i) - \frac{n_i(\hat{\mathcal{D}}_T)}{n} \right) - \sum_{i=1}^{K} \left( \mathbb{E}(\ell(f_{\boldsymbol{\theta}}, \boldsymbol{z}') | \boldsymbol{z}' \in C_i) \right) \left( p_s(C_i) - \frac{n_i(S)}{n} \right) \\
&\leq M \sum_{i=1}^{K} \left| p_t(C_i) - \frac{n_i(\hat{\mathcal{D}}_T)}{n} \right| + M \sum_{i=1}^{K} \left| p_s(C_i) - \frac{n_i(S)}{n} \right|.
\end{aligned} \tag{17}$$

With Breteganolle-Huber-Carol inequality we have

$$\sum_{i=1}^{K} \left| \frac{n_i(S)}{n} - p_s(C_i) \right| \leq \sqrt{\frac{2K \ln 2 + 2 \ln(1/\delta)}{n}}. \tag{18}$$

To integrate these two inequalities, we have

$$\sum_{i=1}^{K} (\mathbb{E}(\ell(f_{\boldsymbol{\theta}}, \boldsymbol{z}')|\boldsymbol{z}' \in C_i))(p_t(C_i) - p_s(C_i))$$

$$\leq \sum_{i=1}^{K} (\mathbb{E}(\ell(f_{\boldsymbol{\theta}}, \boldsymbol{z}')|\boldsymbol{z}' \in C_i)) \left( \frac{n_i(\hat{\mathcal{D}}_T)}{n} - \frac{n_i(S)}{n} \right) + 2M\sqrt{\frac{2K \ln 2 + 2\ln(1/\delta)}{n}}$$

$$\leq M d_{(\epsilon, K)}(S, \hat{\mathcal{D}}_T) + 2M\sqrt{\frac{2K \ln 2 + 2\ln(1/\delta)}{n}}. \tag{19}$$

In summary with probability $1 - \delta$ we have

$$\mathcal{L}_T(f_{\boldsymbol{\theta}}) \leq \mathcal{L}_S(f_{\boldsymbol{\theta}}) + M d_{(\epsilon, K)}(S, \hat{\mathcal{D}}_T) + \epsilon(S) + 2M\sqrt{\frac{2K \ln 2 + 2\ln(1/\delta)}{n}} \tag{20}$$

which completes the proof. $\qquad\square$

With the result of the domain (distribution) shift and the relationship between sharpness and robustness, we can move forward to the final OOD generalization error bound. First, we state the context of ID robustness bound in Xu & Mannor (2012) as follows.

**Lemma C.2** (Xu et al.Xu & Mannor (2012))**.** *Assume that for all $h \in \mathcal{H}$ and $z \in \mathcal{Z}$, the loss is upper bounded by $M$ i.e., $\ell(h, z) \leq M$. If the learning algorithm $\mathcal{A}$ is $(K, \epsilon(\cdot))$-robust, then for any $\delta > 0$, with probability at least $1 - \delta$ over an iid draw of $n$ samples $S = (z_i)_{i=1}^{n}$, it holds that:*

$$\mathbb{E}_z \left[ \ell(\mathcal{A}_S, z) \right] \leq \frac{1}{n} \sum_{i=1}^{n} \ell(\mathcal{A}_S, z_i) + \epsilon(S) + M\sqrt{\frac{2K \ln 2 + 2\ln(1/\delta)}{n}}$$

As the conclusive results, we briefly prove the following result by summarizing Lemma C.2 and Lemma C.1.

**Theorem C.3** (Restatement of Theorem 3.1)**.** *Let $\hat{\mathcal{D}}_T$ be the empirical distribution of size $n$ drawn from $\mathcal{D}_T$. The loss $\ell$ is upper bounded by $M$. With probability at least $1 - \delta$ (over the choice of the samples), for every $f_{\boldsymbol{\theta}}$ trained on $S$, we have*

$$\mathcal{L}_T(\boldsymbol{\theta}) \leq \widehat{\mathcal{L}}_S(\boldsymbol{\theta}) + M d_{(\epsilon, K)}(S, \hat{\mathcal{D}}_T) + 2\epsilon(S) + 3M\sqrt{\frac{2K \ln 2 + 2\ln(2/\delta)}{n}}. \tag{21}$$

*Proof.* Firstly, with Lemma C.1 and probability as least $1 - \frac{\delta}{2}$, we have

$$\mathcal{L}_T(f_{\boldsymbol{\theta}}) \leq \mathcal{L}_S(f_{\boldsymbol{\theta}}) + M d_{(\epsilon, K)}(\hat{\mathcal{D}}_S, \hat{\mathcal{D}}_T) + 2M\sqrt{\frac{2K \ln 2 + 2\ln(2/\delta)}{n}} + \epsilon(S)$$

Secondly, with Lemma C.2 (Xu & Mannor (2012) Theorem 3) and probability as least $1 - \frac{\delta}{2}$, we have

$$\left| \mathcal{L}_S(f_{\boldsymbol{\theta}}) - \hat{\mathcal{L}}_S(f_{\boldsymbol{\theta}}) \right| \leq \epsilon(S) + M\sqrt{\frac{2K \ln 2 + 2\ln(2/\delta)}{n}}$$

By taking the union bound, we conclude our final result that with probability at least $1 - \delta$

$$\mathcal{L}_T(f_{\boldsymbol{\theta}}) \leq \hat{\mathcal{L}}_S(f_{\boldsymbol{\theta}}) + 3M\sqrt{\frac{2K \ln 2 + 2\ln(2/\delta)}{n}} + 2\epsilon(S) + M d_{(\epsilon, K)}(S, \hat{\mathcal{D}}_T) \tag{22}$$

$\qquad\square$

Here $\epsilon(S)$ is the robustness constant that we can replace with any sharpness measure.

## C.1 Proof to Corollary 3.2

**Definition C.4.** $d_{\mathcal{F}\Delta\mathcal{F}}(\mathcal{D}_T; \mathcal{D}_S) := 2 \sup_{\mathcal{A}(f) \in \mathcal{A}_{\mathcal{F}\Delta\mathcal{F}}} |\Pr_{\mathcal{D}_S}(\mathcal{A}(f)) - \Pr_{\mathcal{D}_T}(\mathcal{A}(f))|$ and $\mathcal{F}\Delta\mathcal{F}$ is defined as:
$$\mathcal{F}\Delta\mathcal{F} := \{f(x) \oplus f'(x) : f, f' \in \mathcal{F}\}$$
where $\oplus$ is the XOR operator e.g. $\mathbb{I}(f'(x) \neq f(x))$.

**Lemma C.5** (Lemma 2 Zhao et al. (2018)). *If* $\mathrm{Pdim}(\mathcal{F}) = d'$, *then* $\mathrm{VCdim}(\mathcal{F}\Delta\mathcal{F}) \leq 2d'$ .

**Proposition C.6** (Zhao et al. (2018)). *Let* $\mathcal{F}$ *be a hypothesis class with pseudo dimension* $\mathrm{Pdim}(\mathcal{F}) = d'$. *If* $\widehat{\mathcal{D}}_S$ *is the empirical distributions generated with* $n$ *i.i.d.. samples from source domain, and* $\widehat{\mathcal{D}}_T$ *is the empirical distribution on the target domain generated from* $n$ *samples without labels, then with probability at least* $1 - \delta$, *for all* $f \in \mathcal{F}$, *we have:*

$$\mathcal{L}_T(f) \leq \widehat{\mathcal{L}}_S(f) + \mathcal{E}^* + \sqrt{\frac{2d' \log \frac{en}{d'}}{n}} + \sqrt{\frac{\log \frac{2}{\delta}}{2n}}$$
$$+ \underbrace{\frac{1}{2} d_{\mathcal{F}\Delta\mathcal{F}} \left(\widehat{\mathcal{D}}_T; \widehat{\mathcal{D}}_S\right) + 4\sqrt{\frac{2d' \ln(2n) + \ln \frac{4}{\delta}}{n}}}_{\text{(Empirical div Error)}} \tag{23}$$

*where* $\mathcal{E}^* = \widehat{\mathcal{L}}_S(f^*) + \widehat{\mathcal{L}}_T(f^*)$ *is the total error of best hypothesis* $f^*$ *over source and target domain.*

*Proof.* With Lemma 4 (Zhao et al., 2018), we have

$$\mathcal{L}_T(f) \leq \widehat{\mathcal{L}}_S(f) + \frac{1}{2} d_{\mathcal{F}\Delta\mathcal{F}} + \mathcal{E}^*$$

where

$$\mathcal{E}^* = \inf_{f' \in \mathcal{F}} \mathcal{L}_S(f') + \mathcal{L}_T(f').$$

Lemma 6 (Zhao et al., 2018), which is actually Lemma 1 in (Ben-David et al., 2010), shows the following results

$$d_{\mathcal{F}\Delta\mathcal{F}}(\mathcal{D}_T; \mathcal{D}_S) \leq d_{\mathcal{F}\Delta\mathcal{F}}\left(\widehat{\mathcal{D}}_T; \widehat{\mathcal{D}}_S\right) + 4\sqrt{\frac{\mathrm{VCdim}(\mathcal{F}\Delta\mathcal{F}) \ln(2n) + \ln \frac{2}{\delta}}{n}}.$$

As suggested in Zhao et al. (2018), $\mathrm{VCdim}(\mathcal{F}\Delta\mathcal{F})$ is at most $2d'$. Further, with Theorem 2 (Ben-David et al., 2010), we have at probability at least $1 - \frac{\delta}{2}$

$$\mathcal{L}_T(f) \leq \mathcal{L}_S(f) + \frac{1}{2} d_{\mathcal{F}\Delta\mathcal{F}}\left(\widehat{\mathcal{D}}_T; \widehat{\mathcal{D}}_S\right) + 4\sqrt{\frac{\mathrm{VCdim}(\mathcal{F}\Delta\mathcal{F}) \ln(2n) + \ln \frac{2}{\delta}}{n}} + \mathcal{E}^*$$
$$\leq \mathcal{L}_S(f) + \frac{1}{2} d_{\mathcal{F}\Delta\mathcal{F}}\left(\widehat{\mathcal{D}}_T; \widehat{\mathcal{D}}_S\right) + 4\sqrt{\frac{2d' \ln(2n) + \ln \frac{2}{\delta}}{n}} + \mathcal{E}^* \tag{24}$$

Using in-domain generalization error Lemma 11.6 (Mohri et al., 2018), with probability at least $1 - \frac{\delta}{2}$ the result is

$$\mathcal{L}_S(f) \leq \widehat{L}_S(f) + M\sqrt{\frac{2d' \log \frac{en}{d'}}{n}} + M\sqrt{\frac{\log \frac{1}{\delta}}{2n}}$$

Note in Zhao et al. (2018), the $M = 1$ for the normalized regression loss. Combine them all, we conclude the proof. $\square$

**Corollary C.7.** *If* $K \to \infty, M = 1$, *domain shift bound* $|\mathcal{L}_T(f_{\boldsymbol{\theta}}) - \mathcal{L}_S(f_{\boldsymbol{\theta}})|$ *will be reduced to (Empirical div Error) in Proposition C.6 where*

$$|\mathcal{L}_T(f_{\boldsymbol{\theta}}) - \mathcal{L}_S(f_{\boldsymbol{\theta}})| \leq \frac{1}{2} d_{\mathcal{F}\Delta\mathcal{F}}(\mathcal{D}_S; \mathcal{D}_T) \leq \textit{(Empirical div Error)} \tag{25}$$

*Proof.* According to Theorem C.3, we have

$$\mathcal{L}_T(f_{\boldsymbol{\theta}}) - \mathcal{L}_S(f_{\boldsymbol{\theta}}) = \sum_i^K \mathbb{E}(\ell(f_{\boldsymbol{\theta}}, \boldsymbol{z}') | \boldsymbol{z}' \in C_i) p_t(C_i) - \mathbb{E}(\ell(f_{\boldsymbol{\theta}}, \boldsymbol{z}) | \boldsymbol{z} \in C_i) p_s(C_i) \quad (26)$$

If $K \to \infty$, let's define a domain that $U := \bigcup_{i=1}^{\infty} C_i$. The equation (26) will be

$$
\begin{aligned}
\mathcal{L}_T(f_{\boldsymbol{\theta}}) - \mathcal{L}_S(f_{\boldsymbol{\theta}}) &= \int_{\boldsymbol{z}' \in U} \ell(f_{\boldsymbol{\theta}}, \boldsymbol{z}') p_t(\boldsymbol{z}') d\boldsymbol{z} - \int_{\boldsymbol{z} \in U} \ell(f_{\boldsymbol{\theta}}, \boldsymbol{z}) p_s(\boldsymbol{z}) d\boldsymbol{z} \\
&= \int_{\boldsymbol{z}' \in \mathcal{D}_T} \ell(f_{\boldsymbol{\theta}}, \boldsymbol{z}') p_t(\boldsymbol{z}') d\boldsymbol{z} - \int_{\boldsymbol{z} \in \mathcal{D}_S} \ell(f_{\boldsymbol{\theta}}, \boldsymbol{z}) p_s(\boldsymbol{z}) d\boldsymbol{z} \\
&= \mathbb{E}_{\boldsymbol{z}' \sim \mathcal{D}_T} \ell(f_{\boldsymbol{\theta}}, \boldsymbol{z}') - \mathbb{E}_{\boldsymbol{z} \sim \mathcal{D}_S} \ell(f_{\boldsymbol{\theta}}, \boldsymbol{z}).
\end{aligned}
\quad (27)
$$

In this case, we have,

$$
\begin{aligned}
&|\mathcal{L}_T(f_{\boldsymbol{\theta}}) - \mathcal{L}_S(f_{\boldsymbol{\theta}})| \\
&= |\mathbb{E}_{\boldsymbol{z}' \sim \mathcal{D}_T} \ell(f_{\boldsymbol{\theta}}, \boldsymbol{z}') - \mathbb{E}_{\boldsymbol{z} \sim \mathcal{D}_S} \ell(f_{\boldsymbol{\theta}}, \boldsymbol{z})| \\
&\leq \int_0^\infty |\mathrm{Pr}_{\mathcal{D}_T} \left( \ell(f(\boldsymbol{\theta}, \boldsymbol{x}'), y') > t \right) dt - \mathrm{Pr}_{\mathcal{D}_S} \left( \ell(f(\boldsymbol{\theta}, \boldsymbol{x}), y) > t \right) dt| \\
&= \int_0^1 |\mathrm{Pr}_{\mathcal{D}_T} \left( \ell(f(\boldsymbol{\theta}, \boldsymbol{x}'), y') > t \right) - \mathrm{Pr}_{\mathcal{D}_S} \left( \ell(f(\boldsymbol{\theta}, \boldsymbol{x}), y) > t \right)| dt \quad (M = 1) \\
&\leq \sup_{t \in [0,1]} \sup_{f(\boldsymbol{\theta}, \cdot) \in \mathcal{F}} |\mathrm{Pr}_{\mathcal{D}_T} \left( \ell(f(\boldsymbol{\theta}, \boldsymbol{x}'), y') > t \right) - \mathrm{Pr}_{\mathcal{D}_S} \left( \ell(f(\boldsymbol{\theta}, \boldsymbol{x}), y) > t \right)| \\
&\leq \sup_{\mathcal{A}(f) \in \mathcal{A}_{\mathcal{F} \Delta \mathcal{F}}} |\mathrm{Pr}_{\mathcal{D}_T}(\mathcal{A}(f)) - \mathrm{Pr}_{\mathcal{D}_S}(\mathcal{A}(f))| \\
&= \frac{1}{2} d_{\mathcal{F} \Delta \mathcal{F}}(\mathcal{D}_S; \mathcal{D}_T) \leq (\text{Empirical div error})
\end{aligned}
\quad (28)
$$

where $\mathcal{A}_{\mathcal{F} \Delta \mathcal{F}}$ represents a learning algorithm under the hypothesis $\mathcal{F} \Delta \mathcal{F} = \{f(x) \oplus f'(x) : f, f' \in \mathcal{F}\}$, which completes the proof. $\qquad \square$

## D  SHARPNESS AND ROBUSTNESS

**Lemma D.1** (positive definiteness of Hessian). *Let $\hat{w}_{\min}$ be the minimum value of $|\hat{\boldsymbol{w}}|$ and $\boldsymbol{x}^* = \arg\min_{\boldsymbol{x} \in \mathrm{Unif}(\mathbb{S}^{d-1}(\sqrt{d}))} \ell(f(\hat{\boldsymbol{w}}, A, \boldsymbol{x}), \boldsymbol{y})$. For any $A = (\boldsymbol{a}_1, ..., \boldsymbol{a}_m), \boldsymbol{a}_i \sim \mathrm{Unif}(\mathbb{S}^{d-1}(\sqrt{d})) \forall i \in [m]$ denote $\tilde{\sigma}(d, m) = \lambda_{\min}(\sum_{i=1}^m \boldsymbol{a}_i \boldsymbol{a}_i^\top G_{ii}) > 0$ be the minimum eigenvalue, where $G_{ij} = \sum_{t=0}^\infty \lambda_{d,t}^2(\sigma) N_{d,t}^2 Q_t^{(d)}(\langle \boldsymbol{a}_i, \boldsymbol{a}_j \rangle / \sqrt{d})$ is the polynomial product constant. If $m = Poly(d)$, the hessian $H(\boldsymbol{x}^*)$ can be lower bound by*

$$\mathbb{E}_{\boldsymbol{x}^* \sim \mathrm{Unif}(\mathbb{S}^{d-1}(\sqrt{d}))} H(\boldsymbol{x}^*) \succeq \frac{\hat{w}_{\min}^2 \tilde{\sigma}(d, m) \tilde{M}_1}{d} I_d. \quad (29)$$

*Proof.* As suggested in Lemma D.6 of (Zhong et al., 2017), we have a similar result to bound the local positive definiteness of Hessian. By previous definition, the Hessian w.r.t. $\boldsymbol{x}$ has a following partial order

$$
\begin{aligned}
H(\boldsymbol{x}^*) &= \frac{D_f^2(\boldsymbol{x}^*, y^*)}{d} \left( \sum_{i=1}^m \sum_{j=1}^m \hat{w}_i \hat{w}_j \boldsymbol{a}_i \boldsymbol{a}_j^\top \sigma'(\boldsymbol{a}_i^\top \boldsymbol{x}^*) \sigma'(\boldsymbol{a}_j^\top \boldsymbol{x}^*) \right) \\
&\succeq \frac{\tilde{M}_1}{d} \left( \sum_{i=1}^m \sum_{j=1}^m \hat{w}_i \hat{w}_j \boldsymbol{a}_i \boldsymbol{a}_j^\top \sigma'(\boldsymbol{a}_i^\top \boldsymbol{x}^*) \sigma'(\boldsymbol{a}_j^\top \boldsymbol{x}^*) \right) \\
&\succeq \frac{\hat{w}_{\min}^2 \tilde{M}_1}{d} \left( \sum_{i=1}^m \sum_{j=1}^m \hat{w}_i \hat{w}_j \boldsymbol{a}_i \boldsymbol{a}_j^\top \sigma'(\boldsymbol{a}_i^\top \boldsymbol{x}^*) \sigma'(\boldsymbol{a}_j^\top \boldsymbol{x}^*) \right)
\end{aligned}
\quad (30)
$$

For the ReLU activation function, we further have

$$\sigma(\boldsymbol{a}^\top \boldsymbol{x}^*) \geq \sigma'(\boldsymbol{a}^\top \boldsymbol{x}^*) \tag{31}$$

We extend the $\sigma \in L^2([-\sqrt{d}, \sqrt{d}], \tau_d)$ (where $\tau_d$ is the distribution of $\langle \boldsymbol{x}_1, \boldsymbol{x}_2 \rangle / \sqrt{d}$) by Gegenbauer polynomials that

$$\sigma(x) = \sum_{t=0}^{\infty} \lambda_{d,t}(\sigma) N_{d,t} Q_t^{(d)}(\sqrt{d}x). \tag{32}$$

Let $A = (\boldsymbol{a}_1, ..., \boldsymbol{a}_m) \in \mathbb{R}^{m \times d}$. We assume $\forall \boldsymbol{x} \in \mathrm{Unif}(\mathbb{S}^{d-1}(\sqrt{d}))$. Lemma C.7 in (Mei & Montanari, 2022), suggests that

$$\boldsymbol{U} = \left( \mathbb{E}_{\boldsymbol{x}^* \sim \mathrm{Unif}(\mathbb{S}^{d-1}(\sqrt{d}))}[\sigma(\langle \boldsymbol{a}_i, \boldsymbol{x}^* \rangle / \sqrt{d})\sigma(\langle \boldsymbol{a}_i, \boldsymbol{x}^* \rangle / \sqrt{d})] \right)_{i,j \in [m]} \in \mathbb{R}^{m \times m} \tag{33}$$

which shows matrix $U$ is a positive definite matrix. Similarly, taking the expectation over $\boldsymbol{x}^*$, terms in RHS of (30) bracket can be rewritten as

$$\mathbb{E}_{\boldsymbol{x}^* \sim \mathrm{Unif}(\mathbb{S}^{d-1}(\sqrt{d}))} \left( \sum_{i=1}^m \sum_{j=1}^m \boldsymbol{a}_i \boldsymbol{a}_j^\top \sigma'(\boldsymbol{a}_i^\top \boldsymbol{x}^*) \sigma'(\boldsymbol{a}_j^\top \boldsymbol{x}^*) \right)$$
$$\succeq \sum_{i=1}^m \sum_{j=1}^m \boldsymbol{a}_i \boldsymbol{a}_j^\top \mathbb{E}_{\boldsymbol{x}^*}[\sigma(\boldsymbol{a}_i^\top \boldsymbol{x}^* / \sqrt{d})\sigma(\boldsymbol{a}_j^\top \boldsymbol{x}^* / \sqrt{d})] \tag{34}$$

Besides, we have the following property of Gegenbauer polynomials,

1. For $\boldsymbol{x}, \boldsymbol{y} \in \mathbb{S}^{d-1}(\sqrt{d})$

$$\left\langle Q_j^{(d)}(\langle \boldsymbol{x}, \cdot \rangle), Q_k^{(d)}(\langle \boldsymbol{y}, \cdot \rangle) \right\rangle_{L^2(\mathbb{S}^{d-1}(\sqrt{d}), \gamma_d)} = \frac{1}{N_{d,k}} \delta_{jk} Q_k^{(d)}(\langle \boldsymbol{x}, \boldsymbol{y} \rangle).$$

2. For $\boldsymbol{x}, \boldsymbol{y} \in \mathbb{S}^{d-1}(\sqrt{d})$

$$Q_k^{(d)}(\langle \boldsymbol{x}, \boldsymbol{y} \rangle) = \frac{1}{N_{d,k}} \sum_{i=1}^{N_{d,k}} Y_{ki}^{(d)}(\boldsymbol{x}) Y_{ki}^{(d)}(\boldsymbol{y}).$$

where spherical harmonics $\{Y_{lj}^{(d)}\}_{1 \leq j \leq N_{d,l}}$ forms an orthonormal basis which gives the following results

$$\mathbb{E}_{\boldsymbol{x}^*}[\sigma(\boldsymbol{a}_i^\top \boldsymbol{x}^* / \sqrt{d})\sigma(\boldsymbol{a}_j^\top \boldsymbol{x}^* / \sqrt{d})] = \sum_{t=0}^{\infty} \lambda_{d,t}^2(\sigma) N_{d,t}^2 \mathbb{E}_{\boldsymbol{x}^*} Q_t^{(d)}(\langle \boldsymbol{a}_i, \boldsymbol{x}^* \rangle / \sqrt{d}) Q_t^{(d)}(\langle \boldsymbol{a}_j, \boldsymbol{x}^* \rangle / \sqrt{d})$$
$$= \sum_{t=0}^{\infty} \lambda_{d,t}^2(\sigma) N_{d,t}^2 Q_t^{(d)}(\langle \boldsymbol{a}_i, \boldsymbol{a}_j \rangle / \sqrt{d}) = G_{ij} < \infty. \tag{35}$$

Hence, we have

$$\sum_{i=1}^m \sum_{j=1}^m \boldsymbol{a}_i \boldsymbol{a}_j^\top \mathbb{E}_{\boldsymbol{x}^*}[\sigma(\boldsymbol{a}_i^\top \boldsymbol{x}^* / \sqrt{d})\sigma(\boldsymbol{a}_j^\top \boldsymbol{x}^* / \sqrt{d})] = \sum_{i=1}^m \boldsymbol{a}_i \boldsymbol{a}_i^\top G_{ii} + \mathcal{O}(1/d)\mathrm{Var}(\boldsymbol{a}) \tag{36}$$

Since $m = Poly(d)$, and $\{\boldsymbol{a}\}_{i \in [m]}$ are i.i.d, then rank $\left( \sum_{i=1}^m \boldsymbol{a}_i \boldsymbol{a}_i^\top G_{ii} \right) = $ rank $\left( AA^\top \right) = d$. Let $\tilde{\sigma}(d, m) = \mathbb{E}_{\boldsymbol{a}} \lambda_{\min}(\sum_{i=1}^m \boldsymbol{a}_i \boldsymbol{a}_i^\top G_{ii}) > 0$ we have

$$\mathbb{E}_{\boldsymbol{x}^*} H(\boldsymbol{x}^*) \succeq \frac{\hat{w}_{\min}^2 \tilde{\sigma}(d, m) \tilde{M}_1}{d} I_d \tag{37}$$

$\square$

**Lemma D.2.** *Let $\bigcup_{k=1}^K C_k$ be the whole domain, the notion of $(\epsilon, K)$-robustness is described by*

$$\epsilon(S, A) \triangleq \max_{C_i \subset \bigcup_{k=1}^K C_k} \sup_{\boldsymbol{z}, \boldsymbol{z}_i' \in C_i, \boldsymbol{z} \in S} |\ell_{\hat{\boldsymbol{w}}, A}(\boldsymbol{z}) - \ell_{\hat{\boldsymbol{w}}, A}(\boldsymbol{z}_i')|.$$

*Define $\mathcal{M}_i$ be the set of global minima in $C_i$, where*

$$\mathcal{M}_i \triangleq \{\boldsymbol{z}(A) | \boldsymbol{z}(A) = \min_{\boldsymbol{z} \in C_i} \ell_{\hat{\boldsymbol{w}}, A}(\boldsymbol{z})\}$$

*suppose for some maximum training loss point*

$$\boldsymbol{z}_i(A) \in \left\{ \max_{\boldsymbol{z} \in C_i \cap S} \ell_{\hat{\boldsymbol{w}}, A}(\boldsymbol{z}) - \ell_{\hat{\boldsymbol{w}}, A}(\boldsymbol{z}_i^*(A)) \right\}$$

*there $\exists \boldsymbol{z}_i^*(A)$ where*

$$\boldsymbol{z}_i^*(A) \triangleq \arg \min_{\boldsymbol{z} \in \mathcal{M}_i} \|\boldsymbol{z} - \boldsymbol{z}_i(A)\|$$

*such that $\|\boldsymbol{z}_i(A) - \boldsymbol{z}^*(A)\| \leq \frac{\rho_i(L)}{L}$ almost surely hold for any $A \in \mathrm{Unif}(\mathbb{S}^{d-1}(\sqrt{d}))$ and for any $A$, $\ell_{\hat{\boldsymbol{w}}, A}(\boldsymbol{z})$ is L-Hessian Lipschitz continuous. Then the $\epsilon(S, A)$ can be bounded by*

$$\epsilon(S, A) \leq \max_{i \in [K]} \frac{\rho_i(L)^2}{2L^2} \left( \left\| \nabla^2 \ell_{\hat{\boldsymbol{w}}, A}(\boldsymbol{z}_i(A)) \right\| + \frac{4\rho_i(L)}{3} \right)$$

*Proof.* Let $\boldsymbol{z} \in S$ be a collection of $(\boldsymbol{x}, y)$ from the training set $S$ and $\boldsymbol{z}_i'$ denote any collection from the set $C_i$. We define local minima set $\mathcal{M}_i$ (which is the global minima set of $C_i$). Assume that for some maximum point $\boldsymbol{z}_i(A) \in \max_{\boldsymbol{z} \in C_i \cap S} \ell_{\hat{\boldsymbol{w}}, A}(\boldsymbol{z})$, there exists a $\boldsymbol{z}_i^*(A) \in \mathcal{M}_i$ almost surely for all $A \sim \mathrm{Unif}(\mathbb{S}^{d-1}(\sqrt{d}))$ such that

$$\boldsymbol{z}_i^*(A) = \arg \min_{\boldsymbol{z} \in \mathcal{M}_i} f := \{\boldsymbol{z} \in \mathcal{M}_i : \|\boldsymbol{z}_i(A) - \boldsymbol{z}\|\} \quad \text{s.t.} \quad \|\boldsymbol{z}_i(A) - \boldsymbol{z}\| \leq \frac{\rho_i(L)}{L} \quad (38)$$

By definition, $\epsilon(S, A)$ can be rewritten as

$$\begin{aligned}
\epsilon(S, A) &= \max_{i \in [K]} \sup_{\boldsymbol{z}, \boldsymbol{z}_i' \in C_i, \boldsymbol{z} \in S} |\ell_{\hat{\boldsymbol{w}}, A}(\boldsymbol{z}) - \ell_{\hat{\boldsymbol{w}}, A}(\boldsymbol{z}_i')| \\
&= \max_{i \in [K]} \sup_{\boldsymbol{z} \in C_i \cap S, \boldsymbol{z}^* \in \mathcal{M}_i} \ell_{\hat{\boldsymbol{w}}, A}(\boldsymbol{z}) - \ell_{\hat{\boldsymbol{w}}, A}(\boldsymbol{z}^*) \\
&= \max_{i \in [K]} \ell_{\hat{\boldsymbol{w}}, A}(\boldsymbol{z}_i(A)) - \ell_{\hat{\boldsymbol{w}}, A}(\boldsymbol{z}_i^*(A)).
\end{aligned} \quad (39)$$

According to Lemma B.3, we have

$$\begin{aligned}
\epsilon(S, A) &= \max_{i \in [K]} \ell_{\hat{\boldsymbol{w}}, A}(\boldsymbol{z}_i(A)) - \ell_{\hat{\boldsymbol{w}}, A}(\boldsymbol{z}_i^*(A)) \\
&\overset{(i)}{\leq} \max_{i \in [K]} \langle \nabla \ell_{\hat{\boldsymbol{w}}, A}(\boldsymbol{z}_i^*(A)), \boldsymbol{z}_i(A) - \boldsymbol{z}_i^*(A) \rangle \\
&\quad + \frac{1}{2} \langle \nabla^2 \ell_{\hat{\boldsymbol{w}}, A}(\boldsymbol{z}_i^*(A))(\boldsymbol{z}_i(A) - \boldsymbol{z}_i^*(A)), \boldsymbol{z}_i(A) - \boldsymbol{z}_i^*(A) \rangle + \frac{L}{6} \|\boldsymbol{z}_i(A) - \boldsymbol{z}^*\|^3 \\
&= \max_{i \in [K]} \frac{1}{2} \langle \nabla^2 \ell_{\hat{\boldsymbol{w}}, A}(\boldsymbol{z}_i^*(A))(\boldsymbol{z}_i(A) - \boldsymbol{z}_i^*(A)), \boldsymbol{z}_i(A) - \boldsymbol{z}_i^*(A) \rangle + \frac{L}{6} \|\boldsymbol{z}_i(A) - \boldsymbol{z}_i^*(A)\|^3 \\
&\leq \max_{i \in [K]} \frac{1}{2} \left\| \nabla^2 \ell_{\hat{\boldsymbol{w}}, A}(\boldsymbol{z}_i^*(A)) \right\| \|\boldsymbol{z}_i(A) - \boldsymbol{z}_i^*(A)\|^2 \\
&\quad + \frac{L}{6} \|\boldsymbol{z}_i(A) - \boldsymbol{z}_i^*(A)\|^3 \quad (\text{Cauchy-Schwarz})
\end{aligned}$$

$$(40)$$

where $(i)$ support by the fact $\nabla \ell_{\hat{\boldsymbol{w}}, A}(\boldsymbol{z}^*) = 0$. With Lipschitz continuous Hessian we have

$$\|\nabla^2 \ell_{\hat{\boldsymbol{w}}, A}(\boldsymbol{z}_i^*(A))\| \leq L \|\boldsymbol{z}_i(A) - \boldsymbol{z}_i^*(A)\| + \|\nabla^2 \ell_{\hat{\boldsymbol{w}}, A}(\boldsymbol{z}_i(A))\|. \quad (41)$$

Overall, we have

$$
\begin{aligned}
\epsilon(S, A) &\leq \max_{i \in [K]} \frac{1}{2} \left\| \nabla^2 \ell_{\hat{\boldsymbol{w}}, A}(\boldsymbol{z}_i^*(A)) \right\| \|\boldsymbol{z}_i(A) - \boldsymbol{z}_i^*(A)\|^2 + \frac{L}{6} \|\boldsymbol{z}_i(A) - \boldsymbol{z}_i^*(A)\|^3 \\
&\leq \max_{i \in [K]} \frac{1}{2} \left( \|\nabla^2 \ell_{\hat{\boldsymbol{w}}, A}(\boldsymbol{z}_i(A))\| + L\|\boldsymbol{z}_i(A) - \boldsymbol{z}_i^*(A)\| \right) \|\boldsymbol{z}_i(A) - \boldsymbol{z}_i^*(A)\|^2 \\
&\quad + \frac{L}{6} \|\boldsymbol{z}_i(A) - \boldsymbol{z}_i^*(A)\|^3 \\
&\leq \max_{i \in [K]} \frac{1}{2} \left( \|\nabla^2 \ell_{\hat{\boldsymbol{w}}, A}(\boldsymbol{z}_i(A))\| + \rho_i(L) \right) \frac{\rho_i(L)^2}{L^2} + \frac{\rho_i(L)^3}{6L^2} \\
&= \max_{i \in [K]} \frac{\rho_i(L)^2}{2L^2} \|\nabla^2 \ell_{\hat{\boldsymbol{w}}, A}(\boldsymbol{z}_i(A))\| + \frac{2\rho_i(L)^3}{3L^2}
\end{aligned}
\tag{42}
$$

which completes the proof. $\qquad\square$

**Lemma D.3** (Lemma 2.1 Bourin et al. (2013))**.** *For every matrix in $\mathbb{M}_{n+m}^+$ partitioned into blocks, we have a decomposition*

$$
\begin{bmatrix} A & X \\ X^* & B \end{bmatrix} = U \begin{bmatrix} A & 0 \\ 0 & 0 \end{bmatrix} U^* + V \begin{bmatrix} 0 & 0 \\ 0 & B \end{bmatrix} V^*
$$

*for some unitaries $U, V \in \mathbb{M}_{n+m}$.*

**Lemma D.4.** *Then, given an arbitrary partitioned positive semi-definite matrix,*

$$
\left\| \begin{bmatrix} A & X \\ X^* & B \end{bmatrix} \right\|_s \leq \|A\|_s + \|B\|_s
$$

*for all symmetric norms.*

*Proof.* In lemma D.3 we have

$$
\begin{bmatrix} A & X \\ X^* & B \end{bmatrix} = U \begin{bmatrix} A & 0 \\ 0 & 0 \end{bmatrix} U^* + V \begin{bmatrix} 0 & 0 \\ 0 & B \end{bmatrix} V^*
$$

for some unitaries $U, V \in \mathbb{M}_{n+m}$. The result then follows from the simple fact that symmetric norms are non-decreasing functions of the singular values where $f = \| \cdot \|_s : \mathbb{M} \mapsto \mathbb{R}$, we have

$$
f\left( \begin{bmatrix} A & X \\ X^* & B \end{bmatrix} \right) \leq f\left( U \begin{bmatrix} A & 0 \\ 0 & 0 \end{bmatrix} U^* \right) + f\left( V \begin{bmatrix} 0 & 0 \\ 0 & B \end{bmatrix} V^* \right)
$$

$\qquad\square$

**Lemma D.5.** *For $\boldsymbol{a} \sim \mathrm{Unif}(\mathbb{S}^{d-1}(\sqrt{d}))$ and $\boldsymbol{x}$ are some vector $\in \mathbb{R}^d$ with norm $\|\boldsymbol{x}\| \equiv \sqrt{R(d)} \geq d$, we have*

$$
\mathbb{P}(\langle \boldsymbol{x}, \boldsymbol{a} \rangle^2 \geq \|\boldsymbol{a}\|^2) \geq \min \left\{ \frac{2 \arccos\left( \frac{1}{\sqrt{R(d)}} \right)}{\pi}, \left| 1 - \frac{\sqrt{2d-4}}{\sqrt{\pi R(d)}} \exp\left( \frac{1}{4d-9} \right) \right| \right\}
\tag{43}
$$

*Proof.* We can replace the unit vector of $\boldsymbol{a}$ with $\boldsymbol{e}$ by

$$
\mathbb{P}(\langle \boldsymbol{x}, \boldsymbol{a} \rangle^2 \geq \|\boldsymbol{a}\|^2) = \mathbb{P}(\langle \boldsymbol{x}, \boldsymbol{e} \rangle^2 \geq 1)
\tag{44}
$$

Similarly, we can replace $\boldsymbol{x}$ by unit vector $\boldsymbol{s}$ such that

$$
\mathbb{P}(\langle \boldsymbol{x}, \boldsymbol{e} \rangle^2 \geq 1) = \mathbb{P}\left( \langle \boldsymbol{s}, \boldsymbol{e} \rangle^2 \geq \frac{1}{R(d)} \right)
\tag{45}
$$

Solving $\langle \boldsymbol{s}, \boldsymbol{e} \rangle^2 = \frac{1}{R(d)}$, we get

$$
\langle \boldsymbol{s}, \boldsymbol{e} \rangle^2 = \cos^2 \phi = \frac{1}{R(d)} \Rightarrow \phi = \arccos \pm \frac{1}{\sqrt{R(d)}}
\tag{46}
$$

In this case, the probability will converge to $1$ as $R(d)$ increases. As is known to us, surface area of $\mathbb{S}^{d-1}$ equals

$$A_d = r^{d-1}\frac{2\pi^{d/2}}{\Gamma\left(\frac{d}{2}\right)} \tag{47}$$

An area $C_d$ of the spherical cap equals

$$A_d^{\text{cap}}(r) = \int_0^\phi A_{d-1}(r\sin\theta)r\mathrm{d}\theta = \frac{2\pi^{(d-1)/2}}{\Gamma\left(\frac{d-1}{2}\right)}r^{d-1}\int_0^\phi \sin^{d-2}\theta\mathrm{d}\theta. \tag{48}$$

where $\Gamma\left(n-\frac{1}{2}\right) = \frac{(2(n-1))!}{2^{2(n-1)}(n-1)!}\sqrt{\pi}$.

**1)** When $d = 1$, almost surely we have

$$\mathbb{P}((x\cdot e)^2 \geq e^2) = \mathbb{P}(x^2 \geq 1) = 1. \tag{49}$$

**2)** When $d = 2$, we have $\boldsymbol{a} \sim S^2$ where $S^2$ is a circle $r = 1$ and the probability is the angle between $\boldsymbol{s}, \boldsymbol{e}$ how much the vectors span within the circle where

$$\mathbb{P}\left(\langle \boldsymbol{s}, \boldsymbol{e}\rangle^2 \geq \frac{1}{R(d)}\right) = \frac{2\int_0^\phi r d\theta}{\pi} = \frac{2\phi}{\pi}. \tag{50}$$

**3)** When $d \geq 3$, the probability equals

$$\mathbb{P}\left(|\langle \boldsymbol{s}, \boldsymbol{e}\rangle| \geq \frac{1}{\sqrt{R(d)}}\right) = \frac{A_d^{\text{cap}}(r)}{\frac{1}{2}A_d} = 1 - \frac{\tilde{A}_d^{\text{cap}}(r)}{\frac{1}{2}A_d} \tag{51}$$

where $\tilde{A}_d^{\text{cap}}(r)$ is the remaining area of cutting the hyperspherical caps in half of the sphere,

$$\begin{aligned}
\tilde{A}_d^{\text{cap}}(r) &= \frac{2\pi^{(d-1)/2}}{\Gamma\left(\frac{d-1}{2}\right)}\int_\phi^{\frac{\pi}{2}}\sin^{d-2}\theta\mathrm{d}\theta \\
&\leq \frac{2\pi^{(d-1)/2}}{\Gamma\left(\frac{d-1}{2}\right)}\int_\phi^{\frac{\pi}{2}}\sin\theta\mathrm{d}\theta \\
&\leq \frac{2\pi^{(d-1)/2}}{\Gamma\left(\frac{d-1}{2}\right)}\left(-\cos\frac{\pi}{2}+\cos\phi\right) \\
&= \frac{2\pi^{(d-1)/2}}{\Gamma\left(\frac{d-1}{2}\right)}\cos\phi.
\end{aligned} \tag{52}$$

**3-a)** If $d$ is even then $\Gamma\left(\frac{d}{2}\right) = \left(\frac{d}{2}-1\right)!$, so

$$\frac{\Gamma\left(\frac{d-1}{2}\right)}{\Gamma\left(\frac{d}{2}\right)} = \frac{(d-2)!\sqrt{\pi}}{2^{d-2}\left(\frac{d}{2}-1\right)!^2} = \frac{\sqrt{\pi}}{2^{d-2}}\begin{pmatrix} d-2 \\ \frac{d-2}{2}\end{pmatrix}. \tag{53}$$

Robbins' bounds (Robbins, 1955) imply that for any positive integer $d$

$$\frac{4^d}{\sqrt{\pi d}}\exp\left(-\frac{1}{8d-1}\right) < \begin{pmatrix} 2d \\ d \end{pmatrix} < \frac{4^d}{\sqrt{\pi d}}\exp\left(-\frac{1}{8d+1}\right). \tag{54}$$

So we have

$$\begin{aligned}
\frac{2A_d^{\text{cap}}(r)}{A_d} &\leq \frac{2\Gamma\left(\frac{d}{2}\right)}{\sqrt{\pi}\Gamma\left(\frac{d-1}{2}\right)} = \frac{2^{d-1}}{\pi}\begin{pmatrix} d-2 \\ \frac{d-2}{2}\end{pmatrix}^{-1}\cos\phi \\
&< \frac{2^{d-1}}{\pi}\frac{\sqrt{\pi(d-2)/2}}{2^{d-2}\exp\left(-\frac{1}{4d-9}\right)}\cos\phi \\
&< \frac{\sqrt{2d-4}}{\sqrt{\pi}\exp\left(-\frac{1}{4d-9}\right)}\cos\phi.
\end{aligned} \tag{55}$$

So the probability will be

$$\mathbb{P}\left(\langle s, e \rangle^2 \geq \frac{1}{R(d)}\right) > 1 - \frac{\sqrt{2d-4}}{\sqrt{\pi}} \exp\left(\frac{1}{4d-9}\right) \cos\phi. \tag{56}$$

Suppose $R(d) = kd, k > 1$, we have

$$\lim_{d \to \infty} \frac{\sqrt{2d-4}}{\sqrt{\pi kd} \exp\left(-\frac{1}{4d-9}\right)} = \sqrt{\frac{2}{k\pi}}. \tag{57}$$

**3-b)** Similarly, if $d$ is odd, then $\Gamma\left(\frac{d-1}{2}\right) = \left(\frac{d-3}{2}\right)!$, so

$$\frac{\Gamma\left(\frac{d-1}{2}\right)}{\Gamma\left(\frac{d}{2}\right)} = \frac{2^{d-2}\left(\frac{d-3}{2}\right)!^2}{(d-2)!\sqrt{\pi}} = \frac{2^{d-2}}{(d-2)\sqrt{\pi}}\left(\frac{d-3}{\frac{d-3}{2}}\right)^{-1}. \tag{58}$$

If $d = 3$ then

$$\mathbb{P}\left(\langle s, e \rangle^2 \geq \frac{1}{R(d)}\right) \geq 1 - \frac{2\sqrt{\pi}}{2\sqrt{\pi}}\cos\phi = 1 - \frac{1}{\sqrt{R(d)}}. \tag{59}$$

If $d > 3$ then Robbins' bounds imply that

$$\frac{\sqrt{\pi}\Gamma\left(\frac{d-1}{2}\right)}{2\Gamma\left(\frac{d}{2}\right)} = \frac{2^{d-4}}{d-2}\left(\frac{d-3}{\frac{d-3}{2}}\right)^{-1} > \frac{\sqrt{\pi(d-3)/2}}{d-2}\exp\left(\frac{1}{4d-11}\right). \tag{60}$$

Thus, the probability will be at least

$$\mathbb{P}\left(\langle s, e \rangle^2 \geq \frac{1}{R(d)}\right) > 1 - \frac{d-2}{\sqrt{\pi(d-3)}}\exp\left(-\frac{1}{4d-11}\right)\cos\phi. \tag{61}$$

To simplify the result, we compare the minimum probability that for $\forall d \geq 3$

$$\begin{aligned}
\mathbb{P}\left(\langle s, e \rangle^2 \geq \frac{1}{R(d)}\right) &> 1 - \frac{d-2}{\sqrt{\pi(d-3)}}\exp\left(-\frac{1}{4d-11}\right)\cos\phi \\
&= 1 - \frac{d-2}{\sqrt{\pi(d-3)}}\exp\left(\frac{1}{4d-11}\right)\cos\phi \\
&= 1 - \frac{d-2}{\sqrt{\pi(d-3)R(d)}}\exp\left(\frac{1}{4d-11}\right)\cos\phi \\
&> 1 - \frac{\sqrt{2d-4}}{\sqrt{\pi R(d)}}\exp\left(\frac{1}{4d-9}\right)
\end{aligned} \tag{62}$$

Overall, we have $\forall d \in \mathbb{N}_+$,

$$\mathbb{P}\left(\langle s, e \rangle^2 \geq \frac{1}{R(d)}\right) > \min\left\{\frac{2\arccos\left(\frac{1}{\sqrt{R(d)}}\right)}{\pi}, \left|1 - \frac{\sqrt{2d-4}}{\sqrt{\pi R(d)}}\exp\left(\frac{1}{4d-9}\right)\right|\right\}. \tag{63}$$

$\square$

## D.1 PROOF TO THEOREM 3.4

*Proof.* Let $\mathcal{A}^d = (a_i)_{i \in [d]} \overset{i.i.d.}{\sim} \mathrm{Unif}(\mathbb{S}^{d-1}(\sqrt{d}))$. We consider the random ReLU NN function class to be

$$\mathcal{F}_{relu}(\mathcal{A}^d) = \left\{f(w, A, x) = \frac{1}{\sqrt{d}}\sum_{i=1}^m w_i \sigma\left(x^\top a_i\right) : w_i \in \mathbb{R}, i \in [m]\right\}$$

where $A = [a_1, ..., a_m] \in \mathbb{R}^{d \times m}$. The empirical minimizer of the source domain is

$$\hat{w} = \min_{w \in \mathbb{R}^d} \frac{1}{n}\sum_{x_i, y_i \in S} \ell(f(w, A, x_i), y_i) = \frac{1}{n}\sum_{z_i \in S} \ell_{\hat{w}, A}(z_i). \tag{64}$$

Then with the chain rule, the first derivative of any input $\boldsymbol{x}$ at $\hat{\boldsymbol{w}}$ will be

$$
\begin{aligned}
\nabla_{\boldsymbol{x}} \ell\left(f(\hat{\boldsymbol{w}}, A, \boldsymbol{x}), y\right) &= \frac{\partial \ell\left(f(\hat{\boldsymbol{w}}, A, \boldsymbol{x}), y\right)}{\partial f(\hat{\boldsymbol{w}}, A, \boldsymbol{x})} \frac{\partial \ell f(\hat{\boldsymbol{w}}, A, \boldsymbol{x})}{\sigma(A\boldsymbol{x})} \frac{\partial \sigma(A\boldsymbol{x})}{\partial \boldsymbol{x}} \\
&= \frac{D_f^1(\boldsymbol{x}, y)}{\sqrt{d}} \sum_{i=1}^m \hat{w}_i \boldsymbol{a}_i \mathbb{I}\left\{\boldsymbol{a}_i^\top \boldsymbol{x} \geq 0\right\} \\
&= \frac{D_f^1(\boldsymbol{x}, y)}{\sqrt{d}} \hat{\boldsymbol{w}}^\top \sigma'(A\boldsymbol{x})
\end{aligned}
\tag{65}
$$

where a short notation $D_f^1(\boldsymbol{x}, y)$ denotes the first order directional derivative of $f(\hat{\boldsymbol{w}}, A, \boldsymbol{x}) \, \sigma'(A\boldsymbol{x}) \in \mathbb{R}^{m \times d}$ is the Jacobian matrix w.r.t. input $\boldsymbol{x}$. Apparently, the second order derivative is represented as $D_f^2(\boldsymbol{x}, y)$, thus the Hessian will be

$$
\begin{aligned}
\nabla_{\boldsymbol{x}}^2 \ell(f(\hat{\boldsymbol{w}}, A, \boldsymbol{x}), y) &= \frac{D_f^2(\boldsymbol{x}, y)}{d} \sum_{i=1}^m \sum_{j=1}^m \hat{w}_i \hat{w}_j \boldsymbol{a}_i \boldsymbol{a}_j^\top \mathbb{I}\left\{\boldsymbol{a}_i^\top \boldsymbol{x} \geq 0, \boldsymbol{a}_j^\top \boldsymbol{x} \geq 0\right\} \\
&= \frac{D_f^2(\boldsymbol{x}, y)}{d} \sigma'(A\boldsymbol{x})^\top \hat{\boldsymbol{w}} \hat{\boldsymbol{w}}^\top \sigma'(A\boldsymbol{x})
\end{aligned}
\tag{66}
$$

Similarly, we have

$$
\nabla_y^2 \ell(f(\hat{\boldsymbol{w}}, A, \boldsymbol{x}), y) = D_y^2(\boldsymbol{x}, y) \cdot (\operatorname{sgn}(y))^2 \overset{*}{\leq} D_f^2(\boldsymbol{x}, y)
\tag{67}
$$

where $\operatorname{sgn}(y)$ is the sign function. $*$ holds under our choice of the family of loss functions.

1. Homogeneity in regression, i.e. L1, MSE, MAE, Huber Loss, we have $|D_y^2(\boldsymbol{x}, y)| = |D_f^2(\boldsymbol{x}, y)|$;

2. (Binary) Cross-Entropy Loss:

$$
D_y^2(\boldsymbol{x}, y) = \partial^2\left(y \sum_i \exp(\boldsymbol{x}) / \sum_{c=1}^C \exp(\boldsymbol{x}_c)\right) / \partial y^2 = 0;
$$

3. Negative Log Likelihood (NLL) loss: $D_y^2(\boldsymbol{x}, y) = 0$.

Besides, as a convex loss function, $D_y^2(\boldsymbol{x}, y) \geq 0$. Hence, the range of $D_y^2(\boldsymbol{x}, y)$ will be $[0, D_f^2(\boldsymbol{x}, y)]$.

To combine with robustness, we denote $\boldsymbol{z} = (\boldsymbol{x}; y), \in \mathbb{R}^{d+1}$. Therefore, the Hessian of $\boldsymbol{z}$ will be

$$
H(\boldsymbol{z}|S, A) := \begin{bmatrix} \nabla_{\boldsymbol{x}}^2 \ell(f(\hat{\boldsymbol{w}}, A, \boldsymbol{x}), y) & \frac{\partial^2 \ell(f(\hat{\boldsymbol{w}}, A, \boldsymbol{x}), y))}{\partial y \partial \boldsymbol{x}} \\ \left(\frac{\partial^2 \ell(f(\hat{\boldsymbol{w}}, A, \boldsymbol{x}), y)}{\partial y \partial \boldsymbol{x}}\right)^\top & \nabla_y^2(\ell(f(\hat{\boldsymbol{w}}, A, \boldsymbol{x}), y)) \end{bmatrix}.
\tag{68}
$$

With Lemma D.4, the spectral norm of Hessian $\boldsymbol{z}$ will be bounded by

$$
\|H(\boldsymbol{z})\| \leq \left\|\nabla_{\boldsymbol{x}}^2 \ell(f(\hat{\boldsymbol{w}}, A, \boldsymbol{x}), y)\right\| + \left|\nabla_y^2 \ell(f(\hat{\boldsymbol{w}}, A, \boldsymbol{x}), y)\right|.
\tag{69}
$$

The first term in (69) can be further bounded by

$$
\begin{aligned}
\left\|D_f^2(\boldsymbol{x}, y) \sigma'(A\boldsymbol{x})^\top \hat{\boldsymbol{w}} \hat{\boldsymbol{w}}^\top \sigma'(A\boldsymbol{x})\right\| &\leq |D_f^2(\boldsymbol{x}, y)| \left\|\hat{\boldsymbol{w}} \hat{\boldsymbol{w}}^\top\right\| \left\|\sigma'(A\boldsymbol{x}) \sigma'(A\boldsymbol{x})^\top\right\| \\
&= D_f^2(\boldsymbol{x}, y) \|\hat{\boldsymbol{w}}\|^2 \left\|\sigma'(A\boldsymbol{x}) \sigma'(A\boldsymbol{x})^\top\right\|
\end{aligned}
\tag{70}
$$

where the convexity of loss functions $\forall \boldsymbol{x}, y, D_f^2(\boldsymbol{x}, y) \geq 0$ supports the last equation. The right term has the facts that

$$
\|\sigma'(A\boldsymbol{x}) \sigma'(A\boldsymbol{x})^\top\| \leq \|\sigma'(A\boldsymbol{x}) \sigma'(A\boldsymbol{x})^\top\|_F = \operatorname{tr}\left(\sigma'(A\boldsymbol{x}) \sigma'(A\boldsymbol{x})^\top\right).
\tag{71}
$$

In summary, we have the following inequality:

$$
\begin{aligned}
\|H(\boldsymbol{z})\| &\leq D_f^2(\boldsymbol{x}, y) \left( \frac{1}{d} \|\hat{\boldsymbol{w}}\|^2 \left\| \sigma'(A\boldsymbol{x})\sigma'(A\boldsymbol{x})^\top \right\| + 1 \right) \\
&\leq D_f^2(\boldsymbol{x}, y) \left( \frac{1}{d} \|\hat{\boldsymbol{w}}\|^2 \operatorname{tr}\left( \sigma'(A\boldsymbol{x})\sigma'(A\boldsymbol{x})^\top \right) + 1 \right) \\
&= D_f^2(\boldsymbol{x}, y) \left( \frac{1}{d} \|\hat{\boldsymbol{w}}\|^2 \sum_{j=1}^m \|\boldsymbol{a}_j\|^2 \, \mathbb{I}\left\{ \boldsymbol{a}_j^\top \boldsymbol{x} \geq 0 \right\} + 1 \right)
\end{aligned}
\tag{72}
$$

In Lemma D.2, it depends on some $\boldsymbol{z}_i = (\boldsymbol{x}_i, y_i) \in S \cap C_i$ that

$$
\begin{aligned}
\epsilon(S, A) &\leq \max_{i \in [K]} \frac{\rho_i(L)^2}{2L^2} \left( \|H(\boldsymbol{z}_i(A))\| + \frac{4\rho_i(L)}{3} \right) \\
&\leq \max_{i \in [K]} \frac{\rho_i(L)^2}{2L^2} \left( D_f^2(\boldsymbol{x}_i, y_i) \left( \frac{\|\hat{\boldsymbol{w}}\|^2}{d} \sum_{j=1}^m \|\boldsymbol{a}_j\|^2 \, \mathbb{I}\left\{ \boldsymbol{a}_j^\top \boldsymbol{x}_t \geq 0 \right\} + 1 \right) + \frac{4\rho_i(L)}{3} \right) \\
&\leq \frac{\rho_{\max}(L)^2}{2L^2} \left( \frac{D_f^2(\boldsymbol{x}_k, y_k)}{d} \|\hat{\boldsymbol{w}}\|^2 \sum_{j=1}^m \|\boldsymbol{a}_j\|^2 \, \mathbb{I}\left\{ \boldsymbol{a}_j^\top \boldsymbol{x}_k \geq 0 \right\} + \frac{4\rho_{\max}(L)}{3} + \tilde{o}_\kappa \right)
\end{aligned}
\tag{73}
$$

where $\rho_{\max}(L) = \max\{\rho_i(L)\}_{i=0}^K$, the $\tilde{o}_\kappa = \mathcal{O}(\ell''(f, \boldsymbol{x}_k, y_k)\|\hat{\boldsymbol{w}}\|^2 \sum_{j=1}^m \|\boldsymbol{a}_j\|^2 \, \mathbb{I}\left\{ \boldsymbol{a}_j^\top \boldsymbol{x}_k \geq 0 \right\} d/m)$ is a smaller order term compared to first term, since $m \gg d$. Last equality, the maximum can be taken as we find maximum $(\boldsymbol{x}_k, y_k) \in \hat{C} \in \{C_i\}_{i \in [K]}$. Because $\boldsymbol{x}_k, y_k \in S$ is one of the training sample $k \in [n]$, there must exist $n' \in [0, n]$ that

$$
\begin{aligned}
D_f^2(\boldsymbol{x}_k, y_k)\|\hat{\boldsymbol{w}}\|^2 &\sum_{j=1}^m \|\boldsymbol{a}_j\|^2 \, \mathbb{I}\left\{ \boldsymbol{a}_j^\top \boldsymbol{x}_k \geq 0 \right\} \\
&= \|\hat{\boldsymbol{w}}\|^2 \frac{n'}{n} \sum_{k=1}^n \frac{D_f^2(\boldsymbol{x}_k, y_k)}{d} \sum_{j=1}^m \|\boldsymbol{a}_j\|^2 \mathbb{I}\{\boldsymbol{a}_j^\top \boldsymbol{x}_k \geq 0\}.
\end{aligned}
\tag{74}
$$

Recall that the sharpness of parameter $\hat{\boldsymbol{w}}$ is defined by

$$
\begin{aligned}
\kappa(\hat{\boldsymbol{w}}, S, A) &:= \|\hat{\boldsymbol{w}}\|^2 \operatorname{tr}[H_{S,A}(\hat{\boldsymbol{w}})] \\
&= \|\hat{\boldsymbol{w}}\|^2 \frac{1}{n} \sum_{j=1}^n D_f^2(\boldsymbol{x}_j, y_j) \cdot \operatorname{tr}\left( \sigma(A\boldsymbol{x}_j)\sigma(A\boldsymbol{x}_j)^\top \right) \\
&= \|\hat{\boldsymbol{w}}\|^2 \frac{1}{n} \sum_{j=1}^n \frac{D_f^2(\boldsymbol{x}_j, y_j)}{d} \sum_{i=1}^m (\boldsymbol{a}_i^\top \boldsymbol{x}_j)^2 \mathbb{I}\left\{ \boldsymbol{a}_i^\top \boldsymbol{x}_j \geq 0 \right\}.
\end{aligned}
\tag{75}
$$

Let the $\xi_i = \boldsymbol{a}_i^\top \boldsymbol{x} \sim D(\xi)$ and the expectation of $\mathbb{E}(\xi_i > 0) = q_i$ where $D(\xi)$ is some rotationally invariant distribution, i.e. uniform or normal distribution. Under this circumstance, the sample mean of $\xi_i$ still obeys the same family distribution as $D(t\xi)$. Thus, we have

$$
\begin{aligned}
\mathbb{P}\left( \sum_{j=1}^m \xi_i \mathbb{I}\left\{ \xi_i \geq 0 \right\} \geq \sum_{j=1}^m \|\boldsymbol{a}_j\|^2 \mathbb{I}\left\{ \xi_i \geq 0 \right\} \right) &= \mathbb{P}\left( \sum_{j=1}^{q_i} \xi_i \geq \sum_{j=1}^{q_i} \|\boldsymbol{a}_j\|^2 \right) \\
&= \mathbb{P}((\boldsymbol{a}^\top \boldsymbol{x})^2 \geq \|\boldsymbol{a}\|^2) = \mathbb{E}_{\boldsymbol{x}} p(\boldsymbol{x})
\end{aligned}
\tag{76}
$$

With Lemma D.5, we have at least a probability at

$$
\mathbb{P}((\boldsymbol{a}^\top \boldsymbol{x})^2 \geq \|\boldsymbol{a}\|^2) = \min\left\{ \frac{2}{\pi} \arccos\left( R(d)^{-\frac{1}{2}} \right), \left| 1 - \frac{\sqrt{2d-4}}{\sqrt{\pi R(d)}} e^{\frac{1}{4d-9}} \right| \right\}
\tag{77}
$$

the following inequality holds,

$$\epsilon(S, A) \leq \frac{\rho_{\max}(L)^2}{2L^2} \left( n'\kappa(\hat{\boldsymbol{w}}, S, A) + \frac{4\rho_{\max}(L)}{3} + \tilde{o}_\kappa \right)$$
$$= \frac{\rho_{\max}(L)^2}{2L^2} \left( \left[ n' + \mathcal{O}\left( \frac{d}{m} \right) \right] \kappa(\hat{\boldsymbol{w}}, S, A) + \frac{4}{3}\rho_{\max}(L) \right) \tag{78}$$

$\square$

## D.2 PROOF TO COROLLARY 3.5

**Corollary D.6** (Restatement of Corollary 3.5). *Let $\hat{w}_{\min}$ be the minimum value of $|\hat{\boldsymbol{w}}|$. Suppose $\forall \boldsymbol{x} \sim \mathrm{Unif}(\mathbb{S}^{d-1}(\sqrt{d}))$ and $|\partial^2 \ell(f(\hat{\boldsymbol{w}}, A, \boldsymbol{x}), y)/\partial f^2|$ is bounded by $[\tilde{M}_1, \tilde{M}_2]$. If $m > d$, $\rho_{\max}(L) < (\hat{w}_{\min}^2 \tilde{M}_1 \tilde{\sigma}(d, m))/(2d)$ for any $A = (\boldsymbol{a}_1, ..., \boldsymbol{a}_m), \boldsymbol{a}_i \sim \mathrm{Unif}(\mathbb{S}^{d-1}(\sqrt{d}))\forall i \in [m]$, taking expectation over all $\boldsymbol{x}_j \in \mathrm{Unif}(\mathbb{S}^{d-1}(\sqrt{d}))$ in $S$, we have*

$$\mathbb{E}_{S,A} [\epsilon(S, A)] \leq \mathbb{E}_{S,A} \frac{7\rho_{\max}(L)^2}{6L^2} \left( n'\kappa(\hat{\boldsymbol{w}}, S, A) + \tilde{M}_2 \right). \tag{79}$$

*where $\tilde{\sigma}(d, m) = \mathbb{E}_{\boldsymbol{a}} \lambda_{\min}(\sum_{i=1}^m \boldsymbol{a}_i \boldsymbol{a}_i^\top G_{ii}) > 0$ is the minimum eigenvalue and $G_{ii}$ is product constant of Gegenbauer polynomials*

$$G_{ij} = \sum_{t=0}^{\infty} \lambda_{d,t}^2(\sigma) N_{d,t}^2 Q_t^{(d)}(\langle \boldsymbol{a}_i, \boldsymbol{a}_j \rangle / \sqrt{d}).$$

*Proof.* In our main theorem, with some probability, we have the following relation

$$\epsilon(S, A) \leq \frac{\rho_{\max}(L)^2}{2L^2} \left( n'\kappa(\hat{\boldsymbol{w}}, S, A) + \frac{4\rho_{\max}(L)}{3} + d_\kappa \right).$$

So, we are concerned about the relation between the $\kappa(\hat{\boldsymbol{w}}, S, A)$ second term. If $\rho_i(L) < \kappa(\hat{\boldsymbol{w}}, S, A)$, we may say the RHS is dominated by sharpness term $\kappa(\hat{\boldsymbol{w}}, S, A)$ as well as the main effect is taken by the sharpness. As suggested in Lemma D.1, we have

$$\mathbb{E}_{\boldsymbol{x}^* \sim \mathrm{Unif}(\mathbb{S}^{d-1}(\sqrt{d}))} H(\boldsymbol{x}^*) \succeq \frac{\hat{w}_{\min}^2 \tilde{M}_1 \tilde{\sigma}_A(m, d)}{d} I_d \tag{80}$$

where $\boldsymbol{x}^*$ is the global minimum over the whole set. As defined in (38), the following condition holds true

$$\exists \boldsymbol{z}_i^*(A) \in \mathcal{M}_i, \|\boldsymbol{z}_i(A) - \boldsymbol{z}_i^*(A)\| \leq \frac{\rho_i(L)}{L}, \tag{81}$$

and with Hessian Lipschitz, the relation is almost surely for arbitrary $\boldsymbol{x}$ that

$$\mathbb{E}_{\boldsymbol{z}_i^*(A)} \|H(\boldsymbol{z}_i(A)) - H(\boldsymbol{z}_i^*(A))\| \leq L\|\boldsymbol{z}_i(A) - \boldsymbol{z}_i^*(A)\| \leq \rho_i(L)$$
$$\leq \left\| \frac{\hat{w}_{\min}^2 \tilde{M}_1 \tilde{\sigma}(d, m)}{2d} I_d \right\|$$
$$\leq \mathbb{E}_{A, \boldsymbol{x}^*} \frac{1}{2} \|H(\boldsymbol{x}^*)\|$$
$$\leq \mathbb{E}_{\boldsymbol{z}_i^*(A)} \frac{1}{2} \|H(\boldsymbol{z}_i^*(A))\|. \tag{82}$$

Obviously, $\|H(\boldsymbol{z}_i^*(A))\| > 2\rho_i(L)$. Following Lemma A.2 of Zhang et al. (2019a), for $\boldsymbol{z}_i^*(A), \forall \boldsymbol{z}(A) \in C_i$, we have a similar result that

$$\tilde{\sigma}_{\min}(H(\boldsymbol{z}(A))) \geq \tilde{\sigma}_{\min}(H(\boldsymbol{z}_i^*(A)) - \|H(\boldsymbol{z}(A)) - H(\boldsymbol{z}_i^*(A))\| \geq \rho_i(L) \tag{83}$$

where $\tilde{\sigma}_{\min}$ denotes the minimum singular value. With Lemma D.2, we know that

$$\mathbb{E}_{S,A} \epsilon(S, A) \leq \mathbb{E}_{S,A} \max_{i \in [K]} \frac{\rho_i(L)^2}{2L^2} \left( \|H(\boldsymbol{z}_i(A))\| + \frac{4\rho_i(L)}{3} \right). \tag{84}$$

We also have another condition that

$$|D_f^2(\boldsymbol{x}, y)| := \left| \frac{\partial^2 \ell(f(\hat{\boldsymbol{w}}, A, \boldsymbol{x}), y)}{\partial f^2} \right| \in [\tilde{M}_1, \tilde{M}_2], \forall \boldsymbol{x}, y. \tag{85}$$

Combine all these results, we finally have

$$
\begin{aligned}
\mathbb{E}_{S,A} \epsilon(S, A) &\leq \mathbb{E}_{S,A} \max_{i \in [K]} \frac{\rho_i(L)^2}{2L^2} \left(1 + \frac{4}{3}\right) \|H(\boldsymbol{z}_i(A))\| \\
&\leq \mathbb{E}_{S,A} \frac{7\rho_{\max}(L)^2}{6L^2} \left( \frac{D_f^2(\boldsymbol{x}_k, y_k)}{d} \|\hat{\boldsymbol{w}}\|^2 \sum_{j=1}^m \|\boldsymbol{a}_j\|^2 \mathbb{I}\left\{\boldsymbol{a}_j^\top \boldsymbol{x}_k \geq 0\right\} + \tilde{M}_2 \right).
\end{aligned}
\tag{86}
$$

Recall the definition of $\kappa(\hat{\boldsymbol{w}}, S, A)$ in the main theorem that

$$\mathbb{E}_{S,A} \kappa(\hat{\boldsymbol{w}}, S, A) \leq \mathbb{E}_{\{\boldsymbol{x}\}^n} \|\hat{\boldsymbol{w}}\|^2 \frac{1}{n} \sum_{j=1}^n \frac{D_f^2(\boldsymbol{x}_j, y_j)}{d} \sum_{i=1}^m (\boldsymbol{a}_i^\top \boldsymbol{x}_j)^2 \mathbb{I}\left\{\boldsymbol{a}_i^\top \boldsymbol{x}_j \geq 0\right\} \tag{87}$$

Look at the second sum, we have

$$
\begin{aligned}
\mathbb{E}_{\{\boldsymbol{x}_j\}^n, \{\boldsymbol{a}_i\}^m} \sum_{i=1}^m (\boldsymbol{a}_i^\top \boldsymbol{x}_j)^2 \mathbb{I}\left\{\boldsymbol{a}_i^\top \boldsymbol{x}_j \geq 0\right\} &= \sum_{i=1}^m \mathbb{E}_{\boldsymbol{x}_j} \mathbb{E}_{\boldsymbol{a}_i} \|\boldsymbol{a}_i\|^2 \|\boldsymbol{x}_j\|^2 \cos^2(\beta) \mathbb{I}\left\{\boldsymbol{a}_i^\top \boldsymbol{x}_j \geq 0\right\} \\
&= \sum_{i=1}^m \mathbb{E}_{\boldsymbol{x}_j} \mathbb{E}_{\boldsymbol{a}_i} \|\boldsymbol{a}_i\|^2 \|\mathbb{I}\left\{\boldsymbol{a}_i^\top \boldsymbol{x}_j \geq 0\right\} d \cos^2(\beta).
\end{aligned}
\tag{88}
$$

Suppose $\boldsymbol{x}$ and $\boldsymbol{a}$ are i.i.d. from $\text{Unif}(\mathbb{S}^{d-1}(1))$, let $u = \langle \boldsymbol{x}, \boldsymbol{a} \rangle$, we have a well-known result that

$$
\begin{aligned}
\mathbb{E}_u[u^2] &= \mathbb{E}[\langle \boldsymbol{x}, \boldsymbol{a} \rangle^2] \\
&= \mathbb{E}[\|\boldsymbol{x}\|^2 \|\boldsymbol{a}\|^2 \cos^2(\beta)] \\
&= \mathbb{E}[\cos^2(\beta)] = \frac{1}{d}, \quad \boldsymbol{x}, \boldsymbol{a} \in \mathbb{R}^d
\end{aligned}
\tag{89}
$$

Therefore, in (88),

$$\sum_{i=1}^m \mathbb{E}_{\boldsymbol{x}_j} \mathbb{E}_{\boldsymbol{a}_i} \|\boldsymbol{a}_i\|^2 \|\mathbb{I}\left\{\boldsymbol{a}_i^\top \boldsymbol{x}_j \geq 0\right\} d \cos^2(\beta) = \sum_{i=1}^m \mathbb{E}_{\boldsymbol{x}_j} \mathbb{E}_{\boldsymbol{a}_i} \|\boldsymbol{a}_i\|^2 \|\mathbb{I}\left\{\boldsymbol{a}_i^\top \boldsymbol{x}_j \geq 0\right\} \tag{90}$$

and we have (based on proof of main theorem),

$$
\begin{aligned}
&\mathbb{E}_{S,A} \epsilon(S, A) \\
&\leq \frac{7\rho_{\max}(L)^2}{6L^2} \left( \|\hat{\boldsymbol{w}}\|^2 \frac{n'}{n} \sum_{j=1}^n \frac{D_f^2(\boldsymbol{x}_j, y_j)}{d} \sum_{i=1}^m \mathbb{E}_{\boldsymbol{x}_j} \mathbb{E}_{\boldsymbol{a}_i} \|\boldsymbol{a}_i\|^2 \mathbb{I}\{\boldsymbol{a}_i^\top \boldsymbol{x}_j \geq 0\} + \tilde{M}_2 \right) \\
&\leq \mathbb{E}_{S,A} \frac{7\rho_{\max}(L)^2}{6L^2} \left( n' \kappa(\hat{\boldsymbol{w}}, S, A) + \tilde{M}_2 \right)
\end{aligned}
\tag{91}
$$

$\square$

# E  CASE STUDY

To better illustrate our theorems, we here give two different cases for clearly picturing intuition. The first case is the very basic model, ridge regression. As is known to us, ridge regression provides a straightforward way (by punishing the $\ell_2$ norm of the weights) to reduce the "variance" of the model in order to avoid overfitting. In this case, this mechanism is equivalent to reducing the model's sharpness.

**Example E.1.** *In ridge regression models, the robustness constant $\epsilon$ has a reverse relationship to regularization parameter $\beta$ where $\beta \uparrow$, the more probably flatter minimum $\kappa \downarrow$ and less sensitivity $\epsilon \downarrow$ of the learned model could be. Follow the previous notation that $\epsilon(S, A)$ denotes the robustness and $\kappa(\hat{\boldsymbol{\theta}}, S)$ is the sharpness on training set $S$, then we have*

$$\tau > 0, \ c \in (0, n], \ \epsilon(S, A) \leq c\kappa(\hat{\boldsymbol{\theta}}, S) + \tilde{o}_d$$

*where $\tilde{o}_d$ is a much smaller order than $\kappa(\hat{\boldsymbol{\theta}}, S)$.*

### E.1 RIDGE REGRESSION

We consider a generic response model as stated in Ali et al. (2019).

$$\boldsymbol{y}|\boldsymbol{\theta}_* \sim (X\boldsymbol{\theta}_*, \sigma^2 I)$$

ridge regression minimization problem is defined by

$$\min_{\boldsymbol{\theta}} \frac{1}{2n}\|X\boldsymbol{\theta} - \boldsymbol{y}\|^2 + \frac{\beta}{2}\|\boldsymbol{\theta}\|^2, X \in \mathbb{R}^{n\times d}, n < d. \tag{92}$$

The least-square solution of ridge regression is

$$\hat{\boldsymbol{\theta}} = \left(X^\top X + n\beta I\right)^{-1} X^\top \boldsymbol{y}. \tag{93}$$

With minimizer $\hat{\boldsymbol{\theta}}$, we now focus on its geometry w.r.t. $\boldsymbol{x}$. Let $S = \{\boldsymbol{z}\}_i^n = (X, \boldsymbol{y})$ be the training set, $(\mathcal{Z}, \Sigma, \rho)$ be a measure space. Consider the bounded sample set $\mathcal{Z}$ such that

$$\exists M > 0, \ \ \rho(\mathcal{Z}) < +\infty. \tag{94}$$

The $\mathcal{Z}$ can be partitioned into $K$ disjoint sets $\{C_i\}_{i\in[K]}$. By definition, we have robustness defined by each partition $C_i$,

$$\forall \boldsymbol{z}, \boldsymbol{z}' \in C_i, \left|\ell(\hat{\boldsymbol{\theta}}, \boldsymbol{z}) - \ell(\hat{\boldsymbol{\theta}}, \boldsymbol{z}')\right| \leq \epsilon(S, A). \tag{95}$$

For this convex function $\ell(\hat{\boldsymbol{\theta}}, \boldsymbol{z})$, we have the following upper bound in the whole sample domain

$$\begin{aligned}
\epsilon(S) &= \max_{i\in[K]} \sup_{\boldsymbol{z}, \boldsymbol{z}'\in C_i} \left|\ell(\hat{\boldsymbol{\theta}}, \boldsymbol{z}) - \ell(\hat{\boldsymbol{\theta}}, \boldsymbol{z}')\right| \\
&= \max_{i\in[K]} \sup_{\boldsymbol{z}\in C_i} \ell(\hat{\boldsymbol{\theta}}, \boldsymbol{z}) - \ell(\hat{\boldsymbol{\theta}}, \boldsymbol{z}_i^* \in C_i) \\
&\leq \sup_{\boldsymbol{z}_j\in \mathcal{Z}\cap S} \ell(\hat{\boldsymbol{\theta}}, \boldsymbol{z}_j) - \ell(\hat{\boldsymbol{\theta}}, \boldsymbol{z}^*)
\end{aligned} \tag{96}$$

where the $\boldsymbol{z}_i^*, \boldsymbol{z}^*$ are the global minimum point in $C_i$ and whole domain $\mathcal{Z} = \bigcup_i^K C_i$, respectively. $\boldsymbol{z}_j$ is a training data point that has the maximum loss difference from the optimum. Specifically, it as well as the augmented form of $\hat{\boldsymbol{\theta}}$ can be expressed as

$$\boldsymbol{z} = [x_1, ..., x_d, y]^\top, \quad \hat{\boldsymbol{\theta}}_+ = [\hat{\theta}_1, ..., \hat{\theta}_d, -1]^\top, \in \mathbb{R}^{d+1}.$$

Then the loss difference can be rewritten as

$$\ell_{\hat{\boldsymbol{\theta}}_+}(\boldsymbol{z}) = (\hat{\boldsymbol{\theta}}_+^\top \boldsymbol{z})^2 = (\hat{\boldsymbol{\theta}}^\top \boldsymbol{x} - y)^2 \Rightarrow H(\ell_{\hat{\boldsymbol{\theta}}_+}(\boldsymbol{z}))\text{is P.S.D matrix.}$$

It is a convex function with regards to $\boldsymbol{z}$ such that

$$\begin{aligned}
\epsilon(S) &\leq \sup_{\boldsymbol{z}_j\in S} \ell_{\hat{\boldsymbol{\theta}}_+}(\boldsymbol{z}_j) - \ell_{\hat{\boldsymbol{\theta}}_+}(\boldsymbol{z}^*) \\
&= \sup_{\boldsymbol{z}_j\in S} \nabla\ell_{\hat{\boldsymbol{\theta}}_+}(\boldsymbol{z}^*)^\top(\boldsymbol{z}_j - \boldsymbol{z}^*) + \frac{1}{2}(\boldsymbol{z}_j - \boldsymbol{z}^*)^\top H(\ell_{\hat{\boldsymbol{\theta}}}(\boldsymbol{z}^*))(\boldsymbol{z}_j - \boldsymbol{z}^*) \\
&= \sup_{\boldsymbol{z}_j\in S} \frac{1}{2}(\boldsymbol{z}_j - \boldsymbol{z}^*)^\top H(\ell_{\hat{\boldsymbol{\theta}}_+}(\boldsymbol{z}^*))(\boldsymbol{z}_j - \boldsymbol{z}^*) \\
&\leq \sup_{\boldsymbol{z}_j\in S} \frac{1}{2}\|H(\ell_{\hat{\boldsymbol{\theta}}_+}(\boldsymbol{z}^*))\|\|\boldsymbol{z}_j - \boldsymbol{z}^*\|^2
\end{aligned} \tag{97}$$

where the second equality is supported by convexity and the third equality is due to $\ell_{\hat{\boldsymbol{\theta}}_+}(\boldsymbol{z}_j) = 0$. Further, with Lemma D.4, we have

$$\|H(\ell_{\hat{\boldsymbol{\theta}}_+}(\boldsymbol{z}^*))\| = \left\|\begin{bmatrix} \hat{\boldsymbol{\theta}}\hat{\boldsymbol{\theta}}^\top & \frac{\partial^2 \ell_{\hat{\boldsymbol{\theta}}_+}(\boldsymbol{z}_j))}{\partial y \partial \boldsymbol{x}} \\ \left(\frac{\partial^2 \ell_{\hat{\boldsymbol{\theta}}_+}(\boldsymbol{z}_j))}{\partial \boldsymbol{x} \partial y}\right)^\top & 1 \end{bmatrix}\right\| \leq \left\|\hat{\boldsymbol{\theta}}\hat{\boldsymbol{\theta}}^\top\right\| + 1. \tag{98}$$

In (97), we can also bound the norm of the input difference by

$$\|\boldsymbol{z}_j - \boldsymbol{z}^*\|^2 \leq \|\boldsymbol{x}_j - \boldsymbol{x}^*\|^2 + (y_j - y^*)^2 \tag{99}$$

and then for simplicity, assume $\|\boldsymbol{x}^*\|^2 \leq \|\boldsymbol{x}_j\|^2$ we have

$$
\begin{aligned}
\epsilon(S) &\leq \sup_{\boldsymbol{x}_j \in S} \frac{1}{2} \left( \left\| \hat{\boldsymbol{\theta}} \hat{\boldsymbol{\theta}}^\top \right\| + 1 \right) \left( \|\boldsymbol{x}_j - \boldsymbol{x}^*\|^2 + (y_j - y^*)^2 \right) \\
&\leq \sup_{\boldsymbol{x}_j \in S} \frac{1}{2} \left( \left\| \hat{\boldsymbol{\theta}} \right\|^2 + 1 \right) \left( \|\boldsymbol{x}_j\|^2 + \|\boldsymbol{x}^*\|^2 + (y_j - y^*)^2 \right) \\
&\leq \sup_{\boldsymbol{x}_j \in S} \frac{1}{2} \left\| \hat{\boldsymbol{\theta}} \right\|^2 \left( \|\boldsymbol{x}_j\|^2 + \|\boldsymbol{x}^*\|^2 \right) + \mathcal{O}(d) \\
&\leq \sup_{\boldsymbol{x}_j \in S} \left\| \hat{\boldsymbol{\theta}} \right\|^2 \|\boldsymbol{x}_j\|^2 + \mathcal{O}(d)
\end{aligned}
\tag{100}
$$

where $\left\| \hat{\boldsymbol{\theta}} \right\|^2 \|\boldsymbol{x}_j\|^2 = \mathcal{O}(d^2)$ is the dominate term for large $d$. Now, let's look at the relation to sharpness. By definition,

$$\kappa(\hat{\boldsymbol{\theta}}, S) = \|\hat{\boldsymbol{\theta}}\|^2 \operatorname{tr}\left( H_{\hat{\boldsymbol{\theta}}}(\ell(\hat{\boldsymbol{\theta}}, S)) \right) = \|\hat{\boldsymbol{\theta}}\|^2 \operatorname{tr}\left( \frac{X^\top X}{n} + \beta I \right). \tag{101}$$

Since

$$\operatorname{tr}\left( \frac{X^\top X}{n} + \beta I \right) = \operatorname{tr}\left( \frac{X^\top X}{n} \right) + \operatorname{tr}(\beta I) = \operatorname{tr}\left( \frac{X X^\top}{n} \right) + \beta = \frac{1}{n} \sum_j^n \|\boldsymbol{x}_j\|^2 + \beta, \tag{102}$$

so we have

$$\kappa(\hat{\boldsymbol{\theta}}, S) = \|\hat{\boldsymbol{\theta}}\|^2 \frac{1}{n} \sum_j^n \|\boldsymbol{x}_j\|^2 + \beta \|\hat{\boldsymbol{\theta}}\|^2. \tag{103}$$

As is known to us, the "variance" of ridge estimator Ali et al. (2019) can be defined by

$$\operatorname{Var}(\hat{\boldsymbol{\theta}}) \triangleq \operatorname{tr}\left( \hat{\boldsymbol{\theta}} \hat{\boldsymbol{\theta}}^\top \right) = \operatorname{tr}\left( \left( X^T X + n\beta I \right)^{-1} X^T \boldsymbol{y} \boldsymbol{y}^\top X \left( X^T X + n\beta I \right)^{-1} \right). \tag{104}$$

Note that $\hat{\boldsymbol{\theta}} \hat{\boldsymbol{\theta}}^\top$ is a PSD with $\operatorname{rank}(\hat{\boldsymbol{\theta}} \hat{\boldsymbol{\theta}}^\top) = 1$, thus it has only one eigenvalue $\lambda(\hat{\boldsymbol{\theta}} \hat{\boldsymbol{\theta}}^\top) = \|\hat{\boldsymbol{\theta}}\|^2 > 0$.

$$\operatorname{Var}(\theta) = \operatorname{tr}\left( \hat{\boldsymbol{\theta}} \hat{\boldsymbol{\theta}}^\top \right) = \|\hat{\boldsymbol{\theta}}\|^2 = \mathcal{O}(\beta^{-2}) \tag{105}$$

By definition, the covariance matrix $\mathbb{E}[\boldsymbol{y} \boldsymbol{y}^\top]$ is a diagonal matrix with entries of $\sigma^2$. Averagely, we have

$$
\begin{aligned}
\operatorname{tr}\left( \hat{\boldsymbol{\theta}} \hat{\boldsymbol{\theta}}^\top \right) &= \sigma^2 \operatorname{tr}\left[ \left( X^\top X + n\beta I \right)^{-1} X^\top X \left( X^\top X + n\beta I \right)^{-1} \right] \\
&= \frac{\sigma^2}{n} \operatorname{tr}\left[ \frac{X^\top X}{n} \left( \frac{X^\top X}{n} + \beta I \right)^{-2} \right] \\
&= \frac{\sigma^2}{n} \sum_{i=1}^d \frac{\lambda_i(X^\top X / n)}{\left( \lambda_i(X^\top X / n) + \beta \right)^2}
\end{aligned}
\tag{106}
$$

where $\frac{X^\top X}{n}$ and $\left( \frac{X^T X}{n} + \beta I \right)^{-1}$ are simultaneously diagonalizable and commutable. Therefore, the greater $\beta$ is, the smaller $\operatorname{tr}\left( \hat{\boldsymbol{\theta}} \hat{\boldsymbol{\theta}}^\top \right)$ is.

**Conclusions**   From our above analysis, we have the following conditions.

- Upper bound of robustness $\epsilon(S) \leq \sup_{\boldsymbol{x}_j \in S} \|\hat{\boldsymbol{\theta}}\|^2 \|\boldsymbol{x}_j\|^2 + \mathcal{O}(d)$.

- Sharpness expression $\kappa(\hat{\boldsymbol{\theta}}, S) = \|\hat{\boldsymbol{\theta}}\|^2 \left( \frac{1}{n} \sum_{\boldsymbol{x}_j, y_j \in S} \|\boldsymbol{x}_j\|^2 + \beta \right)$.

- Variance expression $\text{Var}(\theta) = \|\hat{\boldsymbol{\theta}}\|^2$.

**1)** First, let's discuss how $\beta$ influences sharpness.

$$\kappa(\hat{\boldsymbol{\theta}}, S) = \|\hat{\boldsymbol{\theta}}\|^2 \,\text{tr}\left(H_{\hat{\boldsymbol{\theta}}}(\ell(\hat{\boldsymbol{\theta}}, S))\right)$$

$$= \boldsymbol{y}^\top X \left(X^\top X + n\beta I\right)^{-2} X^\top \boldsymbol{y} \left(\frac{1}{n} \sum_{\boldsymbol{x}_j, y_j \in S} \|\boldsymbol{x}_j\|^2 + \beta\right) \tag{107}$$

$$= \mathcal{O}(\beta^{-1}).$$

As dictated in above equation, the $\kappa(\hat{\boldsymbol{\theta}}, S) = \mathcal{O}(\beta^{-1})$ where sharpness holds an inverse relationship to $\beta$.

**2)** Now, it's trivial to get the relationship between robustness and sharpness by combining the first two points. Because $\boldsymbol{x}_j, y_j \in S$ in supermum is one of the training samples there exists a constant $c < n$ and $\tilde{o}_d = o(d^2)$ such that

$$\epsilon(S) \leq \frac{c}{n} \sum_j^n \left(\|\hat{\boldsymbol{\theta}}\|^2 \|\boldsymbol{x}_j\|^2\right) + \tilde{o}_d$$

$$\leq \frac{c}{n} \sum_j^n \left(\|\hat{\boldsymbol{\theta}}\|^2 \|\boldsymbol{x}_j\|^2\right) + c\beta \|\hat{\boldsymbol{\theta}}\|^2 + \tilde{o}_d \tag{108}$$

$$= c\kappa(\hat{\boldsymbol{\theta}}, S) + \tilde{o}_d.$$

This relation is consistent with Theorem 3.4 where the robustness is upper bounded by $n'$ times sharpness $\kappa(\hat{\boldsymbol{\theta}}, S)$. Besides, the relation is simpler here, where robustness only depends on sharpness without other coefficients before $\kappa(\hat{\boldsymbol{\theta}}, S)$.

**3)** Finally, the relation between $\beta$ and robustness will be

$$\epsilon(S) \leq \frac{c}{n} \sum_j^n \left(\text{Var}(\theta)\|\boldsymbol{x}_j\|^2\right) + \tilde{o}_d = \frac{c\sigma^2}{n} \sum_{i=1}^d \frac{\lambda_i(X^\top X/n)}{\left(\lambda_i(X^\top X/n) + \beta\right)^2} \frac{\sum_j^n \|\boldsymbol{x}_j\|^2}{n} + \tilde{o}_d$$

$$= \frac{\sigma^2}{n} \sum_{i=1}^d \frac{\lambda_i(X^\top X/n)}{\left(\lambda_i(X^\top X/n) + \beta\right)^2} \frac{\sum_j^d \lambda_j(X^\top X/n)}{n} + \tilde{o}_d \tag{109}$$

$$= \mathcal{O}(\beta^{-2}).$$

where the $\epsilon(S)$ somehow is the order of $\mathcal{O}(\beta^{-2})$ in the limit of $\beta$. It's clear to us that the greater $\beta$ is, the less sensitive (more robust) model we can get. In practical, we show that "over-robust" may hurt the model's performance (we show the detail empirically in Figure 7).

### E.2 2-LAYER DIAGONAL LINEAR NETWORK CLASSIFICATION

However, our main theorem assumes the loss function satisfying Polyak-Łojasiewicz (PŁ) condition. To extend our result to a more general case, here we study a 2-layer diagonal linear network classification problem whose loss is exponential-based and not satisfied the PŁ condition.

**Example E.2.** *We consider a classification problem using a 2-layer diagonal linear network with exp-loss. The robustness $\epsilon(S, A)$ has a similar relationship in Theorem 3.4. Given training set $S$, after iterations $t > T_\epsilon$, $\exists C_2 > 0$, $\epsilon(S, A) \leq C_2 \sup_{t \geq T_\epsilon} \kappa(\boldsymbol{\theta}(t), S)$.*

Given a training set $S = (X, y)$, $X = [\boldsymbol{x}_1, ..., \boldsymbol{x}_n]$, $X \in \mathbb{R}^{d \times n}$. A depth-2 diagonal linear networks with parameters $\boldsymbol{u} = [\boldsymbol{u}_+, \boldsymbol{u}_-]^\top \in \mathbb{R}^{2d}$ specified by:

$$f(\boldsymbol{u}, \boldsymbol{x}) = \langle \boldsymbol{u}_+^2 - \boldsymbol{u}_-^2, \boldsymbol{x} \rangle$$

We consider exponential loss where $\mathcal{L}(t) = \frac{1}{n} \sum_{i=1}^n \exp(-\boldsymbol{x}_i^\top \boldsymbol{\theta}(t) y_i)$ and $y_i \in \{-1, 1\}$. It has the same tail behavior and thus similar asymptotic properties as the logistic or cross-entropy loss. WLOG,

we assume $\forall i \in [n] : y_i = 1$ such that $\boldsymbol{x}_i = y_i \boldsymbol{x}_i$. Suggest by Moroshko et al. (2020), we have

$$\boldsymbol{\theta}(t) = 2\alpha^2 \sinh\left(4X \int_0^t \mathbf{r}(s)ds\right)$$

where $\mathbf{r}(t) = \frac{1}{n}\exp(-X^\top \boldsymbol{\theta}(t))$ and $\|\mathbf{r}(t)\|_1 = \mathcal{L}(t)$. Note that

$$\frac{d\boldsymbol{\theta}(t)}{dt} = \frac{4}{n}\sqrt{\boldsymbol{\theta}^2(t) + 4\alpha^4 \mathbf{1}} \circ X \exp\left(-X^\top \boldsymbol{\theta}(t)\right) = A(t)X\mathbf{r}(t) \tag{110}$$

where $A(t) = \operatorname{diag}\left(4\sqrt{\boldsymbol{\theta}^2(t) + 4\alpha^4 \mathbf{1}}\right)$.

**Part i, sharpness.** First derivative and Hessian can be obtained by

$$\nabla_{\boldsymbol{\theta}}\mathcal{L}(t) = -(X\mathbf{r}(t))^\top$$
$$H_{\boldsymbol{\theta}}(\mathcal{L}(t)) = -\frac{\partial(X\mathbf{r}(t))^\top}{\partial\boldsymbol{\theta}(t)} = \sum_{i=1}^n \mathbf{r}_i(t)X_i X_i^\top. \tag{111}$$

With the definition of sharpness, we can get

$$\begin{aligned}
\kappa(\boldsymbol{\theta}(t), S) &= \|\boldsymbol{\theta}(t)\|^2 \operatorname{tr}(H_{\boldsymbol{\theta}}[\mathcal{L}(t)]) \\
&= \|\boldsymbol{\theta}(t)\|^2 \operatorname{tr}\left(\sum_{i=1}^n \mathbf{r}_i(t)X_i X_i^\top\right) \\
&= \|\boldsymbol{\theta}(t)\|^2 \operatorname{tr}\left(\sum_{i=1}^n \mathbf{r}_i(t)X_i^\top X_i\right) \\
&= \|\boldsymbol{\theta}(t)\|^2 \left(\sum_{i=1}^n \mathbf{r}_i(t)\|\boldsymbol{x}_i\|^2\right).
\end{aligned} \tag{112}$$

**Part ii, robustness.** Now, let's use the same discussion of the robustness constant in the previous case. Follow the previous definition of $\epsilon(S, A)$, after some iteration number $T_\epsilon$ it is defined by

$$\epsilon(S, A) := \sup_{i \in [n], t \geq T_\epsilon} |n\mathbf{r}_j(t) - \mathbf{r}^*(t)| \tag{113}$$

where $n\mathbf{r}_j(t)$ is the (denormalized) point-wise loss of $\boldsymbol{x}_j$ and $\mathbf{r}^*(t)$ denotes the minimum loss of point $\boldsymbol{x}^*$. There exists $n' < n$, we have

$$\epsilon(S, A) = \sup_{t \geq T_\epsilon} n'\left(\|\mathbf{r}(t)\|_1 - \mathbf{r}^*(t)\right) \leq \sup_{t \geq T_\epsilon} n'\|\mathbf{r}(t)\|_1. \tag{114}$$

Let $\|\boldsymbol{x}_{\min}\| = \min_{i \in [n]} \|\boldsymbol{x}_i\|$, the above equation

$$\begin{aligned}
\epsilon(S, A) &\leq \frac{1}{\|\boldsymbol{x}_{\min}\|^2} \sup_{t \geq T_\epsilon} (n'\|\mathbf{r}(t)\|\|\boldsymbol{x}_{\min}\|^2) \\
&= \frac{n'}{\|\boldsymbol{x}_{\min}\|^2} \sup_{t \geq T_\epsilon} \left(\sum_{i=1}^n \mathbf{r}_i(t)\|\boldsymbol{x}_{\min}\|^2\right) \\
&\leq \frac{n'}{\|\boldsymbol{x}_{\min}\|^2} \sup_{t \geq T_\epsilon} \left(\sum_{i=1}^n \mathbf{r}_i(t)\|\boldsymbol{x}_i\|^2\right).
\end{aligned} \tag{115}$$

**Part iii, connection.** Compare the last part of (112) and (115), we found that robustness and sharpness depend on the same term. Further, we can say that for any step $t$, $\|\boldsymbol{\theta}(t)\|^2$ will have the upper bound.

From Lemma 11 of Moroshko et al. (2020), we have the following inequality,

$$\|\boldsymbol{\theta}(t)\|_\infty \leq 2\alpha^2 \sinh\left(\frac{\bar{x}}{2\gamma_2^2\alpha^2}\tilde{\gamma}(t)\right) \tag{116}$$

where $\mathcal{L}(t) = \exp(-\tilde{\gamma}(t))$. Then, we can bound the $\|\boldsymbol{\theta}(t)\|^2$ via:

$$C_1 \leq \|\boldsymbol{\theta}(t)\|^2 \leq d\|\boldsymbol{\theta}(t)\|_\infty^2 = 4d\alpha^4 \sinh^2\left(\frac{\bar{x}}{2\gamma_2^2\alpha^2}\tilde{\gamma}(t)\right) < \infty \tag{117}$$

Note that $C_1 > 0$ In summary, we have

$$\epsilon(S, A) \leq \frac{n'}{C_1 \|\boldsymbol{x}_{\min}\|} \sup_{t \geq T_\epsilon} \kappa(\boldsymbol{\theta}(t), S) \leq C_2 \sup_{t \geq T_\epsilon} \kappa(\boldsymbol{\theta}(t), S) \tag{118}$$

where $C_2 = \frac{n'}{C_1 \|\boldsymbol{x}_{\min}\|} > 0$ is a constant.

**Part iv, sanity check– asymptotic.** Asymptotically, as $t \to \infty$, $\mathcal{L}(t) \to 0$ while $\|\boldsymbol{\theta}(t)\|_2$ will be explode. So if sharpness $\kappa(\boldsymbol{\theta}(t), X) \to \infty$, then it will fail to imply robustness.

$$
\begin{aligned}
\kappa(\boldsymbol{\theta}(t), S) &= \|\boldsymbol{\theta}(t)\|^2 \left( \sum_{i=1}^n \mathbf{r}_i(t) \|\boldsymbol{x}_i\|^2 \right) \\
&\leq \|\boldsymbol{\theta}(t)\|^2 \left( \sum_{i=1}^n \mathbf{r}_i(t) \|\boldsymbol{x}_{\max}\|^2 \right) \\
&= \|\boldsymbol{\theta}(t)\|^2 \mathcal{L}(\boldsymbol{\theta}(t)) \|\boldsymbol{x}_{\max}\|^2 \\
&= \kappa_{\max}(\boldsymbol{\theta}(t), S)
\end{aligned}
\tag{119}
$$

Let $\kappa_{\max}(\boldsymbol{\theta}(t), X)$ be a upper bound of $\kappa(\boldsymbol{\theta}(t), X)$ at any time step $t$. The dynamics will be

$$
\begin{aligned}
\frac{d\kappa_{\max}(\boldsymbol{\theta}(t), S)}{dt} &= \nabla_{\boldsymbol{\theta}} \kappa_{\max}(\boldsymbol{\theta}(t), S) \dot{\boldsymbol{\theta}}(t) \\
&= \|\boldsymbol{x}_{\max}\|^2 \operatorname{tr}(XX^\top) \left( \|\boldsymbol{\theta}(t)\|^2 \dot{\mathcal{L}}(t) + 2\mathcal{L}(t)\boldsymbol{\theta}(t)^\top \dot{\boldsymbol{\theta}}(t) \right) \\
&= \|\boldsymbol{x}_{\max}\|^2 \operatorname{tr}(XX^\top) \left( -\|\boldsymbol{\theta}(t)\|^2 (X\boldsymbol{r}(t))^\top A(t) X\boldsymbol{r}(t) + 2\mathcal{L}(t)\boldsymbol{\theta}(t)^\top A(t) X\boldsymbol{r}(t) \right) \\
&= \|\boldsymbol{x}_{\max}\|^2 \operatorname{tr}(XX^\top) \left( 2\mathcal{L}(t)\boldsymbol{\theta}(t)^\top - \|\boldsymbol{\theta}(t)\|^2 (X\boldsymbol{r}(t))^\top \right) A(t) X\boldsymbol{r}(t)
\end{aligned}
\tag{120}
$$

As $t \to \infty$, we have $\mathcal{L}(t) \to 0$, thus it is easy to converge that

$$\lim_{t \to \infty} \frac{d\kappa_{\max}(\boldsymbol{\theta}(t), S)}{dt} = -\|\boldsymbol{x}_{\max}\|^2 \|\boldsymbol{\theta}(t)\|^2 (X\boldsymbol{r}(t))^\top A(t) X\boldsymbol{r}(t) = -\|\boldsymbol{x}_{\max}\|^2 \|\boldsymbol{\theta}(t)\|^2 \dot{\mathcal{L}}(t) = 0 \tag{121}$$

As we can see, the dynamics of the derivative of $\kappa_{\max}(\boldsymbol{\theta}(t), X)$ is decreasing to 0 as $t \to \infty$ which means the sharpness $\kappa(\boldsymbol{\theta}(t), X)$ will be upper bounded by a converged number $\kappa(\boldsymbol{\theta}(t \to \infty), X) < C_\infty < \infty$. So sharpness will not explode.

# F  COMPARISON TO FEATURE ROBUSTNESS

## F.1  THEIR RESULTS

**Definition F.1** (Feature robustness Petzka et al. Petzka et al. (2021)). *Let* $\ell : \mathcal{Y} \times \mathcal{Y} \to \mathbb{R}_+$ *denote a loss function,* $\epsilon$ *and* $\delta$ *two positive (small) real numbers,* $S \subseteq \mathcal{X} \times \mathcal{Y}$ *a finite sample set, and* $A \in \mathbb{R}^{m \times m}$ *a matrix. A model* $f(x) = (\psi \circ \phi)(x)$ *with* $\phi(\mathcal{X}) \subseteq \mathbb{R}^m$ *is called* $((\boldsymbol{\delta}, S, A), \epsilon)$-*feature robust, if* $\left| \mathcal{E}_{\mathcal{F}}^\phi(f, S, \alpha A) \right| \leq \epsilon$ *for all* $0 \leq \alpha \leq \delta$. *More generally, for a probability distribution* $\mathcal{A}$ *on perturbation matrices in* $\mathbb{R}^m$, *we define*

$$\mathcal{E}_{\mathcal{F}}^\phi(f, S, \mathcal{A}) = \mathbb{E}_{A \sim \mathcal{A}} \left[ \mathcal{E}_{\mathcal{F}}^\phi(f, S, A) \right],$$

*and call the model* $((\boldsymbol{\delta}, \boldsymbol{S}, \mathcal{A}), \boldsymbol{\epsilon})$-*feature robust on average over* $\mathcal{A}$, *if* $\left| \mathcal{E}_{\mathcal{F}}^\phi(f, S, \alpha \mathcal{A}) \right| \leq \epsilon$ *for* $0 \leq \alpha \leq \delta$

**Theorem F.2** (Theorem 5 Petzka et al. Petzka et al. (2021)). *Consider a model* $f(x, \mathbf{w}) = g(\mathbf{w}\phi(x))$ *as above, a loss function* $\ell$ *and a sample set* $S$, *and let* $O_m \subset \mathbb{R}^{m \times m}$ *denote the set of orthogonal matrices. Let* $\delta$ *be a positive (small) real number and* $\mathbf{w} = \omega \in \mathbb{R}^{d \times m}$ *denote parameters at a local minimum of the empirical risk on a sample set* $S$. *If the labels satisfy that* $y(\phi_{\delta A}(x_i)) = y(\phi(x_i)) = y_i$ *for all* $(x_i, y_i) \in S$ *and all* $\|A\| \leq 1$, *then* $f(x, \omega)$ *is* $((\delta, S, O_m), \epsilon)$-*feature robust on average over* $O_m$ *for* $\epsilon = \frac{\delta^2}{2m} \kappa^\phi(\omega) + \mathcal{O}(\delta^3)$.

**Proof sketch**

1. With assumption that $y[\phi_{\delta A}(x_i)] = y_i$ for all $(x_i, y_i) \in S$ and all $\|A\| \leq 1$, feature perturbation around $w$ is

$$\mathcal{E}_{\mathcal{F}}^{\phi}(f, S, \alpha A) + \mathcal{E}_{emp}(\boldsymbol{w}, S) = \mathcal{E}_{emp}(\boldsymbol{w} + \alpha \boldsymbol{w} A, S)$$

2. Since Taylor expansion for local minimum $\boldsymbol{w} = \omega$ will only remains second order term, thus

$$\mathcal{E}_{emp}(\boldsymbol{w} + \alpha \boldsymbol{w} A, S) = \mathcal{E}_{emp}(\omega, S) + \frac{\alpha^2}{2} \sum_{s,t=1}^{d} (\omega_s A) H_{s,t}(\omega, \phi(S))(\omega_t A)^{\top}$$

3. With basic algebra, one can easily get

$$\mathbb{E}_{A \sim O_m}\left[\mathcal{E}_{\mathcal{F}}^{\phi}(f, S, \alpha A)\right] \leq \frac{\delta^2}{2} \sum_{s,t=1}^{d} \mathbb{E}_{A \sim O_m}\left[(\omega_s A) H_{s,t} (\omega_t A)^T\right] + \mathcal{O}\left(\delta^3\right)$$

$$= \frac{\delta^2}{2m} \sum_{s,t=1}^{d} \langle \omega_s, \omega_t \rangle \cdot \mathrm{Tr}\left(H_{s,t}\right) + \mathcal{O}\left(\delta^3\right)$$

$$= \frac{\delta^2}{2m} \kappa^{\phi}(\omega) + \mathcal{O}\left(\delta^3\right)$$

### F.2 COMPARISON TO OUR RESULTS

We first summarize the commonalities and differences between their results and ours:

- Both of us consider the robustness of the model. But they define the feature robustness while we study the loss robustness Xu & Mannor (2012) which has been studied for many years.
- They consider a non-standard generalization gap by decomposing it into representativeness and the expected deviation of the loss around the sample points. But we strive to integrate sharpness into the general generalization guarantees.

For point 1, their defined feature robustness trivially depends on the sharpness. Because the *sharpness* (the curvature information) is just defined by the robust perturbation areas around the desired point. From step 2 in the above proof sketch we can see, the hessian w.r.t. $\omega$ is exactly the second expansion of perturbed expected risk. So we think this definition provides less information about the optimization landscape. In contrast, we consider the loss robustness for two reasons: 1) it is easy to get in practice without finding the orthogonal matrices $O_m$ first. 2) we highlight its dependence on the data manifold.

For point 2, we try to integrate this optimization property (sharpness) into the standard generalization frameworks in order to get a clearer interplay. Unlike feature robustness, the robustness defined by loss function will be easier analyzed in generalization tools, because it's hard and vague to define the "feature" in general. Besides, our result will also benefit the data-dependent bounds Xu & Mannor (2012); Kawaguchi et al. (2022).

## G EXPERIMENTS AND DETAILS

### G.1 RIDGE REGRESSION WITH DISTRIBUTIONAL SHIFTING

As we stated before, we followed the Duchi & Namkoong (2021) to investigate the ridge regression on distributional shift. We randomly generate $\theta_0^* \in \mathbb{R}^d$ in spherical space, and data from

$$X \overset{\text{iid}}{\sim} \mathcal{N}(0, 1), \quad \boldsymbol{y} = X \theta_0^* \tag{122}$$

To simulate distributional shift, we randomly generate a perpendicular unit vector $\theta_0^{\perp}$ to $\theta_0^*$. Let $\theta_0^{\perp}, \theta_0^*$ be the basis vectors, then shifted ground-truth will be computed from the basis by $\theta_{\alpha}^* = \theta_0^* \cdot \cos(\alpha) + \theta_0^* \cdot \sin(\alpha)$. For the source domain, we use $\theta_0^*$ as our training distribution. We randomly

sample 50 data points and train a linear classifier with a gradient descent of 3000 iterations. Starting from $\alpha = 0$, we gradually increase the $\alpha$ to generate different distributional shifts.

From the left panel in Figure 7 we can see that a larger penalty suffers from lower loss increasing when $\beta$ ranges from 0 to 2. Since we consider a cycling shift of label space, $180°$ corresponds to the maximum shift thus leading to the highest loss increase. According to our analysis of ridge regression, larger $\beta$ means a flatter minimum and more robustness, resulting in a better OOD generalization. This experiment verifies our theoretical analysis. However, it is important to note that too large a coefficient of $\ell_2$ regularization will hurt the performance. As shown in Figure 7 (right panel), the curse of underfitting (indicated by brown colors) appears when $\beta > 4$.

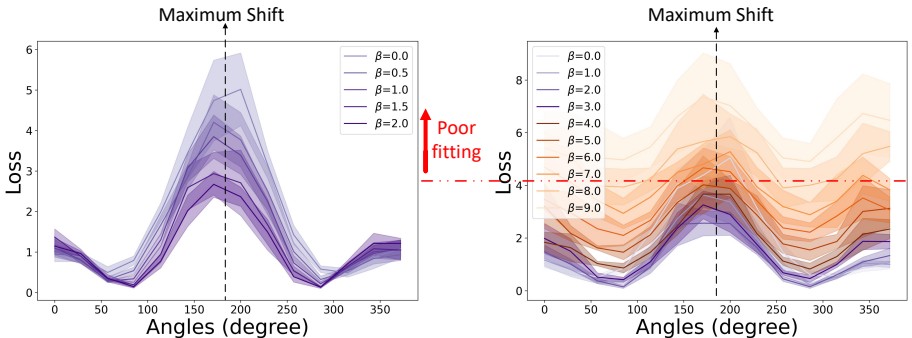

Figure 7: OOD test losses are increasing along distributional shifting. The X-axis is the shifting angle $\alpha$ and the Y-axis is the test loss of the model which is trained on distribution $\alpha = 0$. **Left:** The larger regularization $\beta$ causes a lower increase in test loss (Darker purple lines have lower test losses). **Right:** Too large $\ell_2$ penalty coefficient $\beta$ brings poor fitting and thus fails to generalize on both ID and OOD datasets (orange lines). Best viewed in colors.

### G.2 ADDITIONAL RESULTS ON SHARPNESS

Here we show more experimental results on RotatedMNIST which is a rotation of MNIST handwritten digit dataset with different angles ranging from $0°, 15°, 30°, 45°, 60°, 75°$ (6 domains/distributions). For each environment, we select a domain as a test domain and train the model on all other domains. Then OOD generalization test accuracy will be reported on the test domain. In our experiments, we run each algorithm with 12 different seeds, thus getting 12 trained models of different minima. Then we compute the sharpness (see Algorithm 1) for all these models and then plot them in Figure 8. For algorithms, we choose Empirical Risk Minimization (ERM), and Stochastic Weight Averaging (SWA) as the plain OOD generalization algorithm which is shown in the first column. In robust optimization algorithms, we choose Group Distributional Robust Optimization Sagawa et al. (2019). We also choose CORALSun & Saenko (2016) as a multi-source domain generalization algorithm. Among these different types of out-of-domain generalization algorithms, we can conclude that sharpness will affect the test accuracy on the OOD dataset.

Experimental configurations are listed in Table 1. For each model, we run the 5000 iterations and choose the last model as the test model. To ease the computational burden, we choose the 3-layer MLP to compute the sharpness.

| Algorithms | Optimizer | lr | WD | batch size | MLP size | eta | MMD $\gamma$ |
|---|---|---|---|---|---|---|---|
| ERM(SWA) | Adam | 0.001 | 0 | 64 | 265*3 | - | - |
| DRO | Adam | 0.001 | 0 | 64 | 265*3 | 0.01 | - |
| CORAL | Adam | 0.001 | 0 | 64 | 265*3 | - | 1 |

Table 1: Hyperparameters we use for different DG algorithms in the experiments.

From Figure 8 we can see, the sharpness has an inverse relationship with the out-of-domain generalization performance for every model in each individual environment. To make it clear, we plot similar

tasks from environment 1 to 4 as the last row. Thus, we can see a clearer tendency in all algorithms. It verified our Theorem 3.1. Note that all algorithms have different feature scales. One may need to normalize the results of different algorithms when plotting them together.

---

**Algorithm 1** Pseudocode of model sharpness computation

---

**Require:** feature layer $f(x)$, training loss $\ell$
**Ensure:** Sharpness $S$
  Get Jacobian matrix w.r.t. feature layer $\mathbf{J} = \nabla \ell(f(x), y)$
  **for** each gradient vector $\mathbf{J}_i$ in $\mathbf{J}^\top$ **do**
    Compute Hessian w.r.t. element $i, j$ of $f(x)$ by $\frac{\partial \mathbf{J}_i}{\partial f_l(x)}$
  **end for**
  We store Hessian in the variable $\mathbf{H}$
  Initialize sharpness $S = 0$
  **for** $i$ in feature layer.shape[0] **do**
    **for** $j$ in feature layer.shape[0] **do**
      Retrieve the hessian value of $i, j$ element via $h \leftarrow \mathbf{H}[:, j, i, :]$
      Sharpness $s_{ij} \leftarrow Trace(h) * f_{ij}(x)^2$
      $S = S + s_{ij}$
    **end for**
  **end for**

---

### G.3  COMPARE OUR ROBUST BOUND TO OTHER OOD GENERALIZATION BOUNDS

#### G.3.1  COMPUTATION OF OUR BOUNDS

First, we follow Kawaguchi et al. (2022) to compute the $K$ in an inverse image of the $\epsilon-$ covering in a randomly projected space. The main idea is to partition input space in a projected space with transformation matrix $\tilde{A}$. The specific steps will be (1) To generate a random matrix $\tilde{A}$, we i.i.d. sample each entry from the Uniform Distribution $\mathcal{U}(0, 1)$. (2) Each row of the random matrix $\mathcal{A} \in \mathbb{R}^{3 \times d}$ is then normalized so that $Ax \in [0, 1]^3$, i.e. $A_{ij} = \tilde{A}_{ij} / \sum_{j=1}^{d} \tilde{A}_{ij}$ (3) After generating a random matrix $A$, we use the $\epsilon$-covering of the space of $u = Ax$ to define the pre-partition $\{\tilde{C}_i\}_{i=1}^{K}$.

#### G.3.2  COMPUTATION OF PAC-BAYES BOUND

We follow the definition to compute expected $dis_\rho$ in Germain et al. (2013) where

**Definition G.1.** Let $\mathcal{H}$ be a hypothesis class. For any marginal distributions $D_S$ and $D_T$ over $X$, any distribution $\rho$ on $\mathcal{H}$, the domain disagreement $\mathrm{dis}_\rho (D_S, D_T)$ between $D_S$ and $D_T$ is defined by,

$$\mathrm{dis}_\rho (D_S, D_T) \stackrel{\mathrm{def}}{=} \left| \mathop{\mathbf{E}}_{h, h' \sim \rho^2} [R_{D_T}(h, h') - R_{D_S}(h, h')] \right|.$$

Since the $\mathrm{dis}_\rho (D_S, D_T)$ is defined as the expected distance, we can compute its empirical version according to their theoretical upper bound as follows.

**Proposition G.2** (Germain et al. Germain et al. (2013) Theorem 3)**.** *For any distributions $D_S$ and $D_T$ over $X$, any set of hypothesis $\mathcal{H}$, any prior distribution $\pi$ over $\mathcal{H}$, any $\delta \in (0, 1]$, and any real number $\alpha > 0$, with a probability at least $1 - \delta$ over the choice of $S \times T \sim (D_S \times D_T)^m$, for every $\rho$ on $\mathcal{H}$, we have*

$$\mathrm{dis}_\rho (D_S, D_T) \leq \frac{2\alpha \left[ \mathrm{dis}_\rho(S, T) + \frac{2\mathrm{KL}(\rho \| \pi) \ln \frac{2}{\delta}}{m \times \alpha} + 1 \right] - 1}{1 - e^{-2\alpha}}$$

. With $\mathrm{dis}_\rho(S, T)$, we can then compute the final generalization bound by the following inequality

$$\forall \rho \text{ on } \mathcal{H}, R_{P_T}(G_\rho) - R_{P_T}(G_{\rho_T^*}) \leq R_{P_S}(G_\rho)$$
$$+ \mathrm{dis}_\rho (D_S, D_T) + R_{D_T}(G_\rho, G_{\rho_T^*}) + R_{D_S}(G_\rho, G_{\rho_T^*})$$

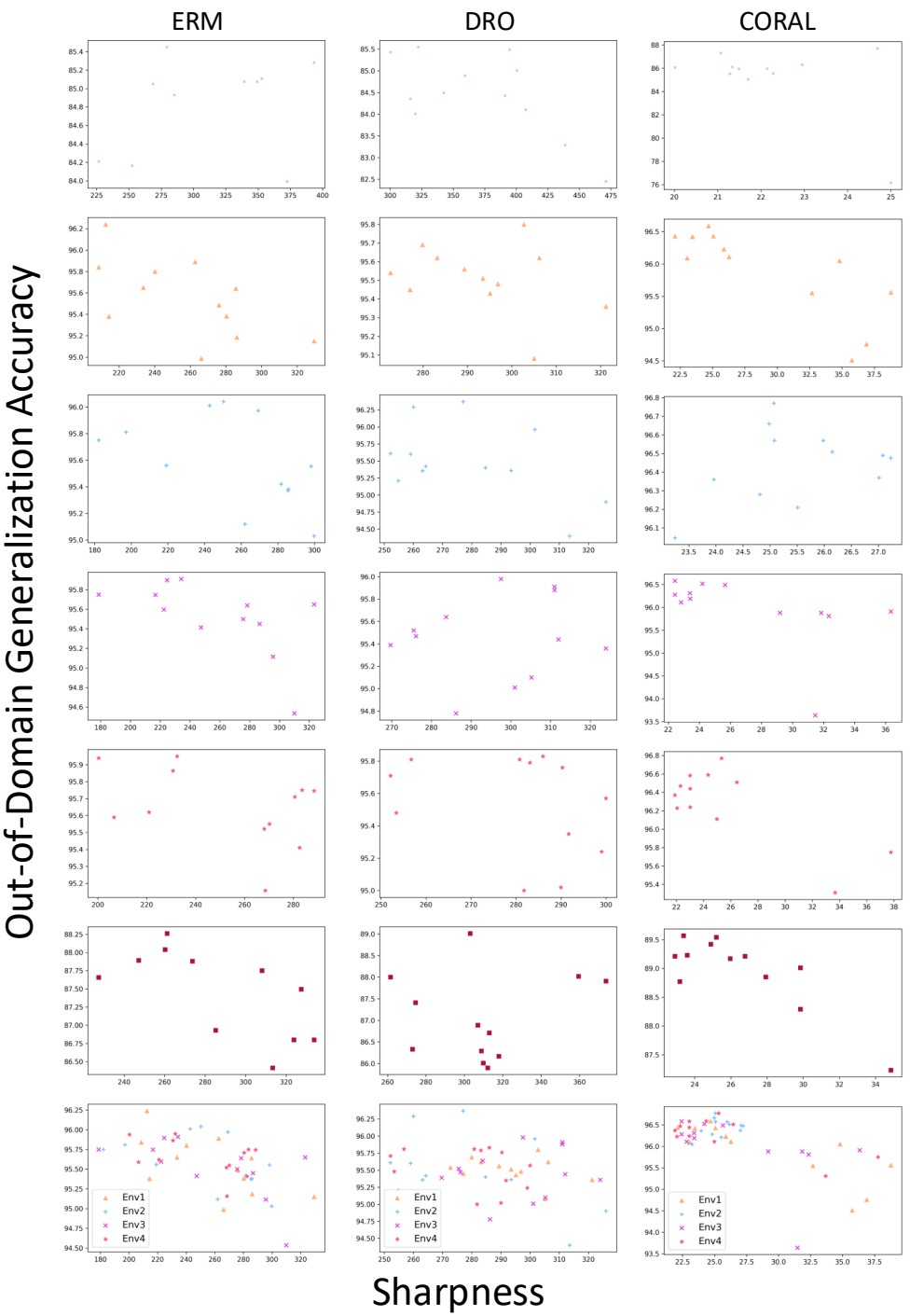

Figure 8: Domain generalization test accuracy on RotatedMNIST. From top to the bottom: environment 0, 1, 2, 3, 4, 5 with angles: $[0°, 15°, 30°, 45°, 60°, 75°]$ and a plot together. Each column shows the 12 runs of each algorithm.

with $\rho_T^* = \mathrm{argmin}_\rho R_{P_T}(G_\rho)$ is the best target posterior, and $R_D\left(G_\rho, G_{\rho_T^*}\right) = E_{h \sim \rho} E_{h' \sim \rho_T^*} R_D\left(h, h'\right)$.

Note that we ignore the expected errors over the best hypothesis by assuming the $R_D\left(G_\rho, G_{\rho_T^*}\right) = 0$. We apply the same operation in $\mathcal{E}^*$ of Proposition C.6 as well.

### G.3.3 COMPARISONS

In this section, we add some additional experiments on comparing to other baselines, i.e. PAC-Bayes bounds Germain et al. (2013). As shown in the first row of Figure 9, our robust framework has a smaller distribution distance in the bound compared to the two baselines when increasing the model size. In the second row, we have similar results in final generalization bounds. From the third and fourth rows we can see, our bound is tighter than baselines when suffering distributional shifts.

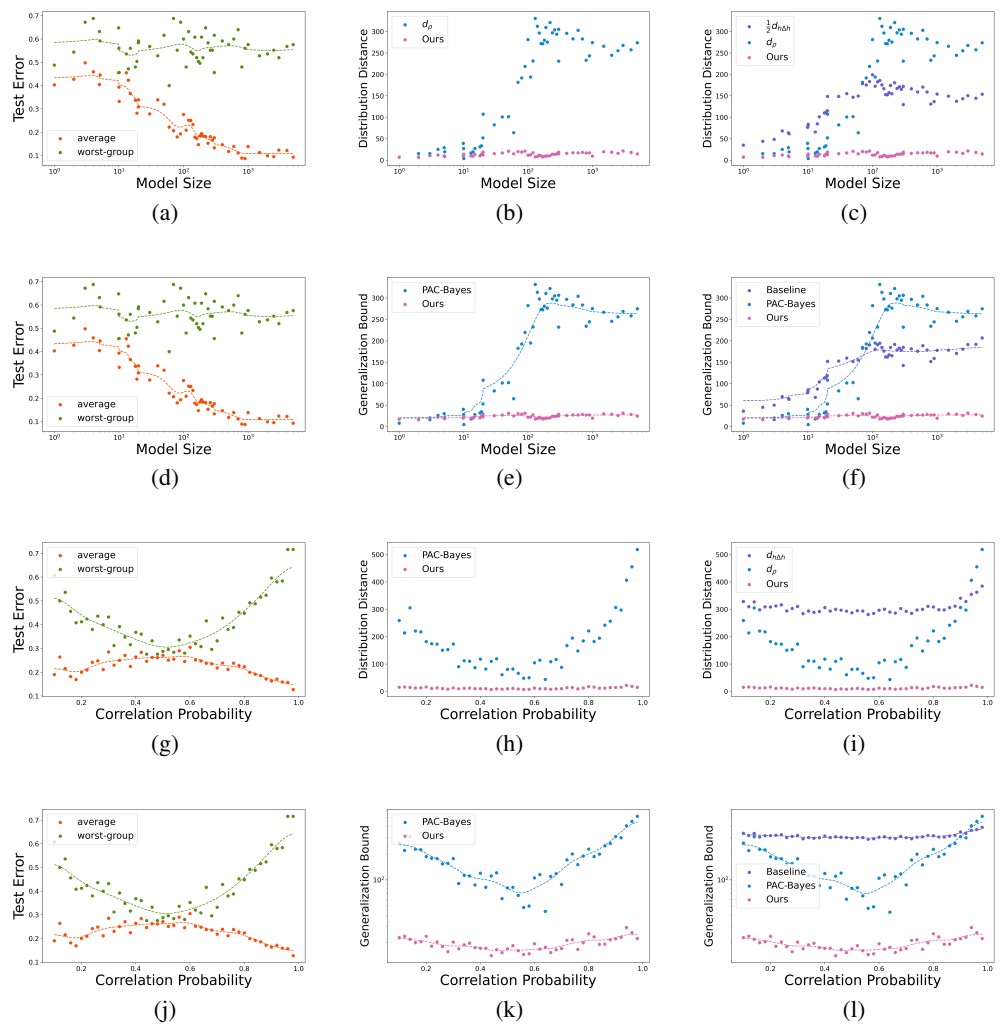

Figure 9: Additional spurious feature synthetic experiment. Each dot represents a trained model. The dash curves are the smoothed function fit by the test data points. The baseline is Proposition 2.1. **(a),(d),(g),(j)**: the generalization error of the logistic regression models with increasing the model size/correlation probability. **(b), (e)**: distribution distances for PAC-Bayes Germain et al. (2013) along model size increases. **(h), (k)**: distribution distances for PAC-Bayes Germain et al. (2013) along model distribution shift. **(c), (f), (i), (l)**: comparisons among baselines Proposition C.6, Germain et al. (2013) and ours. Note that model size > 500 is the overparameterized regime. The further the correlation probability is from 0.5, the greater the distributional shift is.

