# OpenReview forum: "Towards Robust Out-of-Distribution Generalization Bounds via Sharpness"
_ICLR.cc/2024/Conference — ICLR 2024 spotlight_

### Official Review · Reviewer_b2N6 · 2023-10-28

**Soundness:** 3 good
**Presentation:** 3 good
**Contribution:** 3 good
**Rating:** 8
**Confidence:** 4

**Summary:**

This paper gives a tight upper bound of out-of-distribution (OOD) by taking robustness into account and discusses the connection between robustness and sharpness of the loss landscape. The author proves the robust generalization bound by partitioning the sample space and measuring the distributional shift of each sub-group. As to the connection between robustness and sharpness, the author proves a theorem stating that sharper models give a larger upper bound on robustness, which aligns with the intuition that sharper models are more prone to overfitting, which hurts OOD generalization.

**Strengths:**

1. Using the partitioning method to achieve a tighter bound is interesting and also leads to a good result.
2. The paper is well-written and easy to understand, with clear explanations of key concepts and terminology.
3. The new distance metric better tolerates distributional shift compared with previous work.

**Weaknesses:**

The assumptions and constraints are a bit hard to meet and also may hurt the generalizability of the results.(See my questions)

**Questions:**

1. The author can give a clear explanation of ε(S) when describing definition of robustness (Def 2.2).
2. Have you considered the impact of different hyperparameters on the performance of your method? Can give some deeper discussion on this.

---

> ### Author Response · Authors · 2023-11-21
> **Thanks for your reviews**
>
> Thank you for the feedback and recognition of our contributions. Below are our responses.
>
> **Q1:The author can give a clear explanation of ε(S) when describing definition of robustness (Def 2.2).**
>
> Thanks for the nice suggestion. We have added the description
> > This definition captures the robustness of the model in terms of the input. Within each partitioned set
> $C_i$, the loss difference between any sample z belonging to $C_i$ and training sample $\boldsymbol{s} \in C_i$ will be
> upper bounded by a robustness constant $\epsilon(S)$. Hence, the number of partition $K$ and robustness constant $\epsilon(S)$ define robustness for learning models.
>
> below the Definition 2.2 in Section 2.2. Please refer to the highlighted revision in our new version.
>
> ---
>
> **Q2: Have you considered the impact of different hyperparameters on the performance of your method? Can give some deeper discussion on this.**
>
> Thanks for your nice suggestion. We conducted an ablation study on the spurious feature synthetic experiment with different numbers of partitions $K$ (Figure 7 in Appendix A.2). The results show that the bounds with partitioning ($K > 1$) can better capture the OOD generalization error changes along the distribution shifts.

---

### Official Review · Reviewer_q8BG · 2023-10-30

**Soundness:** 2 fair
**Presentation:** 1 poor
**Contribution:** 3 good
**Rating:** 6
**Confidence:** 3

**Summary:**

This paper provides generalization bounds for Out of Distribution Generalization based on the sharpness of the loss function minimum attained. One of the main ideas is to partition the input space according to the suggestion in [1]: it is assumed that the loss function doesn't vary too much inside each "relevant" subset of a partition of space (this is loosely related to the flatness of the minima) where "relevant" means close to an observed sample. Based on this assumption, the authors are able to bound (in Theorem 3.1)  the error as a sum of:

1 the empirical error in the source domain
2 the relaxed TV distance between the distributions in the partition considered
3 The robustness of the algorithm/loss (how much the loss can vary for points in a given partition element close to a sample
4 some function class capacity term.

The proof is a straightforward manipulation of the definitions.

The, corollary 3.2 attempts to show that the limiting behavior of Theorem 3.1 is the same as that of the main result of [1] which relies on the traditional TV distance.

In the next results, the authors rely on some ideas from [2] as well as brute force calculations to relate the concept of robustness of [1] and this work to that of the flatness of the local minima in the case of a kernel regression problem equivalent to a two layer neural network where only the last layer is trained.  The flatness and the norm of the parameter vectors are then claimed to be related in the case of ridge regression, and synthetic data experiments demonstrate superior OOD performance is associated with stronger genralization.







[1] Robustness and generalization. Huan Xu and Shie Mannor. Machine Learning. 2012.

[2] Henning Petzka, Michael Kamp, Linara Adilova, Cristian Sminchisescu, Mario Boley, "Relative Flatness and Generalization", NeurIPS 2021.













=====================Post rebuttal=================




Post-Rebuttal

I thank the authors for the rebuttal and for substantially improving the quality of the paper by fixing many of the errors I mentioned in the proofs.
I am not an expert in this particular field, so it is difficult for me to vouch for the correctness of all of the work, especially given that the first version had a very large number of errors that made reading impossible and necessitated a very deep revision. To the best of my understanding, the revision makes sense at a high level, but there are still minor errors, including both in parts of the paper I had not originally commented on, and parts of the paper that were rewritten for this revision (details below). **However, since those remaining errors appear minor and reviewers 2MLh and b2N6 are very positive about the paper and the authors have made an effort already, I am increasing my score from 5 to six.  In addition, since I have not found any fatal or serious errors in the revision, I am increasing my soundness rating to fair (from poor).**

However, I **very strongly recommend** the authors take at least two weeks of **full time** work to re-read all of the paper and **fix all the errors** and improve the proofs. To the best of my understanding, this is still necessary.  I don’t mean just fixing the errors that I caught, I mean fixing the ones that none of the reviewers caught.

**Strengths:**

Assuming they are correct, the results are certainly very interesting and promise to be the basis for further research. There is quite a lot of non trivial work in this paper, and many proofs require complicated calculations and take inspiration from a variety of existing work.

**Weaknesses:**

The appendix and many parts of the main paper very severely lack polishing to the extent that communication is difficult. Many of the proofs look more like a first draft than a serious submission. In particular, I can't vouch for the correctness of any results past Theorem 3.1 (which is not 100 percent correct either. In addition to poor language and presentation, the organization of the paper in terms of which assumptions are made is not good. The paper just needs more calm work and rewording: the authors of the appendix appear in a rush.


Here are more details:

First, although this is a minor point, it is worth mentioning that the constants in Theorem 3.1 don't seem to be fully correct. Indeed, the proof relies on a combination of Lemma 1 and Lemma C.3 from the arXiv version of [1], with both theorems relying on excluding a low probability event. Since combining both results requires a union bound, all the $\delta$s should be changed to $\delta/2$, resulting in different constants throughout. The same issued shows up in the proof of Proposition C.5. Note also that the result cited from [1] in Lemma C.3. is not in the published version, but in the arXiv version only. There are no Theorems 2 and 3.2 in the NeurIPS version, THis is Theorem 2 from NeurIPS and Theorem 3.2 from arXiv (
However, the same result can be lifter from [3] instead.) In addition, the combination of both results should be done explicitly since the constants are wrong again due to not applying the union bound. Also, I am really not sure the authors are actually using Theorem 3.2 rather than a simple generalization bound in terms of VC dimension.


Theorem 3.4 makes sense as a statement (though I couldn't check the full proof), but is very hard to read and contains plenty of minor errors. Why is $z_i(A)=argmax...$ unique? It seems the authors mean to say that $z_i\in argmax....$. When defining $\mathcal{M}_i$ as "the set of gobal minima in $C_i$", the authors actually mean that $\mathcal{M}_i$ as a set is the argmin of the objective function over $C_i$ (in particular, the elements are "global" minima inside the partition element $C_i$, but not necessarily global minima for the whole problem). This is absolutely not clearly written. It is also not clear why a different definition style is used for $|mathcal{M}_i$ and for $z-i(A)$ in that case Every statement in this paper is written vaguely using words instead of maths, adding to the burden of trying to make sense of what is attempted here.







==================Minor comments============


What is Theorem C.2 a restatement of? It seems it is Theorem 3.1 but it is not written (only the title "restate").

The wording of transitions  is quite sloppy. For instance, in page 14 we have the sentence "In order to give a fine-grained analysis, we follow the common choice where a two-layer NN function class is widely analysed in most literature, i.e. Neural Tangent Kernel, rich regime, random feature models." What do the authors mean by "rich regime"?

Note also at the top of page 14, the integral is not properly written as I believe the authors actually meant to write $\int f(x)g(y) \tau_d(d(x,y))$ instead of $\int f(x)g(y) \tau_d(dx)$  (otherwise the quantity is just equal to 1).


The statement of Corollary 3.2 is not really mathematically well defined. What the authors mean is that the bound from Theorem 3.1 converges to the RHS of equation (3), but what is written is just that the OOD generalization error is bounded by it. The "proof" is not very clean either.

The second sentence of Theorem 3.1 should contain a word such as "assume that".

The description of the case where $K=1$ above corollary 3.2 seems quite dodgy as it implies that the loss is approximately constant in the whole of input space!


Note that even the definition of one of the main concepts of this paper, i.e. robustness as in Definition 2.2, is not very well introduced: there "for all" which relates to $s,z$ is duplicated, and there is a "for all i$ after the $forall $s,z\in C_i$ (which depends on $i$). This makes this much harder to read than necessary. Note that the definition in [1] is much cleaner. It should be copied more faithfully.



====================Typos====================

There are plenty of issues with the language, not only in terms of improper use of grammar to an extent that harms understanding, but also in terms of polishing, spelling and punctuation. Below is a highly non exhaustive list.


In the abstract, fourth line "sharpness of learned minimum influences" should be "the sharpness of learned minima influences"

Page 2: to the left of the figure, "as shown in \cite... that when the loss lanscape...": a coma would be better than "that"

Every theorem which is lifted from a reference involves the text of the name of the authors written twice (once manually and once via the \cite function). This is probably due to copy and paste from another template with a different citation style without checking through it again. (prop. 2.1, definition 3.3

Page 21, "under this case" should be "in this case"

Equation (44), there is an equal sign with three bars instead of 2 where I think the authors actually mean equality in the strict sense.

Line 3 of Section 3.1 " $(K,\epsilon)$-robust" should be "..robustness..."

In page 5 (theorem 3.4), there is a "w.p. a" which probably should just be "w.p."


There are plenty of "w.r.t" and "i.i.d" (missing period) throughout the text.


In appendix D, note the use of "where definition of ....please refer to the corollary 3.5"

In page 18 at the beginning of the proof of Lemma D.1, although I agree that equation (29) makes sense, the sentence which introduces it speaks of the smallest eigenvalue whilst the equation itself shows the equivalent statement expressed as an PSD inequality between matrices.

Just below equation (27), it is unclear what the authors mean by saying that "$\mathcal{A}_{\mathcal{F}\Delta \mathcal{F}}$ represents the algorithm" WHat algorithm?


====================References===========================

[1] Robustness and generalization. Huan Xu and Shie Mannor. Machine Learning. 2012.

[2] Henning Petzka, Michael Kamp, Linara Adilova, Cristian Sminchisescu, Mario Boley, "Relative Flatness and Generalization", NeurIPS 2021.

[3] Learning Bounds for Domain Adaptation. John Blitzer, Koby Crammer, Alex Kulesza, Fernando Pereira, and Jennifer Wortman. NeurIPS 2007

**Questions:**

Could you write down a proof of  Proposition C.5 clarifying if you are using a standard VC bound or Theorem 3.2 from arxiv [1], and fixing the constants from the union bound?


Do you stand by your proof of Lemma D.5? It seems sloppy to the point of being completely incomprehensible, though I believe a similar result probably holds. Is equation (46) an equation or an approximation ? It seems that the probability (LHS) should be equal to $2\phi/\pi$ and not involvea $cos$ at all. For equation (47), it seems that you are calculating the bound the wrong way around: equation (47) holds if the inequality is reversed inside the probability statement. In the next substantial revision you upload, could you also include a precise restatement and citation for "Robbin's bounds".


What is going on in Equation (64)? The statement of Theorem 3.4 appears to be agnostic w.r.t. the choice of loss function, but the proof says "ohh we can verify that this holds for common loss functions such as L1, MSE, MAE etc.'. How do you get the first equality in (64), and does it hold for all loss functions? Please rewrite the proof accordingly.

Could you fix the "definitions" in corollary 3.5? When taken at face value, I maintain that $L_{x^*}=0$: there is something missing there. Also, although I guess that $\hat{w}_min$ is the minimum absolute value of any entry of $\hat{w}$ this is not clear from the definition since the notation $|\cdot|$ to mean element wise absolute value is non standard.



Please rewrite all the proofs much more carefully and upload a revision. It may help to ask another author to read the proofs.

---

> ### Author Response · Authors · 2023-11-21
> **Thanks for your detailed reviews.**
>
> Thank you for your valuable feedback and constructive suggestions. We have carefully reviewed and revised the proofs, as well as corrected any typos in the paper. Please find our responses to your comments below:
>
> ## Main weaknesses
>
> **W1: The constants in Theorem 3.1 don't seem to be fully correct.**
>
> Thank you for your careful review. We have corrected the constant in Theorem 3.1 as well as Proposition C.5. Following your suggestion, we have added the proof of Proposition C.5 and corrected the notions of VC-dimension by pseudo-dimension.
>
> ---
>
> **W2: Theorem 3.4 makes sense as a statement, but is very hard to read and contains plenty of minor errors.**
>
> We carefully reviewed all the proofs. Our revisions for Theorem 3.4 and Corollary 3.5 can be summarized below:
> - We have re-proven the Lemma D.1 by replacing the assumption $L_{x^*}>0$ by $m=Poly(d)$.
> - We have corrected the numbers in Lemma D.5.
> - We refine the definition of $\mathcal{M}_i$ and $z_i(A)$.
> - Revised the setting and assumptions in Appendix B.
> ----
>
> ## Minor comments
> For point 2, the term "rich regime" refers to a non-kernel regime in which training overparameterized networks through gradient descent leads to implicit biases that diverge from RKHS norms. We have supplemented this term with a previously omitted reference. For the remaining points, we have conducted the necessary revisions. Please consult our recently updated version for details.
>
> ----
>
> **[Typos]**
>
> Thank you for your careful review of our manuscript and for taking the time to point out the typographical errors in the original submission. The revised version of the manuscript has been uploaded with corrections highlighted in red.
>
> ## Questions
>
> **Q1: Could you write down a proof of Proposition C.5 clarifying if you are using a standard VC bound or Theorem 3.2 from arXiv [1], and fixing the constants from the union bound?**
>
> We have added proof of Proposition C.5 and corrected the error constant.
>
> ---
>
> **Q2: Lemma D.5**
>
> We have thoroughly revised the proof.
>
> ---
>
> **Q3: What is going on in Equation (64)?**
>
> We derived the first inequality by the chain rule. We made the choice of loss functions explicitly in Appendix B.1, please refer to the changes.
>
> ---
>
> **Q4: Could you fix the "definitions" in corollary 3.5?**
>
> We have refined our statement and introduced a supplementary assumption to Corollary 3.5, stipulating that the model size ($m$) is considerably greater than the data dimension ($d$). This new assumption is widely used in overparameterized regimes within which we have re-proven Lemma D.1. Revised Lemma D.1 has allowed us to present more precise and refined results for Corollary 3.5 in the revised version of our paper.
>
> ----
>
> Overall, we do appreciate your recognition of our novelty as well as for providing thorough and thoughtful reviews that improve the rigor of our manuscript. In response to your insightful comments and constructive feedback, we have revised the proofs thoroughly. We are grateful for the opportunity to improve our manuscript with your guidance.

---

> > ### Author Response · Authors · 2023-11-23
> > **Looking forward to further comments**
> >
> > Dear Reviewer q8BG,
> >
> > As the discussion window closes soon, we kindly request to know whether our revisions have addressed all your concerns. We sincerely appreciate that if you could recognize our efforts or provide any further comments.

---

> > > ### Comment · Reviewer_q8BG · 2023-12-05
> > >
> > > Thanks for fixing the union bounds and citation issues in Proposition C.5 and trying so hard to fix several of the proofs.  Thanks especially for the improved proof of Lemma D.5., which now seems mostly correct, though I still have concerns the exact numerical values might contain errors (see details).  I am raising my score to six but as written in my final review, please revise the paper before the camera ready version.
> > >
> > > Here are a few minor issues that still remain based on an incomplete reading of the revision:
> > >
> > > 1.	(minor) In equation (27), I think the $p$s shld be $p_t$ and $p_s$ (otherwise the RHS is just zero)
> > > 2.	(minor) In the third and fourth lines in equation (28), the $dt$s should be outside the absolute value.
> > > 3.	(minor) In the fourth line in equation (28), there shouldn’t be any $dt$s at all
> > > 4.	(minor) In Theorem 3.4, there is a capital letter after a coma (and no space) in the red, modified version.
> > > 5.	(somewhat minor) You should also remind the reader of what $R(d)$ is as it is only introduced in Lemma D5. The first sentence also needs to be rewritten
> > > 6.	(minor) What do you mean by the three bars quality? It seems that $\|x\|^2$ should just be equal to $R(d)$.
> > > 7.	(Lemma D.5., somewhat minor). It’s hard to process the first equality in equation (53) without some context, maybe you want to remind the reader of the fact that $\Gamma(n-\frac{1}{2})=\frac{(2(n-1))!}{2^{2(n-1)}(n-1)!}\sqrt{\pi}$?
> > > 8.	(Lemma D.5., somewhat minor) There is a missing $cos(\phi)$ in equation (55). Fortunately, it comes back in equation (56), but this should still be fixed.
> > > 9.	(Lemma D.5., not that minor) Equation (57) is a description of the consequences of the lemma, it doesn’t seem like it belongs in the proof.
> > > 10.	(Lemma D.5., somewhat minor) I think that $\grtapprox$ should be $\geq$ at equation (59): if that is not the case, then the proof is wrong.
> > > 11.	(Lemma D.5., medium severity) I apologize in advance if I am getting something wrong but in equation (58), to the best of my understanding there is an error. I can see how you are deriving the expression $\frac{2^{d-2} (\frac{d-3}{2})!^2}{(d-2)!\sqrt{\pi}}$ by analogy with the case where $d$ is even. But it seems to me that the correct formula should instead be $\frac{2^{d-1}  (\frac{d-1}{2})!(\frac{d-3}{2})!}  {(d-1)! \sqrt{\pi}}$. Indeed, you are presumably using the formula $\Gamma(\frac{1}{2}+n)=\frac{(2n)!}{2^{2n}n!}\sqrt{\pi}$ directly with $n$ being $\frac{d-1}{2}$ instead of using the reverse formula $\Gamma(n-\frac{1}{2})=\frac{(2(n-1))!}{2^{2(n-1)}(n-1)!}\sqrt{\pi}$ as in the (correct) calculation earlier in the proof in equation (53).

---

### Official Review · Reviewer_m8jH · 2023-10-31

**Soundness:** 2 fair
**Presentation:** 2 fair
**Contribution:** 2 fair
**Rating:** 5
**Confidence:** 4

**Summary:**

The authors introduce an out-of-distribution (OOD) generalization bound based on Sharpness that incorporates robustness, leading to a more precise bound compared to non-robust guarantees. The given results are validated through experiments conducted on both a ridge regression model and some deep learning classification tasks.

**Strengths:**

-	The given bound seem to be tighter than existing non-robust guarantees.
-	The experiments are well conducted on ridge regression and simple classification tasks using MLP.

**Weaknesses:**

-	The datasets being used are overly simplistic, making it difficult to ensure the validity of theoretical outcomes. Empirical evidence from [1], [2], [3], [4], [5] demonstrate that sharpness is limited in the context of small datasets like CIFA10 or SVHN. Comprehensive evaluations involving transformers, ConvNet, CLIP, and BERT  trained on larger datasets such as ImageNet and MNLI reveal a weak correlation between sharpness and generalization. Surprisingly, a consistent negative correlation between sharpness and out-of-distribution (OOD) generalization suggests that sharper minima might actually generalize better, contradicting the authors' proposed theory. Moreover, experiments in [6] indicate that the choice of sharpness definition heavily depends on the specific dataset. To substantiate their theoretical claims, it is essential for the authors to conduct more extensive experiments on larger and more intricate datasets using complex models.

-	Numerous studies propose that flatter minima might offer better generalization for both standard and out-of-distribution (OOD) data such as [7] and [8]. However, traditional sharpness definitions often fall short in capturing generalization accurately, potentially because they are not consistent under reparametrizations that do not change the model itself. To tackle these issues, Adaptive Sharpness and reparametrization-invariant sharpness have been developed. I anticipate seeing the theoretical results related to Adaptive Sharpness and reparametrization-invariant sharpness.

[1] Kwon, J., Kim, J., Park, H., and Choi, I. K. Asam: Adaptive sharpness-aware minimization for scale-invariant learning of deep neural networks. ICML, 2021.
[2] Keskar, N. S., Mudigere, D., Nocedal, J., Smelyanskiy, M., and Tang, P. T. P. On large-batch training for deep learning: Generalization gap and sharp minima. ICLR, 2016.
[3] Jiang, Y., Neyshabur, B., Mobahi, H., Krishnan, D., and Bengio, S. Fantastic generalization measures and where to find them. ICLR, 2020.
[4] Dziugaite, G. K. and Roy, D. Entropy-sgd optimizes the prior of a pac-bayes bound: Generalization properties of entropy-sgd and data-dependent priors. In ICML, pp. 1377–1386. PMLR, 2018.
[5] Bisla, D., Wang, J., and Choromanska, A. Low-pass filtering sgd for recovering flat optima in the deep learning optimization landscape. AISTATS, 2022.
[6] Andriushchenko, Maksym, et al. "A modern look at the relationship between sharpness and generalization." arXiv preprint arXiv:2302.07011 (2023).
[7] Xing, C., Arpit, D., Tsirigotis, C., and Bengio, Y. A walk with sgd. arXiv preprint arXiv:1802.08770, 2018.
[8] Zhou, P., Feng, J., Ma, C., Xiong, C., Hoi, S. C. H., et al. Towards theoretically understanding why SGD generalizes better than Adam in deep learning. NeurIPS, 2020.

**Questions:**

Furthermore, transfer learning has become the standard choice for vision and language tasks. However, there is limited understanding of sharpness in this context. Do the theoretical results still apply to transfer learning scenarios?

---

> ### Author Response · Authors · 2023-11-21
> **Thank you for your reviews**
>
> Thank you for the feedback and suggestions. Below are our responses.
>
> **Q1: Empirical evidence from existing work demonstrates that sharpness is limited in the context of small datasets like CIFA10 or SVHN**
>
> [1] has demonstrated that Stochastic Weight Averaging towards flat minima can improve OOD generalization on five standard DG benchmarks, namely PACS, VLCS, OfficeHome, TerraIncognita, and DomainNet. In addition, we have observed a **consistent phenomenon on larger-scale datasets** such as PACS, and WILDS-Camelyon17. Please refer to Figure 5,6 in Appendix A in our revision.
>
> -------
>
> **Q2: [3] shows a negative correlation between sharpness and out-of-distribution (OOD) generalization suggesting that sharper minima might actually generalize better, contradicting the authors' proposed theory.**
>
> The distinction in the definition of sharpness can lead to disparate observations and interpretations. It is crucial to note that our measure (Definition 3.3) is different from the one presented in [3] where its sharpness is defined by
> $$S_{avg}^\rho(w, c):=  E_{S \sim P_m, \delta \sim(0, \rho^2 \text{diag}(c^2)} L_{S}(w+\delta)-L_{S}(w) ~~~~(1)$$
> which captures the loss changes within a neighbor area. In contrast, our sharpness (following [4]) integrates the norm of weights and the trace of the Hessian matrix for each layer:
> $$\kappa(w, S, A):= \langle w, w \rangle \text{tr}(H_{S, A}(w)). ~~~(2)$$
> Our theoretical results derived the dependence of sharpness and robustness upon Eqn(2). Both experiments in [1] and our paper have validated the positive correlation between flatness and OOD generalization.
>
> -------
>
> **Q3: It is essential for the authors to conduct more extensive experiments on larger and more intricate datasets using complex models.**
>
> Thanks for the nice suggestion. To better evaluate the correlation between our sharpness and Out-Of-Distribution (OOD) generalization, we conducted additional experiments on PACS with ResNet50, and on Wilds-Camelyon17 with DenseNet121 according to your suggestion (Figure 5 Appendix A in our revision).
>
> -------
>
> **Q4: I anticipate seeing the theoretical results related to Adaptive Sharpness and reparametrization-invariant sharpness.**
>
> We have adopted the "relative sharpness" defined in [4] to account for **reparameterization-invariant sharpness** that alters certain flatness measures while leaving generalization unaffected. Moreover, [4] has established the rigorous link between sharpness and generalization. Works on adaptive sharpness such as ASAM [2] and [3] examine reparametrization-invariant sharpness from a different perspective. The differences we have shown in Q2.
>
> -------
>
> **Q5: Do the theoretical results still apply to transfer learning scenarios?**
>
> Yes, we believe our results are applicable to transfer learning contexts, specifically domain adaptation. The framework of our bound is commonly shared in domain adaptation [5].
>
> -------
>
> **References**
>
> [1] Cha, Junbum, et al. "Swad: Domain generalization by seeking flat minima." Advances in Neural Information Processing Systems 34 (2021): 22405-22418
>
> [2] Kwon, Jungmin, et al. "Asam: Adaptive sharpness-aware minimization for scale-invariant learning of deep neural networks." International Conference on Machine Learning. PMLR, 2021.
>
> [3] Andriushchenko, Maksym, et al. "A modern look at the relationship between sharpness and generalization."
>
> [4] Petzka, Henning, et al. "Relative flatness and generalization." Advances in neural information processing systems 34 (2021): 18420-18432.
>
> [5] Redko, I., Morvant, E., Habrard, A., Sebban, M., & Bennani, Y. (2020). A survey on domain adaptation theory: learning bounds and theoretical guarantees. arXiv preprint arXiv:2004.11829.

---

> > ### Author Response · Authors · 2023-11-23
> > **Looking forward to further comments**
> >
> > Dear Reviewer m8jH,
> >
> > As the discussion window closes soon, we kindly request to know whether our responses have addressed all your concerns. We sincerely appreciate that if you could provide any further comments.

---

> > > ### Comment · Reviewer_m8jH · 2023-12-03
> > > **Response to the rebuttal**
> > >
> > > I thank the authors for addressing my comments. I have raised my score to reflect the effort of the authors during the rebuttal phase.

---

### Official Review · Reviewer_2MLh · 2023-11-04

**Soundness:** 3 good
**Presentation:** 3 good
**Contribution:** 2 fair
**Rating:** 8
**Confidence:** 3

**Summary:**

This paper studies the relationship between robustness and sharpness in improving OOD generalization performance. The authors provided rigorous theoretical analyses through the PAC learning framework to give an intuitive insight into why minimizing sharpness can lead to better OOD performance. This paper is the first work to reveal the benefits of flatness to OOD generalization. Moreover, based on existing theoretical results, a titer error bound is proposed. Empirically, the authors choose two simple datasets RotateMNIST and a self-generated dataset to validate the findings of their theory. This is a pure theoretical paper which is a solid contribution to related fields.

**Strengths:**

- The theoretical derivation is rigorously demonstrated. It is solid work by providing support for encouraging flat minima on OOD generalization problems.
- All the results are very intuitive and confirm many assumptions of my own.
- This paper is well-written and easy to follow.

**Weaknesses:**

- Why conducting partition on all training data is still not clear to me. It is reasonable that a small distribution shift in the same sub-group could lead to noise and should be ignored. But how such a strategy is formulated is unknown. Besides, could it be possible to conduct an ablation study to justify that such kind of partition is indeed helpful for OOD generalization?
- The results in Figure 4 are interesting. However, some intuitive explanations are missing. For example, why did the test error of the worst-case group first decrease along the correlation probability and then increase?
- It would be much more helpful if the authors could provide the experimental results on larger datasets such as DomainBed and WILDS. I understand theoretical papers could still have gaps in realistic application, but it would be helpful for us to find problems and look for possible directions.

**Questions:**

Please see the weaknesses for detials.

---

> ### Author Response · Authors · 2023-11-21
> **Thank you for your reviews.**
>
> Thank you for the feedback and suggestions. Below are our responses.
>
> **W1-(a): Why conducting partition on all training data is still not clear to me.**
>
> According to the original definition in [1], the (whole) sample space $\mathcal{Z}$ is partitioned into $K$ disjoint sets, not just on training data such that all unknown samples from OOD may fall into the same set as the training data. An example of formulation can be found in Example 1 in [1] where
>
> > Example 1 Fix $\gamma>0$ and put $K=2 N (\gamma / 2, X, ||\cdot||_2 )$. If $\mathcal{A}s$ has a margin $\gamma$, then $\mathcal{Z}$ can be partitioned into $K$ disjoint sets,  such that if $s_j$ and $z \in \mathcal{Z}$ belong to a same $C_i$, then $|l(\mathcal{A}s, s_j)-l(\mathcal{A}s, z)|=0$.
>
> This example suggests that according to the covering number, sample space $X$ can be partitioned into $N (\gamma / 2, X, ||\cdot||_2 )$ subsets such that each subset has a diameter less or equal to $\gamma$.
>
>
> **W1-(b): Could it be possible to conduct an ablation study to justify that such kind of partition is indeed helpful for OOD generalization?**
>
> Yes, we conducted an ablation study on the spurious feature synthetic experiment with different numbers of partitions $K$ (Figure 7 in Appendix A.2). The results show that the bounds with partitioning ($K > 1$) can better capture the OOD generalization error changes along the distribution shifts.
>
> ----
>
> **W2: The results in Figure 4 are interesting. However, some intuitive explanations are missing.**
>
> Thanks for the nice suggestions, we added the following intuitive explanation on Page 9. Please refer to our highlighted revision.
>
> > As shown in Figure 4(d), when $p_{\text{maj}} = 0.5$, there is no distributional shift between training and test data due to no spurious features correlated to training data. Thus, the training and test distributions align closer and closer when $p_{\text{maj}}<0.5$ and increase, resulting in an initial decrease in the test error for the worst-case group.
> However, as $p_{\text{maj}} > 0.5$ and deviates from 0.5, introducing the spurious features, a shift in the distribution occurs. This deviation is likely to impact the worst-case group differently, leading to an increase in the test error.
>
> ----
>
> **W3: Experimental results on larger datasets such as DomainBed and WILDS.**
>
> We have added the new experimental results on PACS and WILDS-Camelyon17. To extend the sharpness measure to complicated architectures, such as DenseNet-121, we compute the Hessian matrix from the last FC layer. Even though, the result is still consistent with what we report in Figure 3. The phenomenon that **sharper minima can lead to poor OOD generalization performance** can also be observed on large-scale datasets.
>
> ---
>
> **References**
>
> [1] Xu, H., & Mannor, S. (2012). Robustness and generalization. Machine learning, 86, 391-423.

---

> ### Comment · Reviewer_2MLh · 2023-11-22
> **Thanks for your reply**
>
> The authors have carefully answered my questions. Additional experiments are also added to show its empirical effectiveness. I think this paper shows some insight into OOD generalization. The theoretical analyses are extensive. Therefore, I would like to raise my score to 8, but my confidence remains at 3.

---

> > ### Author Response · Authors · 2023-11-22
> > **Thanks for your reply**
> >
> > Thank you very much for your nice suggestions. We are pleased to hear that our efforts have addressed your questions.

---

### Meta-Review · Area_Chair_fdzx · 2023-12-06

**Metareview:**

This paper establishes a theoretical link between generalizing to out-of-distribution (OOD) data and the sharpness of the minimum of the learned model, by relating the former to algorithmic robustness. This is an important step as there is a growing list of empirical evidence suggesting a connection between flat minima and OOD performance, but a provable connection is mostly missing. The paper verifies the correctness of its theory by semi-analytically studying a ridge-regression problem and empirically on DomanBed setting with deep neural networks. Majority of reviewers find the contributions of this submission significant and recommend accept. In concordance with them, I believe this work is a timely study of the interplay between robustness, generalization and loss geometry, and I recommend accept. Please incorporate reviewers' comments in the final version of the paper.

**Justification For Why Not Higher Score:**

The paper would have been stronger if its theory was verified on more commonly used architectures such as CNNs and ResNets. Currently the deep learning experiments use a 4-layer MLP.

**Justification For Why Not Lower Score:**

The theoretical insights from the paper are very interesting, and are verified in a ridge regression scenario as well as some limited deep learning scenarios.

---

### Decision · Program_Chairs · 2024-01-16

Accept (spotlight)